# RocketKV: Accelerating Long-Context LLM Inference via Two-Stage KV Cache Compression

Payman Behnam [* 1 2]   Yaosheng Fu [* 1]   Ritchie Zhao [1]   Po-An Tsai [1]   Zhiding Yu [1]   Alexey Tumanov [2]

## Abstract

Transformer-based Large Language Models rely critically on the KV cache to efficiently handle extended contexts during the decode phase. Yet, the size of the KV cache grows proportionally with the input length, burdening both memory bandwidth and capacity as decoding progresses. To address this challenge, we present RocketKV, a training-free KV cache compression strategy containing two consecutive stages. In the first stage, it performs coarse-grain permanent KV cache eviction on the input sequence tokens. In the second stage, it adopts a hybrid sparse attention method to conduct fine-grain *top-k* sparse attention, approximating the attention scores by leveraging both head and sequence dimensionality reductions. We show that RocketKV provides a compression ratio of up to $400\times$, end-to-end speedup of up to $3.7\times$ as well as peak memory reduction of up to 32.6% in the decode phase on an NVIDIA A100 GPU compared to the full KV cache baseline, while achieving negligible accuracy loss on a variety of long-context tasks. We also propose a variant of RocketKV for multi-turn scenarios, which consistently outperforms other existing methods and achieves accuracy nearly on par with an oracle *top-k* attention scheme. The source code is available here: https://github.com/NVlabs/RocketKV.

## 1. Introduction

In Transformer-based LLM inference (Vaswani et al., 2017; Pawar et al., 2024), the key-value cache (KV cache)—which stores past attention keys and values to avoid recompu-

tation—becomes a major bottleneck during the decode phase, as its size scales linearly with both the sequence length and batch size. For example, the Llama3.1-70B-Instruct (AI@Meta, 2024) model with a batch size of 32, and a context length of 32K requires around 320GB of KV cache storage at FP16 precision, which even advanced hardware (NVIDIA, 2024; Jouppi et al., 2023; AMD, 2024) can hardly handle.

Fortunately, previous work has shown that only a small subset of KV tokens is required at each decode step to maintain accuracy (Zhang et al., 2023; Tang et al., 2024; Ribar et al., 2024; Li et al., 2024; Singhania et al., 2024; Ge et al., 2024). Therefore, if those KV tokens can be accurately predicted in advance, dense attention operations can be replaced with sparse attention operations with significant memory bandwidth and capacity improvement. These methods often fall into two categories: 1) permanent KV token eviction, and 2) dynamic KV token selection. The former results in both memory bandwidth and storage savings, but could lead to noticeable accuracy loss if KV tokens dropped earlier are needed by later decode steps. The latter avoids this shortcoming by keeping all KV tokens in memory and dynamically selecting a subset each time. Hence, it only results in memory bandwidth savings, but often requires extra memory storage overhead for auxiliary data.

To understand the effectiveness of existing KV token dropping methods, Figure 1 presents the accuracy comparison of four different methods (two in each category) for *qasper* benchmark in LongBench (Bai et al., 2024) on the Mistral-7B-Instruct-v0.2 model (Jiang et al., 2023). We observe that the accuracy of all four methods drops significantly as the token budget becomes lower than 1024 while an oracle *top-k* attention scheme achieves negligible accuracy drop even with a token budget of 256. This indicates that existing practical methods fail to accurately predict *top-k* KV tokens under low token budgets.

To improve the prediction accuracy, we propose RocketKV, a two-stage KV cache compression method that combines permanent KV token eviction with dynamic KV token selection to accelerate the decode phase of LLM inference. The RocketKV framework enables flexible integration of a wide range of KV cache compression techniques at each stage. To

---

*Equal contribution: work done during internship/part-time at NVIDIA  [1]NVIDIA, Santa Clara, USA [2]Georgia Institute of Technology, Atlanta, USA. Correspondence to: Payman Behnam <payman.behnam@gatech.edu>, Yaosheng Fu <yfu@nvidia.com>.

*Proceedings of the $42^{nd}$ International Conference on Machine Learning*, Vancouver, Canada. PMLR 267, 2025. Copyright 2025 by the author(s).

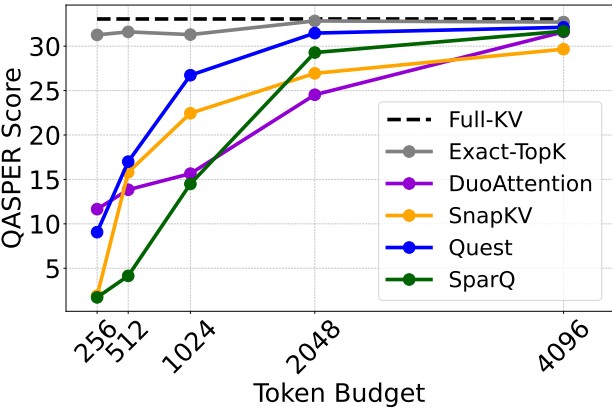

Figure 1: Existing KV token dropping methods fail to match the accuracy scores of oracle *top-k* attention (Exact-TopK) on Mistral-7B-Ins-v0.2 in the *qasper* benchmark.

achieve optimal performance, we directly adopt an existing method SnapKV (Li et al., 2024) for coarse-grain permanent KV token eviction in the first stage. In the second stage, we propose hybrid sparse attention (HSA) to perform fine-grain dynamic KV token selection, which estimates KV token indices with *top-k* attention scores via two-dimensional reductions. To balance the compression between two stages, RocketKV introduces an adaptive compression decomposition mechanism that intelligently splits any given target compression ratio across two stages. Combining these two stages together, RocketKV achieves significant memory bandwidth and storage savings with negligible accuracy loss across a wide variety of models and downstream tasks.

Moreover, recent work (Li et al., 2025) demonstrates that permanent KV token eviction suffers in multi-turn decoding because important KV tokens vary significantly across queries. To address this challenge, we propose a variant of RocketKV called RocketKV-MT for multi-turn scenarios where we do not evict the unselected KV tokens in the first stage but keep them all for later turns. Yet, the decode phase is still restricted to perform dynamic selection on the filtered KV tokens from the first stage in each turn. By doing this, RocketKV-MT achieves the same memory traffic savings as RocketKV but does not introduce memory storage savings.

In summary, we make the following contributions:

- We analyze and identify the limitations of existing KV token dropping methods and then propose RocketKV as a two-stage KV cache compression scheme. We further introduce a variant RocketKV-MT for multi-turn decoding scenarios.

- We propose a hybrid sparse attention (HSA) for dynamic KV token selection with two-dimensional reductions in the second stage while directly adopting

SnapKV in the first stage. We design an adaptive compression decomposition mechanism to balance the compression between the two stages.

- We conduct a comprehensive evaluation of RocketKV and RocketKV-MT on a wide variety of models and downstream tasks. RocketKV consistently demonstrates comparable accuracy with full KV attention at up to $400\times$ compression ratio, while achieving up to $3.7\times$ end-to-end speedup and 32.6% peak memory saving at the decode phase on an NVIDIA A100 GPU. Meanwhile, RocketKV-MT is more suitable for multi-turn decoding and performs on par with oracle *top-k* attention.

## 2. Related Work

A number of recent approaches have focused on improving the efficiency of attention mechanisms in LLMs, particularly when dealing with long contexts. One feasible solution is KV cache sharing across multiple layers (Brandon et al., 2024) or selectively dropping attention for some layers (Ling et al., 2024). Others (Sun et al., 2024; Ho et al., 2024) propose mixed attention designs where some layers use global attention while others use local attention. Multi-Query Attention (MQA) (Shazeer, 2019), Grouped-Query Attention (GQA) (Ainslie et al., 2023), and Multi-head Latent Attention(MLA) (Liu et al., 2024a)are widely adopted by many recent LLMs (AI@Meta, 2024; Jiang et al., 2023; Team et al., 2024; Liu et al., 2024b), which reduce the KV cache by sharing key-value pairs across multiple attention heads. All above techniques directly modify the attention architecture and require integration since pre-training.

Another direction is to improve attention efficiency with training-free techniques. These techniques can be further categorized as prefill phase acceleration, decode phase acceleration, or accelerating both phases together. For example, StreamingLLM (Jiang et al., 2023) combines initial and local-window attention to reduce the KV cache into a constant size regardless of the sequence length. DuoAttention (Xiao et al., 2025) and RazorAttention (Tang et al., 2025) improve upon StreamingLLM by applying global attention on retrieval heads and StreamingLLM-style attention on rest heads. While these techniques can be applied to both the prefill and decode phases, other techniques focus on accelerating only one of the two phases.

For prefill phase acceleration, MInference (Jiang et al., 2024) identifies three distinct patterns in the attention matrix that can be harnessed for efficient sparse operations with customized GPU kernels. SeerAttention (Gao et al., 2024) explores dynamic block-level sparsity in the attention module with a learnable gate. Meanwhile, XAttention (Xu et al., 2025) leverages the sum of antidiagonal values in the

attention matrix for estimating block importance.

A common approach for attention acceleration at the decode phase is through permanent KV cache eviction, which can save both memory bandwidth and storage requirements. H2O (Zhang et al., 2023) observes that a small subset of tokens, known as heavy-hitters, dominates the attention computation, thus only keeps recent and heavy-hitter tokens. SnapKV (Li et al., 2024) employs an observation window at the end of the prompt to identify critical KV tokens of the input prompt. Then it uses a clustering algorithm via pooling to retain critical KV token clusters without losing information completeness. Ada-KV (Feng et al., 2024) proposes an adaptive budget allocation strategy to provide better token budget utilization across individual heads.

Meanwhile, Quest (Tang et al., 2024) observes that permanent KV cache eviction could lead to inevitable accuracy loss and proposes query-aware selection on *top-k* KV tokens based on approximation attention with representative vectors of contiguous key cache pages. On the other hand, SparQ and Loki (Ribar et al., 2024; Singhania et al., 2024) conduct approximation attention by selecting only important indices on the head dimension instead. The above approaches can save KV cache compute and data fetching from memory, but not KV cache storage. InfiniGen (Lee et al., 2024) tackles this challenge by offloading the entire KV cache to CPU memory and only fetching the selected KV tokens to GPU memory when needed. MagicPIG(Chen et al., 2025) discovers that using importance sampling is more efficient than *top-k* estimation and proposes an approximation attention solution leveraging locality sensitive hashing (LSH) and CPU offloading.

In contrast to prior work, RocketKV combines both permanent KV cache eviction and dynamic KV token selection as two consecutive stages for accelerating the decode phase and utilizes the advantages of both worlds. As a result, it achieves remarkable memory bandwidth and storage savings without the need for sophisticated system-level optimizations such as CPU offloading.

# 3. Proposed Method: RocketKV

## 3.1. Observation

As shown previously in Figure 1, existing token dropping methods, regardless of permanent KV token eviction or dynamic KV token selection, fail to match the accuracy of oracle *top-k* attention (Exact-TopK) under low token budgets. To further understand what causes the accuracy mismatch, we analyze a random attention head (layer 31 head 0) from Mistral-7B-Instruct-v0.2 and present the cumulative distribution function (CDF) of both the maximum sequence length and number of unique KV indices selected by Exact-TopK (k=256) across all decoding steps in the

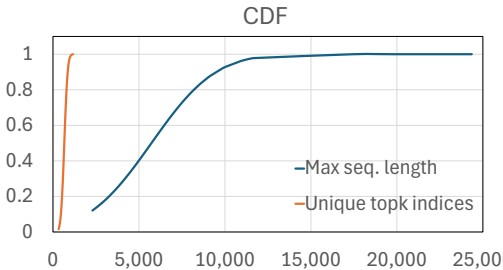

Figure 2: The CDF of maximum sequence length and number of unique *top-k* indices (k=256) across all decoding steps for 200 questions in the *qasper* benchmark. Data was collected from a random head (layer 31 head 0) in Mistral-7B-Instruct-v0.2.

*qasper* benchmark. The reason behind this analysis is that in order to match the accuracy of Exact-TopK, we need to keep all important KV tokens that are selected by at least one *top-k* attention operation across all decoding steps. As shown in Figure 2, although the maximum sequence length can reach as high as 25000, the number of unique *top-k* indices only goes up to 1200. This implies that, ideally, a permanent KV token eviction method should be able to close the accuracy gap under a token budget of 1200. To further reduce the token budget, we realize that dynamic KV token selection can be applied on the filtered KV token set after permanent KV token eviction. Since the filtered set is much smaller than the original full KV cache, the difficulty for accurate *top-k* prediction is greatly reduced. Therefore, an ideal solution would be to perform permanent KV cache eviction with a larger token budget first and then conduct dynamic KV token selection on the remaining KV tokens. This fusion evicts unimportant tokens and also makes the dynamic selection more accurate, motivating our RocketKV design.

## 3.2. RocketKV Overview

Based on our observation, we propose RocketKV, a two-stage KV cache compression method for decode phase acceleration. As shown in Figure 3, RocketKV performs coarse-grain KV cache eviction in the first stage. The purpose of this stage is to remove KV tokens with low importance while keeping the majority of important tokens. In the second stage, it conducts fine-grain dynamic KV token selection on the remaining KV tokens, followed by *top-k* sparse attention. The RocketKV framework is generic, and many existing KV cache compression methods can fit into the corresponding stage. For example, SnapKV (Li et al., 2024) or Ada-KV (Feng et al., 2024) can be used for the first stage, and Quest (Tang et al., 2024) or SparQ (Ribar et al., 2024) can be applied to the second stage. To achieve the best performance at each stage, we directly adopt SnapKV

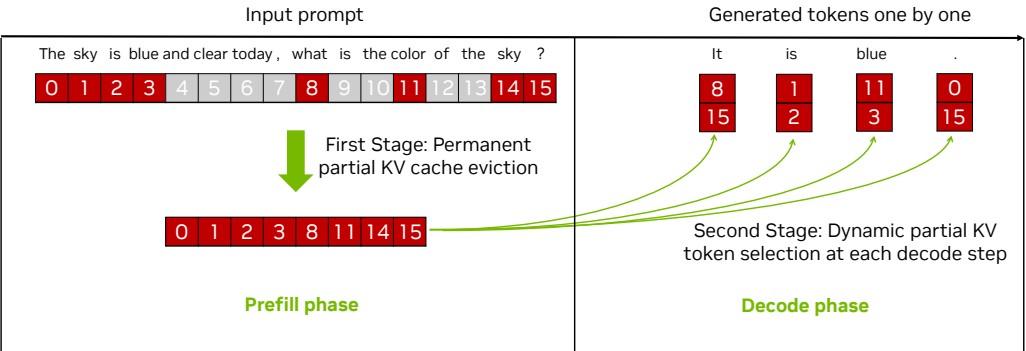

Figure 3: Overview of RocketKV with two consecutive stages.

in the first stage while proposing a hybrid sparse attention (HSA) method in the second stage.

### 3.3. First Stage: SnapKV

In the first stage, we directly adopt SnapKV for permanent KV token eviction on the input KV tokens. SnapKV's key idea is to rely on the aggregated attention scores between the input context and observation window in the end to select the most relevant tokens to keep within the input prompt. The original SnapKV method selects crucial KV tokens on a per attention head basis. In case of grouped-query attention (GQA), each attention head within an attention group keeps a separate set of KV cache tokens, which could introduce redundant storage of the same KV token. To reduce KV token storage with GQA, we follow the Ada-KV work to perform token selection on a per-group basis according to aggregated per-group attention scores so that the selected KV tokens are shared across the entire attention group. SnapKV uses pooling along the sequence dimension to ensure critical KV tokens are selected along with their neighbor tokens. It demonstrates better accuracy with pooling because it retains the completeness of selected information. The employed kernel sizes for pooling are quite small (e.g., a kernel size of 7 for LongBench (Bai et al., 2024)). Since in our case, SnapKV is only used for coarse-grain KV token eviction at the first stage, we discovered that the optimal kernel sizes for pooling are much larger. We empirically set the kernel size to 63 in all our experiments.

### 3.4. Second Stage: Hybrid Sparse Attention

Previous methods on dynamic KV token selection often estimate *top-k* KV indices with reduced computation along a single dimension. For example, Quest (Tang et al., 2024) uses element-wise minimum and maximum values to represent continuous pages along the sequence dimension. Meanwhile, SparQ (Ribar et al., 2024) and Loki (Singhania et al., 2024) leverage sparsity in the head dimension to conduct low-rank estimations. Unfortunately, relying on

one-dimensional sparsity can only achieve a certain degree of compression ratio, beyond which the accuracy could drop rapidly, as shown previously in Figure 1. In contrast, we propose hybrid sparse attention (HSA), which takes advantage of two-dimensional reduction, in both sequence and head dimensions together to achieve better estimation accuracy on KV token indices with *top-k* attention scores.

Figure 4 shows the detailed implementation of our proposed algorithm, which is inspired by Quest (Tang et al., 2024) and SparQ (Ribar et al., 2024). Our HSA algorithm can be decomposed into three steps:

- Step 1: Group tokens in key tensor into consecutive pages along the sequence dimension and store element-wise maximum ($K_{max}$) and minimum ($K_{min}$) values of each page as auxiliary storage similar to Quest. Unlike Quest, they are stored with a different layout by aligning along the head dimension to enable efficient gathering in Step 2. The auxiliary storage is updated accordingly each time a new key token is generated.

- Step 2: For each query $q$, find $k_1$ largest absolute values along the head dimension. Then, fetch only the corresponding indices in either element-wise maximum or minimum tensors, depending on the sign of $q$ at those indices. The goal is to compute element-wise $\max(q \times K_{max}, q \times K_{min})$ for each page to approximate the highest possible attention scores within a page. To further reduce the approximation overhead, we only calculate on $k_1$ partial positions along the head dimension with a large magnitude of $q$ and ignore others, similar to SparQ. Once the approximation attention scores are calculated, $k_2$ indices with the largest attention scores along the sequence dimension are selected.

- Step 3: Perform sparse attention by fetching the original key and value vectors from the predicted $k_2$ indices.

Our HSA algorithm is fully compatible with GQA. To achieve this, we perform all key tensor selections on a per attention group basis. More details are shown in Algorithm 1

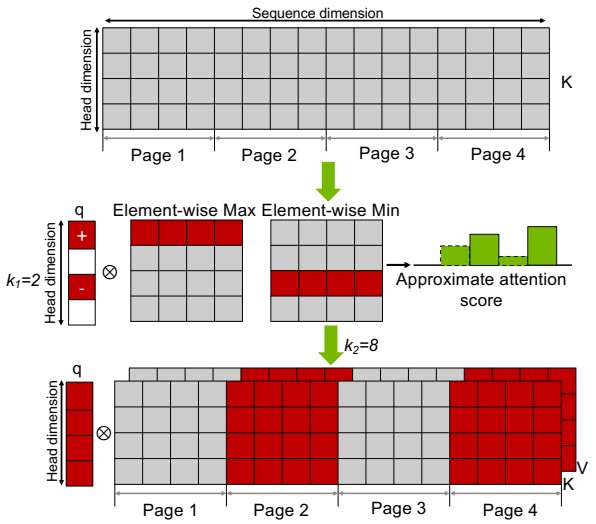

**Algorithm 1** HSA Algorithm (only contains step 2 and 3)

**Input:** query vector $q$, key tensor $K$, value tensor $V$, element-wise max/min key tensor $K_{max}/K_{min}$
*# get top-$k_1$ indices along head dim from sum of $|q|$ in group dim*
$i_1 \leftarrow argtopk(sum(|q|, dim = group), k_1)$
*# get signs of top-$k_1$ indices from sum of $q$ in group dim*
$g \leftarrow sign(sum(q_{[i_1]}, dim = group))$
*# fetch corresponding indices from paged min or max*
$P \leftarrow K_{max[i:g_i \geq 0]}, K_{min[i:g_i < 0]}$
*# compute approximation attention scores*
$s_1 \leftarrow score(q_{[i1]}, P)$
*# get indices with top-$k_2$ attention scores along seq. dim*
$i_2 \leftarrow argtopk(s_1, k_2)$
*# perform sparse attention*
$y \leftarrow attn(q, K_{[i2]}, V_{[i2]})$
**return** $y$

Figure 4: Illustration of hybrid sparse attention with figure (left) and algorithm (right).

where we perform the sum of $q$ or $|q|$ in the group dimension as needed to guarantee that all attention heads within a group are making the same selection at each step.

### 3.5. RocketKV-MT

In a multi-turn conversation setting, KV tokens that were pruned in earlier turns may become essential for answering queries in later turns because KV token importance could vary significantly between different turns (Li et al., 2025). Consequently, permanently removing these tokens could lead to a noticeable accuracy drop in subsequent turns.

To mitigate this issue, we introduce a multi-turn variant of our approach called RocketKV-MT. The core idea in RocketKV-MT is to avoid permanently evicting any KV tokens during the first stage; instead, all KV tokens are retained in memory across turns, ensuring that no potentially useful context is lost. Meanwhile, to preserve computational efficiency, the second stage still performs dynamic selection over the subset of KV tokens filtered by the first stage, similar to the original RocketKV. In other words, the model generates responses using a reduced KV token set for speed, while reserving the full KV history for future turns. For example, suppose the first stage of RocketKV-MT retains only $N$ out of $M$ total KV tokens from the input prompt in the first turn. RocketKV-MT will still keep all $M$ tokens in memory but restrict the second stage to dynamically select from these $N$ input tokens (plus any new generated tokens) during the decode phase. In the next turn, the full set of previously stored KV tokens (all $M$ input tokens plus all output tokens) is added to the new input KV cache. The filtering process (with SnapKV in our case) is applied again

on this whole input KV cache to select a new subset of important tokens for this turn's decode phase. By following this strategy, RocketKV-MT achieves similar decoding speedups to RocketKV in each turn, while retaining the full KV cache history across all turns. This approach effectively eliminates the accuracy degradation caused by permanent KV token eviction in multi-turn scenarios at the cost of no memory storage savings.

### 3.6. Adaptive Compression Decomposition

The decode phase of LLM inference is typically memory bound (Ribar et al., 2024); so, the time spent in the attention module is roughly proportional to the total memory traffic. In this work, we use the token budget $t$ to estimate the amount of memory traffic for each attention operation in the decode phase (we mainly focus on KV cache traffic since it contributes to the majority of memory traffic in this scenario). For example, a token budget of 512 means each attention module needs to fetch an equivalent total amount of 512 key and value pairs from memory. Unlike prior work (Tang et al., 2024) where the token budget only reflects the memory traffic for *top-k* attention (Step 3 in HSA), we define the token budget to also include the memory traffic of *top-k* estimation (Step 2 in HSA). By doing this, the token budget can reflect the overall memory traffic in the attention module more precisely. For simplicity, we evenly split the token budget between these two steps for HSA and all other dynamic KV token selection methods in our later experiments. For models with GQA, this token budget is defined for the entire attention group rather than each attention head. For a given sequence length of $S$, the total

compression ratio of $c$ can be defined as $c = S/t$.

Since RocketKV is a two-stage KV cache compression framework where the filtered KV token set by the first stage serves as the input of the second stage for dynamic selection, it is important to determine the intermediate token budget for the filtered KV token set. For an overall compression ratio of $c$, we define a split factor $r$ so that $c$ is split into $c^r$ for the first stage and $c^{(1-r)}$ for the second stage, where $0 <= r <= 1$. We use the following formula to adaptively determine $r$:

$$r = min(0.2 + 0.06 * log_2(c), 0.8)$$

The insight behind this formula is that when $c$ is small, we would like to minimize the number of permanent evicted KV tokens in the first stage to prevent information loss. As $c$ increases, the accuracy drop caused by HSA in the second stage gets larger thus it is better to assign a higher compression ratio to the first stage because SnapKV can estimate important KV tokens more precisely using exact attention scores. We further limit the range of $r$ between 0.2 and 0.8 to balance the compression decomposition between these two stages.

For compression decomposition within HSA, we simply split the compression ratio $c^{(1-r)}$ evenly between the sequence and head dimensions so that each dimension gets a compression ratio of $c^{(1-r)/2}$. Notice that the compression ratio along the sequence dimension is equivalent to the page size, so we need to round it up the nearest integer $\lceil c^{(1-r)/2} \rceil$ and the other dimension gets $c^{(1-r)}/\lceil c^{(1-r)/2} \rceil$.

Because RocketKV strategically decomposes KV cache compression across multiple stages and dimensions, it significantly enhances the potential for high compression ratios while maintaining strong accuracy—surpassing methods that rely on single-stage, single-dimension approaches. For example, given a compression ratio of $64\times$, the split factor can be calculated as $r = 0.2 + 0.06 * log_2(64) = 0.56$. Therefore, the compression ratio is split into $64^{0.56} = 10.3\times$ in the first stage and $64^{(1-0.56)} = 6.2\times$ in the second stage. HSA further splits its compression ratio into $3\times$ in the sequence dimension (with a page size of 3) and $2.1\times$ in the head dimension. We can see that each individual stage and dimension gets assigned with a much smaller compression ratio after decomposition.

In RocketKV, the first stage results in both KV cache storage and traffic reduction of $c^r$. In the second stage, we need to take the additional memory storage overhead introduced by the approximation attention into consideration. Since we evenly split the compression ratio $c^{(1-r)}$ into $c^{(1-r)/2}$ (ignore the round up operation here for simplicity) between two dimensions in HSA, it introduces a memory storage overhead of $(1/c^r) \times (1/c^{(1-r)/2}) \times 2 = 2/c^{(1+r)/2}$ where $1/c^r$ is the relative KV cache storage after the first stage and

Table 1: Normalized KV cache storage (including auxiliary data) and traffic comparison between RocketKV, RocketKV-MT and other methods.

| Method | Compression Ratio | Storage | Traffic |
|---|---|---|---|
| Full-KV | 1 | 1 | 1 |
| DuoAttention | $c$ | $1/c$ | $1/c$ |
| SnapKV | $c$ | $1/c$ | $1/c$ |
| Quest | $c$ | $1 + 1/c$ | $1/c$ |
| SparQ | $c$ | 2 | $1/c$ |
| RocketKV | $c$ | $1/c^r + 2/c^{(1+r)/2}$ | $1/c$ |
| RocketKV-MT | $c$ | $1 + 2/c^{(1+r)/2}$ | $1/c$ |

both element-wise maximum and minimum tensors introduce a storage overhead of $1/c^{(1-r)/2}$ on top of it. Therefore, the total KV cache storage and traffic in RocketKV are $1/c^r + 2/c^{(1+r)/2}$ and $1/c$ of the full KV baseline, respectively. RocketKV-MT does not lead to storage saving in the first stage, so its relative KV cache storage is $1 + 2/c^{(1+r)/2}$ instead. Table 1 compares the KV cache storage and traffic of RocketKV and RocketKV-MT against other methods at a given compression ratio $c$. We can see that while all methods lead to the same KV cache traffic savings, only RocketKV, DuoAttention and SnapKV provide additional KV cache storage savings, but RocketKV-MT, Quest and SparQ require extra storage for auxiliary data.

### 3.7. System Implications

RocketKV is fully compatible with FlashAttention (Dao et al., 2022) because it does not modify attention in the prefill phase. Additionally, it seamlessly integrates with tensor parallelism (Shoeybi et al., 2019) because all operations are symmetric across attention heads/groups. It is worth noting that both RocketKV and RocketKV-MT work well with the disaggregated serving system, where different GPUs are used for prefill and decode phases (NVIDIA, 2025; Qin et al., 2025). For RocketKV, the KV cache storage is reduced on both the prefill and decode GPU, as well as the KV cache transfer traffic between them. While the full KV cache needs to be stored in the prefill GPU for RocketKV-MT, only the filtered set needs to be transferred and stored in the decode GPU, resulting in the same communication and decoding benefits as RocketKV.

## 4. Experiments

### 4.1. Experimental Settings

We conduct our experiments on three widely used long-context models: Llama3.1-8B-Instruct (AI@Meta, 2024), Mistral-7B-Instruct-v0.2 (Jiang et al., 2023), and LongChat-7B-v1.5-32k (Li et al., 2023). We refer to these models as Llama3.1-8B-Ins, Mistral-7B-Ins-v0.2, LongChat-7B-v1.5, respectively, throughout the paper. The first two models are

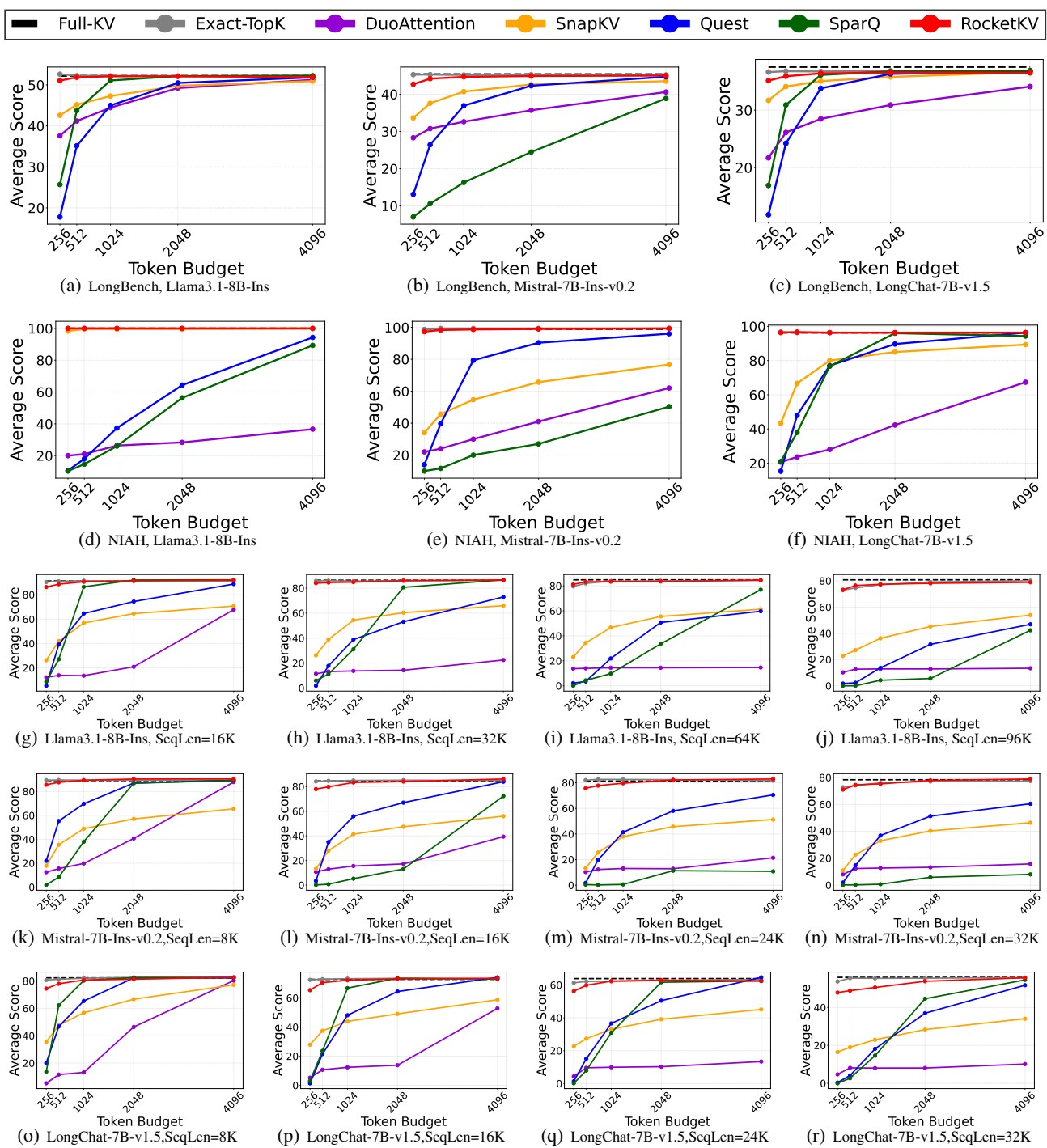

Figure 5: Comparing the accuracy of RocketKV with other methods on LongBench (a-c), Needle-in-a-Haystack (NIAH) (d-f), and RULER with various sequence lengths (g-r).

based on GQA, while the last one is based on MHA. For downstream tasks, we utilize LongBench (Bai et al., 2024), Needle-in-a-Haystack (Kamradt, 2023), RULER (Hsieh et al., 2024), and SCBench (Li et al., 2025). We compare RocketKV and RocketKV-MT with several other methods:

Full-KV, Exact-TopK, DuoAttention (Xiao et al., 2025), SnapKV (Li et al., 2024), Quest (Tang et al., 2024), and SparQ (Ribar et al., 2024). Exact-TopK serves as an oracle method for sparse attention with exact *top-k* KV token selection. Notice that RocketKV-MT acts the same as Rock-

etKV in single-turn scenarios, so we only evaluate it on SCBench under multi-turn mode. We evaluate all methods across various KV token budgets per attention group, except for Full-KV. RocketKV, RocketKV-MT, Quest, and SparQ involve additional memory traffic for *top-k* approximation, which is converted into an equivalent KV token budget such that the total token budget precisely reflects the total amount of memory traffic in the attention module. More details on our experimental settings are available in Appendix A. We also conduct additional ablation studies in Appendix B.1.

## 4.2. Accuracy Results

In our accuracy evaluation, we vary the token budget of each method from 256 to 4096 for all single-turn benchmarks and from 1024 to 16384 for SCBench under multi-turn mode. We compare the average accuracy across all individual tasks for each benchmark. More detailed results with accuracy breakdown for individual tasks are available in Appendix B.

**LongBench Benchmark:** The first row in Figure 5 shows the average score comparison of RocketKV with other methods on LongBench across all three models. Based on the figure, we can see that RocketKV consistently outperforms all other methods, especially under lower token budgets. For Llama3.1-8B-Ins, RocketKV achieves almost no accuracy loss with a token budget of 512 and above, and only 1.1% average accuracy drop with a token budget of 256. RocketKV results in slightly higher accuracy losses for Mistral-7B-Ins-v0.2 and LongChat-7B-v1.5, which might be because Llama3.1-8B-Ins is better-trained than the other two models, making it more robust to sparse attention methods, as a similar trend can be found with Exact-TopK. SparQ performs well on Llama3.1-8B-Ins and LongChat-7B-v1.5 with a token budget of 1024 and above but underperforms all other methods on Mistral-7B-Ins-v0.2. SnapKV alone achieves relatively good accuracy under low token budgets, but we still observe widening accuracy gaps between SnapKV and RocketKV as the token budget decreases.

**Needle-in-a-Haystack (NIAH) Benchmark:** Presented in the second row of Figure 5, RocketKV achieves near the Full-KV accuracy across all models, even with a token budget of 256. In fact, it reaches 100% accuracy on Llama3.1-8B-Ins under 256 token budget, which corresponds to a compression ratio of over 400× with a maximum sequence length of 109K tokens. In contrast, all other methods suffer from substantial accuracy drops. While SnapKV leads to comparable accuracy to RocketKV on Llama3.1-8B-Ins, its accuracy for Mistral-7B-Ins-v0.2 and LongChat-7B-v1.5 decreases by more than 20% compared to Full-KV, even with a token budget of 1024. We present the heatmaps of RocketKV under various token budgets in Appendix B.2.

**RULER Benchmark:** For the RULER benchmark, we evaluate all methods across three models with varying sequence

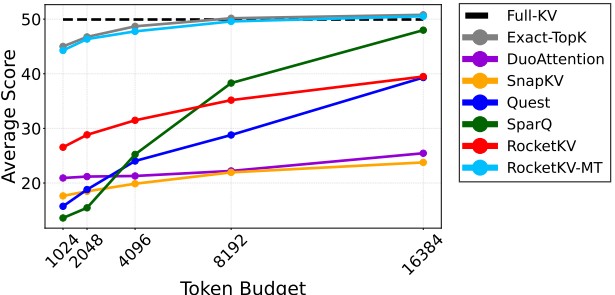

Figure 6: Comparing the accuracy of RocketKV and RocketKV-MT with other methods on Llama3.1-8B-Ins for SCBench.

lengths as shown in Figure 5 (rows 3-5). Again, RocketKV shows robust performance and a clear advantage over all other methods across various token budget and sequence length settings. Overall, we can see that the accuracy loss of RocketKV is negligible under short sequence lengths and gradually becomes larger as sequence length increases. We believe this is because these models are less robust to sparse attention beyond their effective sequence lengths as defined in the RULER paper (Hsieh et al., 2024). Notice that the accuracy gaps between RocketKV and other methods become even wider under longer sequence lengths.

**SCBench Benchmark:** Figure 6 presents the accuracy comparison of various methods on SCBench under the multi-turn setting. While RocketKV still outperforms other methods under low token budgets, it provides lower accuracy than SparQ for token budgets of 8192 and beyond. Moreover, there is still a noticeable accuracy gap between RocketKV and Exact-TopK under all token budgets, showing room for further improvement. As we discussed earlier, this is primarily caused by the fact that KV token importance varies significantly across different turns, so the unimportant KV tokens evicted by earlier turns could lead to a significant accuracy drop in later turns. After fixing this issue in RocketKV-MT, we can see that RocketKV-MT achieves a significant accuracy boost with comparable accuracy to Exact-TopK across all token budgets.

## 4.3. Efficiency Results

Our efficiency experiments are conducted with Llama3.1-8B-Ins (AI@Meta, 2024) under FP16 precision, running on NVIDIA H100 and A100 GPUs at a batch size of 1. Similar to SparQ, we leverage gpt-fast (PyTorchLabs, 2023), a low-latency python-native LLM framework for running the efficiency experiments. We found a python-based implementation of RocketKV under gpt-fast is sufficient to demonstrate its efficiency benefit, but it could be further improved with customized CUDA kernels and more advanced frameworks such as FlashInfer (Ye et al., 2025). Figure 7

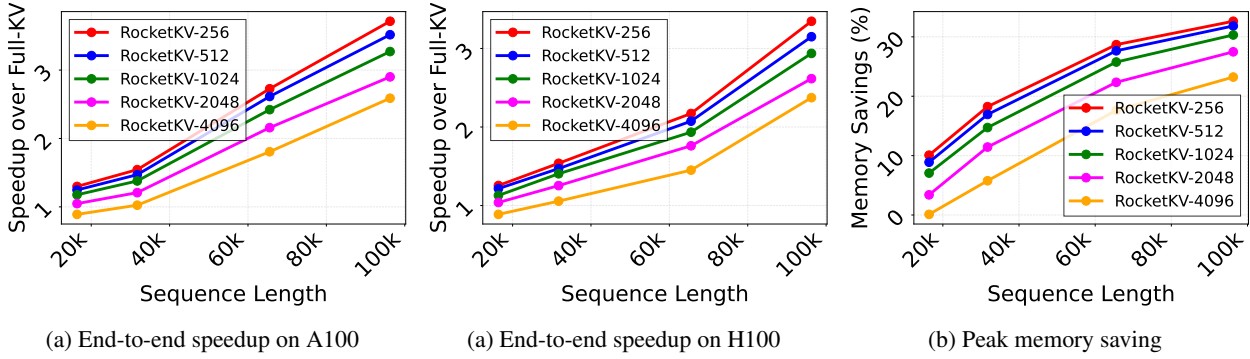

(a) End-to-end speedup on A100      (a) End-to-end speedup on H100      (b) Peak memory saving

Figure 7: End-to-end speedup and peak memory savings of RocketKV with various token budgets compared to Full-KV.

demonstrates the end-to-end speedups and peak memory savings of RocketKV with varying token budgets at the decode phase, with all values normalized to Full-KV. Notice that the peak memory usage is the same regardless of the underlying GPU, and it is measured includes memory allocations for weights, activations, KV cache, and all other metadata at the decode phase.

As shown in the Figure 7, RocketKV achieves up to 3.7× and 3.3× end-to-end speedups on an A100 and H100 GPU, respectively, as well as up to 32.6% peak memory saving. The maximum speedup on A100 is 12% higher than H100 because A100 has a lower memory bandwidth to compute ratio compared to H100. As a result, LLM inference execution is more memory-bound on A100 and can benefit more from the memory traffic savings of the KV cache offered by RocketKV. We expect the speedup of RocketKV will be even higher on cheaper GPUs such as RTX 4090/5090 since they are not equipped with High Bandwidth Memory (HBM).

## 5. Conclusion

RocketKV presents a novel, training-free approach to KV cache compression, addressing the challenges of memory bandwidth and capacity demands during the decode phase of LLM inference. RocketKV contains two consecutive stages: SnapKV for coarse-grain permanent KV cache eviction and hybrid sparse attention (HSA) for fine-grain dynamic KV token selection. Our evaluations on various models and long-context benchmarks demonstrate that RocketKV maintains comparable accuracy to full KV cache attention while significantly lowering memory bandwidth and capacity usage with a compression ratio of up to 400×, as well as up to 3.7× end to end speedup and 32.6% peak memory reduction at the decode phase, highlighting its efficiency and potential for widespread application in optimizing LLM performance. We also propose a variant of RocketKV called RocketKV-MT that is optimized for multi-turn scenarios.

## Acknowledgment

We thank the anonymous ICML reviewers for their valuable and constructive feedback. We are grateful to David Nellans for overseeing the internship process of Payman Behnam, during which this work was supported by the NVIDIA Graduate Fellowship. We also thank Song Han and Christos Kozyrakis for their insightful discussions and helpful feedback.

## Impact Statement

This paper presents work whose goal is to advance the field of machine learning. There are many potential societal consequences of our work, none of which we feel must be specifically highlighted here.

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

# A. Detailed Experiment Settings

## A.1. Models

We evaluate our methods on the three widely used long-context models: Llama3.1-8B-Ins (AI@Meta, 2024), Mistral-7B-Ins-v0.2 (Jiang et al., 2023), and LongChat-7B-v1.5 (Bai et al., 2024). They are all decoder-only Transformer models (Vaswani et al., 2017). Llama3.1-8B-Ins and Mistral-7B-Ins-v0.2 use GQA, while LongChat-7B-v1.5 uses MHA in the attention module. In terms of sequence length, Llama3.1-8B-Ins supports a maximum sequence length of 128K while the other two support up to 32K sequence length. Prior work (Tang et al., 2024; Chen et al., 2025) usually skips the first two layers during KV cache compression to maintain accuracy. We found that this is only necessary for LongChat-7B-v1.5; so, we conduct KV cache compression on all attention layers for Llama3.1-8B-Ins and Mistral-7B-Ins-v0.2 but skip the first two layers for LongChat-7B-v1.5.

## A.2. Benchmarks

Our LongBench and Needle-in-a-Haystack settings mostly follow the prior work (Yuan et al., 2024). For LongBench, we set the maximum prompt length to 127,500 for Llama3.1-8B-Ins and 31,500 for Mistral-7B-Ins-v0.2 and LongChat-7B-v1.5. Prompts beyond the maximum prompt length will be middle-truncated by keeping the first and last half of the input tokens. For Needle-in-a-Haystack, we evaluate with 10 different input sequence lengths uniformly spanning from 2048 to 81,920 words and 10 different depths for each sequence length in Llama3.1-8B-Ins. The maximum word count of 81,920 roughly converts to 109K tokens by the tokenizer in Llama3.1-8B-Ins. For the other two models, the input sequence lengths range from 512 to 20480 words since they support up to 32K sequence length, which convert up to about 30K and 31K tokens for Mistral-7B-Ins-v0.2 and LongChat-7B-v1.5, respectively. For RULER, we mostly follow the configurations in the original benchmark, except for reducing the number of examples per task from 500 to 50 to speed up the evaluation. For SCBench, we follow the multi-turn setting and set the maximum prompt length to 127,500 for Llama3.1-8B-Ins.

## A.3. Baselines

Since RocketKV primarily focuses on accelerating the decode phase of LLM inference, we do not perform any KV cache compression or sparse attention mechanism in the prefill phase across all comparing baselines, including DuoAttention, for a fair comparison.

**Exact-TopK:** Exact-TopK serves as an oracle method to demonstrate the effectiveness of *top-k* sparse attention. We assume the *top-k* KV tokens can be directly identified with no cost so a given token budget will directly correspond to its *top-k* value.

**DuoAttention:** We set the initial and recent tokens to 128 and 512, respectively, for streaming heads if the token budget is larger than 640, and vary the ratio of retrieval heads to match the given token budget on average. If the token budget is lower than 640, all attention heads become streaming heads. In this case, we set the number of initial tokens for the attention sink to 128 if the token budget is larger than 256, otherwise, it is set to 20% of the token budget. The rest of the token budget is assigned to recent tokens within the sliding window.

**SnapKV:** We set the observation window size to 32 for all single-turn benchmarks and 128 for multi-turn SCBench. The kernel size for pooling is set to 7, following the SnapKV paper. The token budget is divided by the number of heads per attention group in case of GQA as the SnapKV performs on each attention head separately and the pruned KV cache is not shared within the attention group.

**Quest:** We evenly split the token budget between approximate attention for identifying *top-k* KV token indices and *top-k* sparse attention so that the token budget can accurately reflect the total memory fetching bandwidth required in the attention module. The original Quest method is not compatible with GQA. We modify it to select indices of *top-k* accumulated attention scores per attention group rather than per attention head.

**SparQ:** Similar to Quest, we evenly split the token budget between approximate attention for identifying *top-k* KV token indices and *top-k* sparse attention.

**RocketKV/RocketKV-MT:** For SnapKV in the first stage, We set the observation window size to 32 for all single-turn benchmarks and 128 for multi-turn SCBench. For HSA in the second stage, we evenly split the token budget between approximate attention for identifying *top-k* indices and *top-k* sparse attention.

# B. Additional Results

## B.1. Ablation Studies

In this subsection, we present a series of ablation studies to further demonstrate the effectiveness of RocketKV.

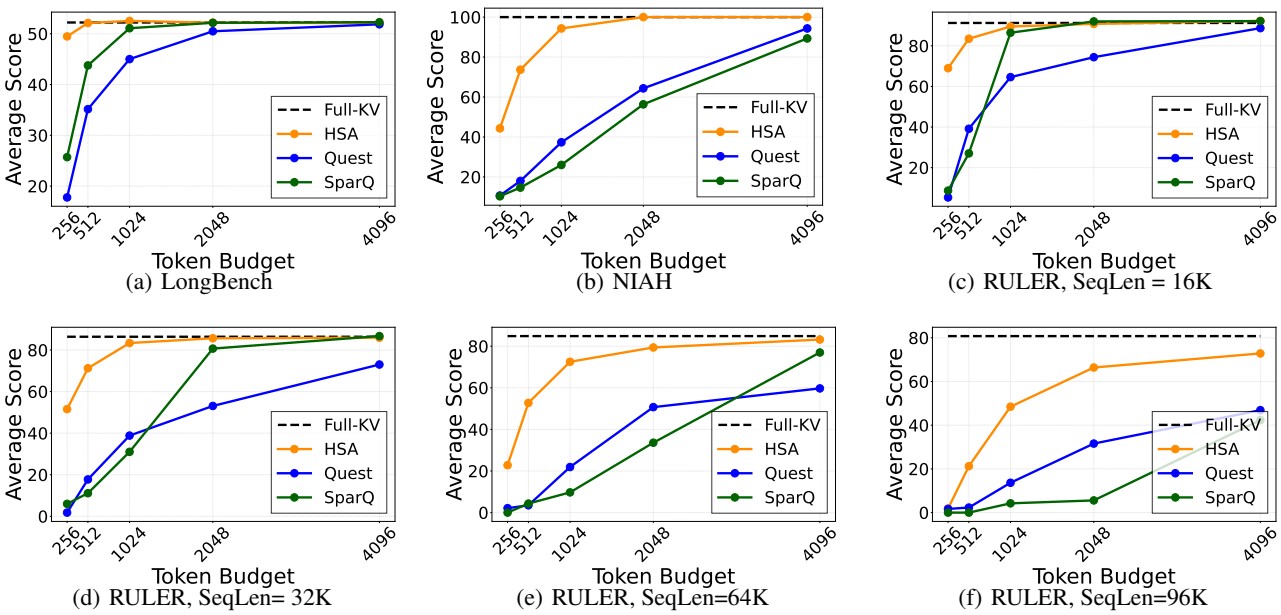

Figure 8: Accuracy comparison among HSA, Quest, and SparQ on Llama3.1-8B-Ins.

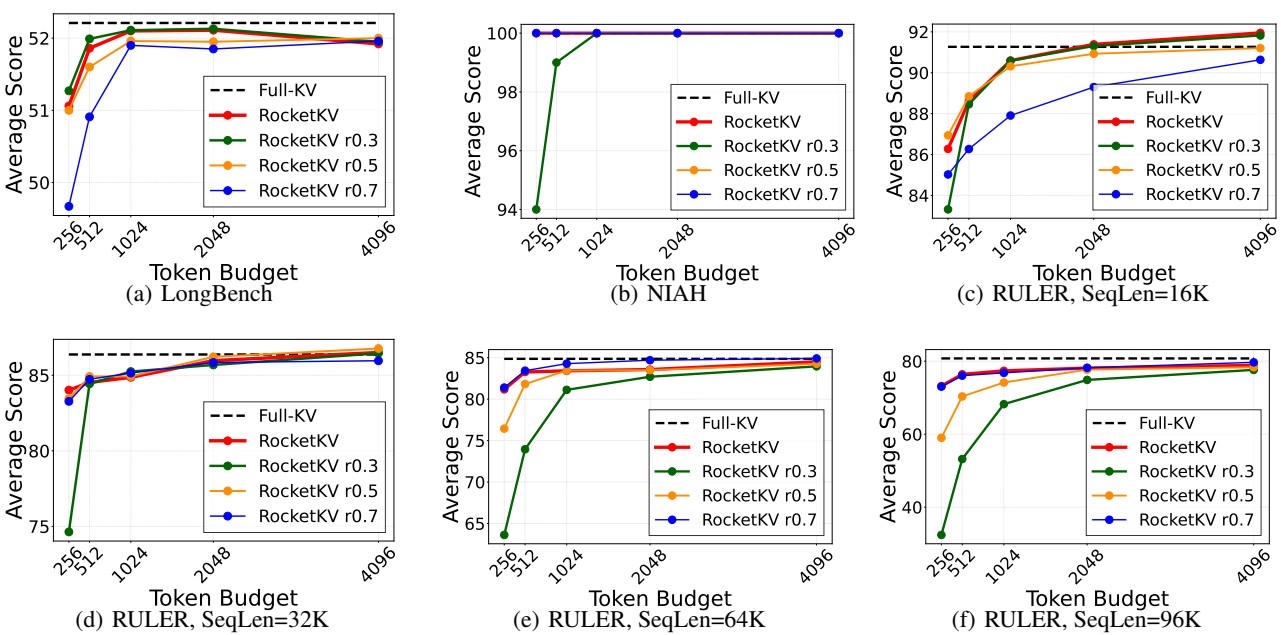

Figure 9: Comparing adaptive against static split factors on Llama3.1-8B-Ins.

### B.1.1. COMPARING HSA, QUEST, AND SPARQ

To illustrate the effectiveness of hybrid sparse attention (HSA), we compare the accuracy of the standalone HSA mechanism against Quest and SparQ. Figure 8 demonstrates the results on Llama3.1-8B-Ins across multiple different benchmarks. In all cases, HSA consistently outperforms Quest and SparQ, especially at low token budgets. This clearly demonstrates the advantage of HSA, which intelligently leverages approximations in both sequence and head dimensions compared to single dimension approximation methods such as Quest and SparQ.

### B.1.2. SPLIT FACTOR

In this study, we compare the adaptive compression decomposition method against statically determined split factors $r$ ranging from 0.3 to 0.7. A split factor of 0.5 indicates an even split between the first and second stage of RocketKV. As shown in Figure 9, the best static split factor varies with different sequence lengths and token budgets, while adaptive compression decomposition provides comparable accuracy to the best static split factor in most cases.

### B.2. Needle-in-a-Haystack Visualization

Needle-in-a-haystack (NIAH) is a type of synthetic challenge designed to test how effectively an LLM can retrieve specific information in a large volume of text (Kamradt, 2023). In Figures 10, 11, 12, the x-axis shows the document length (i.e., "haystack"), while the y-axis marks the relative position of the "needle" (i.e., a short sentence) within the text. As shown in the results, RocketKV can retrieve the needle with almost the same accuracy as Full-KV throughout the whole text across all three models with token budgets as low as 256.

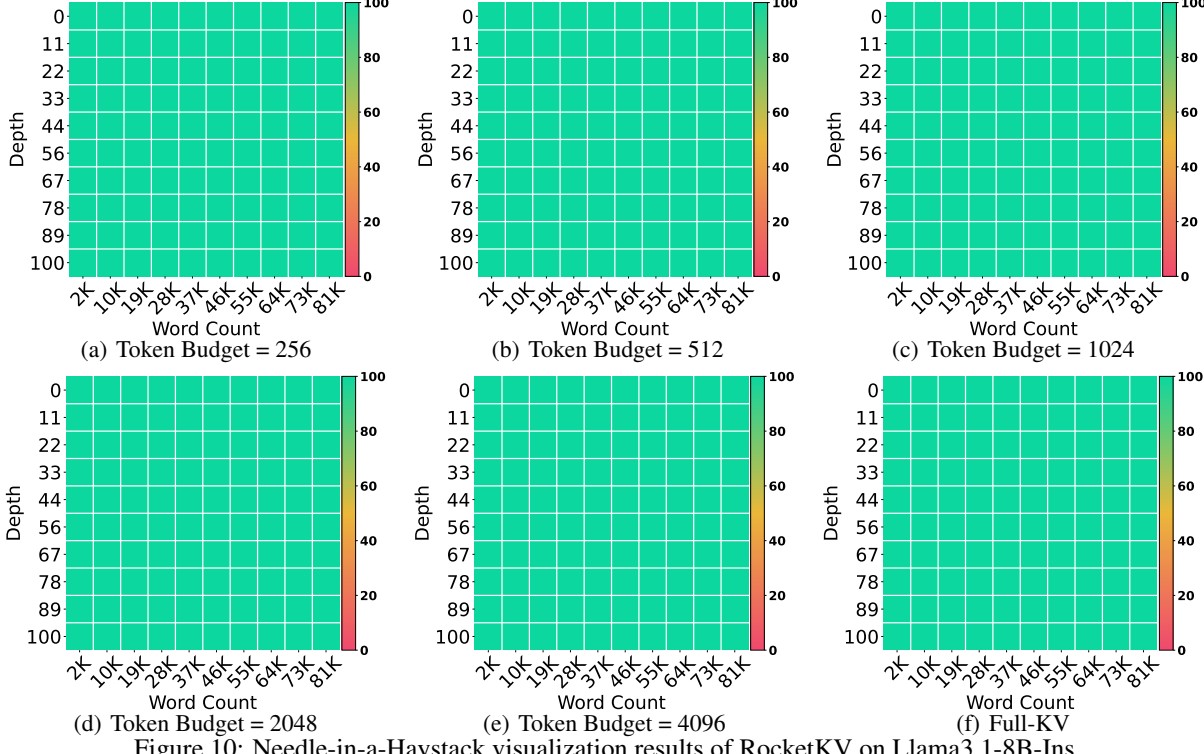

Figure 10: Needle-in-a-Haystack visualization results of RocketKV on Llama3.1-8B-Ins.

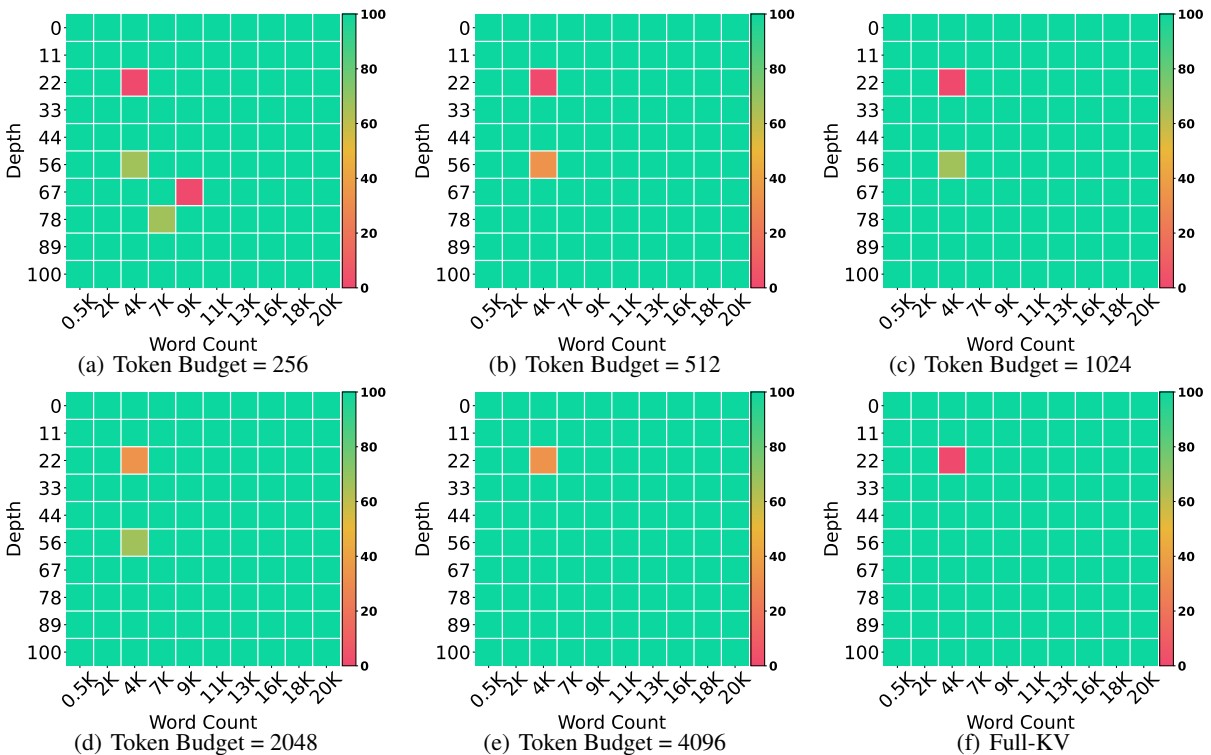

Figure 11: Needle-in-a-Haystack visualization results of RocketKV on Mistral-7B-Ins-v0.2.

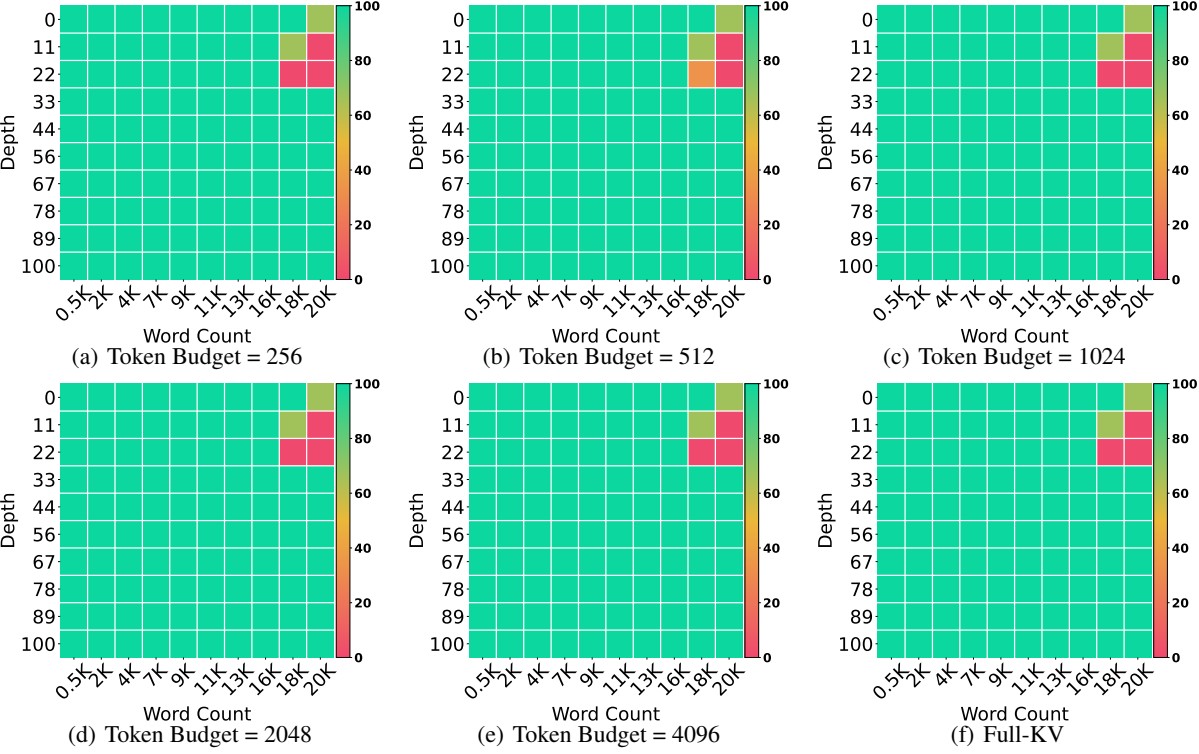

Figure 12: Needle-in-a-Haystack visualization results of RocketKV on LongChat-7B-v1.5.

## B.3. Detailed Accuracy Results

In this section, we present 20 detailed tables that display the accuracy results discussed in previous sections. These tables offer additional information and insights into the results we captured.

Table 2: Average results of LongBench and NIAH for Llama3.1-8B-Ins

| Token Budget | Method | Single QA | Multi. QA | Summ. | Few-shot | Synthetic | Code | LB Avg. | NIAH |
|---|---|---|---|---|---|---|---|---|---|
| N/A | Full-KV | 43.5 | 44.5 | 29.2 | 69.3 | 53.9 | 62.0 | 52.2 | 100.0 |
| 256 | Exact-TopK | 43.6 | 44.4 | 28.9 | 69.5 | 53.6 | 65.3 | 52.6 | 100.0 |
| | DuoAttention | 24.4 | 33.5 | 18.2 | 39.0 | 51.2 | 60.1 | 37.6 | 20.0 |
| | SnapKV | 32.5 | 39.1 | 20.3 | 50.7 | 53.0 | 56.8 | 42.6 | 98.3 |
| | Quest | 16.0 | 14.0 | 14.3 | 14.4 | 4.0 | 41.4 | 17.8 | 10.7 |
| | SparQ | 26.3 | 18.7 | 18.2 | 24.5 | 15.5 | 47.0 | 25.7 | 10.3 |
| | HSA | 41.3 | 41.9 | **26.8** | 64.2 | 51.7 | 61.1 | 49.5 | 44.3 |
| | RocketKV | **43.7** | **43.5** | 25.2 | **67.3** | **53.8** | **64.0** | **51.1** | **100.0** |
| 512 | Exact-TopK | 43.7 | 44.5 | 28.8 | 69.1 | 53.7 | 63.1 | 52.2 | 100.0 |
| | DuoAttention | 27.7 | 34.9 | 21.6 | 47.4 | 51.5 | 61.7 | 41.2 | 21.0 |
| | SnapKV | 36.0 | 41.9 | 21.9 | 54.8 | **53.4** | 57.9 | 45.2 | 99.7 |
| | Quest | 27.9 | 27.7 | 19.8 | 38.0 | 43.5 | 52.6 | 35.1 | 18.0 |
| | SparQ | 37.1 | 35.1 | 24.5 | 53.2 | 48.0 | 57.3 | 43.8 | 14.7 |
| | HSA | **43.8** | 43.2 | **28.5** | 68.6 | 53.1 | **65.2** | **52.1** | 73.7 |
| | RocketKV | 43.7 | **43.8** | 26.8 | **69.5** | 53.4 | 63.5 | 51.9 | **100.0** |
| 1024 | Exact-TopK | 43.8 | 44.6 | 29.0 | 69.3 | 53.5 | 61.9 | 52.2 | 100.0 |
| | DuoAttention | 31.8 | 36.4 | 23.7 | 54.8 | 53.3 | 64.2 | 44.5 | 26.3 |
| | SnapKV | 38.2 | 42.7 | 23.5 | 59.0 | 53.7 | 59.6 | 47.3 | 99.7 |
| | Quest | 36.5 | 37.5 | 24.8 | 54.9 | 50.9 | 57.3 | 45.0 | 37.3 |
| | SparQ | 41.5 | 42.9 | 28.6 | 67.8 | 52.0 | 62.2 | 51.1 | 26.0 |
| | HSA | **43.9** | **44.0** | **28.8** | **69.7** | **54.0** | **64.6** | **52.5** | 94.3 |
| | RocketKV | **43.9** | 43.8 | 28.3 | 69.2 | 53.4 | 63.2 | 52.1 | **100.0** |
| 2048 | Exact-TopK | 43.6 | 44.5 | 29.2 | 69.5 | 53.8 | 61.1 | 52.1 | 100.0 |
| | DuoAttention | 39.3 | 40.9 | 25.4 | 64.5 | 52.2 | **64.8** | 49.2 | 28.3 |
| | SnapKV | 40.6 | 43.4 | 25.5 | 65.0 | 52.8 | 61.7 | 49.8 | 99.7 |
| | Quest | 42.5 | 43.0 | 28.4 | 65.5 | 52.3 | 59.8 | 50.5 | 64.3 |
| | SparQ | 43.5 | 43.5 | **29.3** | 69.4 | 53.6 | 62.9 | 52.1 | 56.3 |
| | HSA | **43.7** | 43.8 | 29.2 | 69.4 | 53.6 | 62.3 | **52.2** | **100.0** |
| | RocketKV | 43.6 | **44.0** | 28.9 | **69.5** | **53.9** | 62.2 | 52.1 | **100.0** |
| 4096 | Exact-TopK | 43.5 | 44.8 | 29.1 | 69.3 | 53.3 | 60.4 | 52.0 | 100.0 |
| | DuoAttention | 42.4 | 41.7 | 27.5 | 68.9 | 53.2 | **63.9** | 51.2 | 36.7 |
| | SnapKV | 42.3 | 43.7 | 26.9 | 67.5 | 53.4 | 61.5 | 50.9 | **100.0** |
| | Quest | **43.9** | **44.3** | 28.9 | 69.0 | 53.5 | 60.0 | 51.9 | 94.3 |
| | SparQ | 43.7 | 44.0 | 29.1 | 69.2 | 53.1 | 63.2 | **52.3** | 89.3 |
| | HSA | **43.9** | 44.1 | **29.1** | 69.4 | 53.5 | 61.1 | 52.1 | **100.0** |
| | RocketKV | 43.3 | 43.9 | 29.0 | **69.6** | **53.6** | 61.1 | 51.9 | **100.0** |

Table 3: Individual results of LongBench for Llama3.1-8B-Ins

| Token Budget | Method | narrativeqa | qasper | multifieldqa | hotpotqa | 2wikimqa | musique | gov-report | qmsum | multi-news | trec | triviaqa | samsum | pass-ret. | lcc | repobench-p | passagecnt |
|---|---|---|---|---|---|---|---|---|---|---|---|---|---|---|---|---|---|
| N/A | Full-KV | 30.2 | 45.5 | 54.9 | 55.5 | 46.7 | 31.3 | 35.2 | 25.2 | 27.2 | 72.5 | 91.7 | 43.8 | 99.5 | 65.1 | 58.8 | 8.4 |
| 256 | Exact-TopK | 30.7 | 44.7 | 55.4 | 55.0 | 46.5 | 31.7 | 34.8 | 25.1 | 26.8 | 71.5 | 92.2 | 44.8 | 99.5 | 67.1 | 63.6 | 7.8 |
| | DuoAttention | 19.4 | 22.8 | 31.1 | 42.3 | 35.9 | 22.4 | 16.9 | 20.6 | 17.2 | 20.0 | 57.4 | 39.7 | 98.0 | 61.2 | 59.1 | 4.5 |
| | SnapKV | 23.7 | 27.3 | 46.3 | 52.3 | 40.8 | 24.2 | 19.2 | 22.6 | 19.2 | 29.0 | 84.1 | 39.0 | 97.5 | 57.3 | 56.2 | 8.6 |
| | Quest | 3.0 | 22.4 | 22.6 | 14.9 | 23.2 | 4.1 | 9.8 | 12.7 | 20.5 | 11.5 | 17.3 | 14.4 | 7.5 | 49.5 | 33.3 | 0.5 |
| | SparQ | 5.0 | 38.0 | 35.8 | 19.7 | 28.8 | 7.5 | 13.7 | 15.0 | 25.8 | 16.5 | 32.1 | 24.8 | 28.8 | 60.1 | 34.0 | 2.2 |
| | HSA | 26.4 | 43.5 | 54.1 | 54.2 | 43.7 | 27.8 | **30.7** | 23.2 | **26.6** | 63.5 | 87.0 | 42.2 | 97.0 | **66.3** | 56.0 | 6.5 |
| | RocketKV | **31.0** | 43.9 | **56.2** | **55.2** | 45.2 | 30.0 | 26.6 | **24.7** | 24.2 | 67.5 | 91.0 | 43.3 | 99.0 | 65.4 | 62.7 | **8.7** |
| 512 | Exact-TopK | 30.9 | 45.0 | 55.1 | 55.3 | 47.0 | 31.0 | 34.5 | 25.0 | 26.9 | 71.5 | 92.0 | 43.9 | 99.5 | 65.0 | 61.1 | 7.9 |
| | DuoAttention | 23.0 | 25.0 | 35.1 | 44.8 | 36.4 | 23.6 | 22.3 | 20.2 | 22.4 | 34.5 | 65.9 | 41.7 | **99.5** | 63.0 | 60.4 | 3.5 |
| | SnapKV | 25.5 | 32.3 | 50.2 | 54.1 | 44.2 | 27.4 | 21.5 | 23.0 | 21.2 | 34.5 | 89.4 | 40.5 | 98.0 | 59.7 | 56.1 | **8.7** |
| | Quest | 9.0 | 36.5 | 38.2 | 32.2 | 36.4 | 14.4 | 16.1 | 17.7 | 25.6 | 35.2 | 51.2 | 27.5 | 82.0 | 59.7 | 45.4 | 5.0 |
| | SparQ | 13.6 | **45.4** | 52.1 | 43.8 | 39.4 | 22.2 | 25.6 | 21.2 | 26.7 | 49.5 | 72.2 | 37.9 | 92.0 | 64.2 | 50.5 | 4.1 |
| | HSA | 30.4 | 45.2 | **55.8** | 53.8 | 46.1 | 29.8 | **33.8** | 24.9 | **26.9** | 70.5 | 90.1 | **45.2** | 99.0 | **66.3** | **64.2** | 7.1 |
| | RocketKV | 30.6 | 44.8 | 55.7 | 54.7 | 46.5 | 30.2 | 30.2 | 24.7 | 25.5 | **73.0** | **91.8** | 43.6 | **99.5** | 64.8 | 62.3 | 7.3 |
| 1024 | Exact-TopK | 30.8 | 45.5 | 55.0 | 54.7 | 48.0 | 31.1 | 34.7 | 25.1 | 27.1 | 72.5 | 91.5 | 43.8 | 99.5 | 64.2 | 59.6 | 7.5 |
| | DuoAttention | 24.1 | 30.4 | 41.1 | 45.3 | 39.4 | 24.4 | 24.5 | 21.2 | 25.3 | 45.5 | 76.3 | 42.5 | 98.5 | **66.8** | 61.5 | 8.0 |
| | SnapKV | 28.1 | 36.6 | 50.0 | **56.0** | 44.2 | 28.0 | 23.8 | 23.8 | 23.1 | 44.5 | 91.3 | 41.3 | **99.5** | 61.8 | 57.4 | 8.0 |
| | Quest | 15.4 | 44.4 | 49.8 | 48.9 | 40.8 | 22.9 | 27.1 | 21.0 | 26.4 | 57.0 | 70.0 | 37.9 | 99.0 | 62.6 | 52.0 | 2.7 |
| | SparQ | 23.1 | **46.1** | 55.1 | 54.7 | 42.9 | **31.2** | 34.0 | 24.9 | **27.0** | 69.5 | 89.9 | 44.0 | **99.5** | 65.0 | 59.3 | 4.5 |
| | HSA | **31.1** | 45.3 | 55.3 | 54.2 | 46.7 | **31.2** | 34.6 | 24.9 | 26.9 | 72.5 | **92.1** | **44.5** | **99.5** | **66.3** | **62.9** | **8.5** |
| | RocketKV | 30.7 | 44.9 | **56.0** | 54.6 | 46.7 | 30.0 | 33.0 | **25.2** | 26.6 | 72.0 | **92.1** | 43.5 | **99.5** | 64.7 | 61.8 | 7.3 |
| 2048 | Exact-TopK | 30.7 | 45.3 | 54.7 | 54.8 | 47.2 | 31.3 | 34.9 | 25.4 | 27.2 | 73.0 | 91.5 | 44.1 | 99.5 | 63.8 | 58.4 | 8.2 |
| | DuoAttention | 25.6 | 42.3 | 50.0 | 52.3 | 42.1 | 28.1 | 27.5 | 22.1 | 26.6 | 66.0 | 84.9 | 42.8 | 98.5 | **66.1** | **63.6** | 6.0 |
| | SnapKV | 28.8 | 39.9 | 53.1 | 53.9 | 46.5 | 29.9 | 26.0 | 24.7 | 26.0 | 60.0 | **92.4** | 42.6 | **99.5** | 63.8 | 59.6 | 6.0 |
| | Quest | 28.0 | **45.5** | 54.2 | 54.3 | 44.7 | 29.8 | 33.1 | 24.9 | 27.2 | 68.5 | 84.5 | 43.4 | **99.5** | 63.2 | 56.5 | 5.1 |
| | SparQ | 30.3 | 45.2 | 54.9 | 53.7 | 46.5 | 30.4 | **35.3** | 25.4 | 27.1 | 73.0 | 91.2 | 44.0 | **99.5** | 65.3 | 60.5 | 7.7 |
| | HSA | **30.7** | 45.2 | **55.2** | 54.1 | 46.2 | **31.1** | 35.1 | 25.1 | **27.4** | 73.0 | 91.2 | **44.2** | **99.5** | 64.3 | 60.3 | 7.7 |
| | RocketKV | 30.6 | **45.5** | 54.8 | 54.5 | 47.1 | 30.3 | 33.9 | **25.5** | 27.2 | **73.5** | 91.5 | 43.4 | **99.5** | 64.4 | 59.9 | **8.2** |
| 4096 | Exact-TopK | 30.2 | 45.5 | 54.8 | 55.5 | 47.1 | 31.8 | 35.2 | 25.1 | 27.2 | 72.5 | 91.7 | 43.7 | 99.5 | 63.4 | 57.5 | 7.2 |
| | DuoAttention | 26.5 | 45.1 | **55.5** | 55.0 | 43.3 | 26.7 | 31.9 | 23.7 | 27.0 | **73.5** | 89.9 | 43.3 | **99.5** | 65.5 | **62.3** | 7.0 |
| | SnapKV | 29.9 | 43.8 | 53.2 | **55.7** | 45.7 | 29.9 | 29.0 | 24.9 | 26.9 | 67.5 | **91.7** | 43.3 | **99.5** | 63.9 | 59.0 | 7.3 |
| | Quest | 30.7 | 45.5 | 55.4 | 55.6 | 46.0 | 31.3 | **34.9** | 24.8 | 27.1 | 72.0 | 90.9 | 44.0 | **99.5** | 63.3 | 56.8 | 7.4 |
| | SparQ | 30.8 | 45.3 | 54.9 | 54.7 | **46.4** | 31.0 | **34.9** | 25.2 | **27.2** | 72.5 | 91.5 | 43.7 | **99.5** | **65.6** | 60.7 | 6.6 |
| | HSA | **31.2** | **45.7** | 54.7 | 54.3 | **46.4** | **31.6** | **34.9** | **25.3** | **27.2** | 72.5 | 91.5 | 44.2 | **99.5** | 63.7 | 58.6 | 7.5 |
| | RocketKV | 30.3 | 45.3 | 54.3 | 54.8 | 46.3 | 30.4 | 34.7 | 25.0 | 27.1 | 73.0 | 91.5 | **44.3** | **99.5** | 63.7 | 58.5 | **7.6** |

Table 4: Average SCBench results for Llama3.1-8B-Ins

| Token Budget | Method | Retr.String | Retr.Semantic | Global | Multi-task | Avg. |
|---|---|---|---|---|---|---|
| N/A | Full-KV | 49.5 | 40.7 | 36.2 | 64.6 | 49.9 |
| 1024 | Exact-TopK | 32.3 | 41.0 | 36.8 | 61.8 | 45.0 |
| | DuoAttention | 0.1 | 26.6 | 33.5 | 11.6 | 20.9 |
| | SnapKV | 0.2 | 17.0 | 34.7 | 10.7 | 17.6 |
| | Quest | 0.0 | 22.4 | 25.2 | 4.0 | 15.7 |
| | SparQ | 0.0 | 19.2 | 23.6 | 0.9 | 13.6 |
| | HSA | 4.1 | 27.2 | 27.2 | 20.0 | 22.1 |
| | RocketKV | 14.3 | 22.7 | 35.3 | 26.2 | 26.6 |
| | RocketKV-MT | **40.5** | **35.7** | **36.8** | **56.2** | **44.3** |
| 2048 | Exact-TopK | 38.0 | 41.2 | 36.6 | 62.9 | 46.7 |
| | DuoAttention | 0.1 | 26.6 | 33.9 | 12.2 | 21.2 |
| | SnapKV | 1.1 | 17.1 | 35.9 | 12.1 | 18.5 |
| | Quest | 2.0 | 24.4 | 26.6 | 11.8 | 18.8 |
| | SparQ | 0.1 | 21.1 | 25.8 | 4.1 | 15.5 |
| | HSA | 8.4 | 35.3 | 35.7 | 53.4 | 34.6 |
| | RocketKV | 16.5 | 25.0 | 35.4 | 30.5 | 28.8 |
| | RocketKV-MT | **42.2** | **38.4** | **37.0** | **59.5** | **46.4** |
| 4096 | Exact-TopK | 43.4 | 42.1 | 36.4 | 63.8 | 48.7 |
| | DuoAttention | 0.1 | 26.7 | 33.9 | 12.7 | 21.3 |
| | SnapKV | 3.5 | 19.1 | 35.1 | 13.2 | 19.9 |
| | Quest | 8.6 | 26.8 | 28.5 | 23.0 | 24.0 |
| | SparQ | 3.0 | 29.9 | 29.2 | 30.8 | 25.2 |
| | HSA | 13.1 | 38.7 | 36.3 | **60.9** | 38.6 |
| | RocketKV | 18.9 | 28.5 | 35.9 | 33.9 | 31.5 |
| | RocketKV-MT | **44.8** | **39.9** | **36.8** | 60.3 | **47.8** |
| 8192 | Exact-TopK | 48.0 | 42.4 | 36.4 | 64.7 | 50.2 |
| | DuoAttention | 0.1 | 27.3 | 34.0 | 16.5 | 22.2 |
| | SnapKV | 5.0 | 20.3 | 35.7 | 19.0 | 21.9 |
| | Quest | 10.1 | 31.6 | 32.0 | 31.9 | 28.8 |
| | SparQ | 6.8 | 41.5 | 35.6 | **64.1** | 38.3 |
| | HSA | 34.1 | 41.8 | **36.5** | 63.8 | 46.0 |
| | RocketKV | 22.9 | 33.2 | 35.8 | 39.1 | 35.2 |
| | RocketKV-MT | **47.7** | **42.0** | 36.2 | 62.9 | **49.6** |
| 16384 | Exact-TopK | 49.5 | 42.8 | 36.4 | 65.0 | 50.8 |
| | DuoAttention | 0.4 | 28.9 | 34.0 | 30.5 | 25.4 |
| | SnapKV | 6.9 | 22.0 | 34.5 | 24.7 | 23.8 |
| | Quest | 12.5 | 40.6 | 36.2 | 61.8 | 39.3 |
| | SparQ | 40.3 | 42.4 | 36.1 | **64.5** | 48.0 |
| | HSA | 46.5 | **43.1** | **36.6** | 64.4 | 50.0 |
| | RocketKV | 29.4 | 37.2 | 36.1 | 44.6 | 39.5 |
| | RocketKV-MT | **49.9** | 42.1 | 36.3 | 64.3 | **50.5** |

Table 5: Individual SCBench results for Llama3.1-8B-Ins

| Token Budget | Method | kv | prefix.suffix | vt | repoqa | qa.eng | qa.chn | choice.eng | many.shot | summary | mf | summary.needles | repoqa.and.kv |
|---|---|---|---|---|---|---|---|---|---|---|---|---|---|
| N/A | Full-KV | 76.2 | 34.6 | 37.6 | 48.0 | 21.5 | 20.9 | 72.6 | 59.3 | 37.5 | 12.0 | 60.4 | 68.8 |
| 1024 | Exact-TopK | 43.0 | 18.6 | 35.4 | 42.0 | 30.1 | 19.3 | 72.5 | 60.4 | 36.5 | 13.7 | 59.3 | 64.2 |
| | DuoAttention | 0.2 | 0.0 | 0.1 | 0.5 | 18.3 | 15.8 | 71.7 | 60.0 | 28.3 | 12.2 | 22.9 | 0.3 |
| | SnapKV | 0.0 | 0.0 | 0.6 | 0.9 | 8.7 | 12.2 | 46.2 | **69.6** | 27.2 | 7.2 | 21.2 | 0.1 |
| | Quest | 0.0 | 0.0 | 0.1 | 0.0 | 7.2 | 10.1 | 72.1 | 50.7 | 11.7 | 13.2 | 8.0 | 0.0 |
| | SparQ | 0.0 | 0.0 | 0.0 | 0.0 | 4.6 | 3.7 | 68.6 | 47.8 | 6.5 | **16.5** | 1.9 | 0.0 |
| | HSA | 0.0 | 0.0 | 12.4 | 5.7 | 22.9 | 9.0 | 71.2 | 59.6 | 9.7 | 12.2 | 35.3 | 4.7 |
| | RocketKV | 10.2 | 2.4 | 30.4 | 5.0 | 16.5 | 15.2 | 54.1 | 65.6 | 30.0 | 10.3 | 49.1 | 3.3 |
| | RocketKV-MT | **64.4** | **21.8** | **35.2** | 20.5 | 31.3 | **18.5** | 72.5 | 61.5 | **36.2** | 12.7 | **61.1** | **51.4** |
| 2048 | Exact-TopK | 53.4 | 23.8 | 36.7 | 43.9 | 29.6 | 18.7 | 72.5 | 60.0 | 37.2 | 12.5 | 59.7 | 66.1 |
| | DuoAttention | 0.2 | 0.0 | 0.1 | 0.5 | 18.8 | 16.0 | 71.2 | 61.1 | 28.9 | 11.8 | 24.0 | 0.3 |
| | SnapKV | 0.0 | 0.2 | 3.2 | 1.4 | 9.9 | 12.0 | 44.9 | **70.0** | 27.9 | 9.7 | 23.6 | 0.6 |
| | Quest | 0.0 | 0.0 | 5.9 | 0.0 | 12.8 | 12.8 | **72.1** | 50.0 | 14.1 | **15.5** | 23.2 | 0.3 |
| | SparQ | 0.0 | 0.0 | 0.4 | 0.0 | 9.3 | 6.9 | 68.1 | 61.1 | 6.0 | 10.2 | 8.2 | 0.0 |
| | HSA | 0.0 | 0.0 | 25.3 | 23.6 | 29.6 | 16.4 | 71.7 | 61.1 | 31.9 | 14.0 | 57.2 | 49.6 |
| | RocketKV | 11.6 | 4.2 | 33.7 | 8.9 | 19.7 | 14.8 | 56.8 | 64.1 | 32.3 | 9.8 | 55.1 | 6.0 |
| | RocketKV-MT | **61.2** | **30.6** | **34.7** | 31.8 | 30.4 | 19.5 | 71.7 | 61.5 | **36.9** | 12.5 | **61.0** | **58.1** |
| 4096 | Exact-TopK | 66.0 | 26.8 | 37.4 | 46.6 | 29.6 | 18.3 | 73.9 | 59.6 | 37.5 | 12.2 | 59.7 | 67.9 |
| | DuoAttention | 0.2 | 0.0 | 0.0 | 0.5 | 19.7 | 15.4 | 71.2 | 60.7 | 28.6 | 12.5 | 25.1 | 0.3 |
| | SnapKV | 0.0 | 0.2 | 10.3 | 2.7 | 12.2 | 12.7 | 48.9 | 67.8 | 28.6 | 8.8 | 25.6 | 0.9 |
| | Quest | 0.0 | 0.0 | 25.7 | 0.5 | 21.4 | 13.5 | 71.7 | 55.9 | 16.5 | 13.0 | 44.8 | 1.3 |
| | SparQ | 0.0 | 0.0 | 9.0 | 17.0 | 22.3 | 8.8 | 71.2 | 60.0 | 16.2 | 11.3 | 42.6 | 18.9 |
| | HSA | 4.0 | 0.8 | 34.4 | 34.3 | 29.3 | **18.6** | 72.5 | 59.3 | 36.4 | **13.3** | 60.3 | **61.5** |
| | RocketKV | 17.6 | 4.8 | 34.2 | 13.2 | 20.3 | 17.8 | 62.9 | 62.2 | 34.6 | 10.8 | 58.2 | 9.7 |
| | RocketKV-MT | **65.8** | **32.6** | **36.1** | 37.5 | 30.7 | 18.5 | **73.0** | 60.7 | **37.4** | 12.3 | **60.4** | 60.2 |
| 8192 | Exact-TopK | 72.6 | 33.0 | 38.3 | 45.9 | 30.7 | 18.9 | 73.9 | 59.6 | 37.4 | 12.2 | 60.4 | 68.9 |
| | DuoAttention | 0.2 | 0.0 | 0.0 | 0.5 | 21.2 | 16.0 | 71.7 | 60.7 | 29.3 | 11.8 | 32.4 | 0.6 |
| | SnapKV | 0.0 | 0.2 | 14.9 | 4.1 | 13.0 | 14.4 | 49.7 | **69.6** | 28.5 | 9.0 | 36.5 | 1.4 |
| | Quest | 0.2 | 0.0 | 30.0 | 10.7 | 27.5 | 17.1 | 71.2 | 59.6 | 22.3 | **14.0** | 44.3 | 19.6 |
| | SparQ | 0.0 | 0.4 | 20.0 | **50.7** | 27.8 | 15.6 | 71.7 | 59.3 | 35.5 | 12.2 | 60.5 | **67.6** |
| | HSA | 47.4 | 20.0 | 34.8 | 44.8 | 29.9 | **19.7** | 73.0 | 59.6 | 36.8 | 13.0 | 60.1 | 67.5 |
| | RocketKV | 25.0 | 7.4 | 36.3 | 23.2 | 23.5 | 17.8 | 68.2 | 60.4 | 36.2 | 10.7 | **60.6** | 17.6 |
| | RocketKV-MT | **70.4** | **35.0** | **37.7** | 44.3 | 31.0 | 18.7 | **73.9** | 60.0 | **37.2** | 11.5 | 60.5 | 65.2 |
| 16384 | Exact-TopK | 74.8 | 35.2 | 38.5 | 48.4 | 30.7 | 19.1 | 73.0 | 59.3 | 37.4 | 12.3 | 60.4 | 69.5 |
| | DuoAttention | 0.2 | 0.0 | 1.1 | 3.6 | 24.1 | 16.7 | 71.2 | 59.6 | 29.9 | 12.5 | 46.4 | 14.5 |
| | SnapKV | 0.0 | 0.4 | 20.4 | 5.9 | 14.2 | 14.5 | 53.2 | **64.1** | 29.9 | 9.5 | 47.8 | 1.7 |
| | Quest | 2.8 | 2.2 | 32.6 | 43.0 | 28.7 | 18.2 | 72.6 | 59.6 | 36.1 | **13.0** | 59.8 | 63.9 |
| | SparQ | 60.8 | 26.0 | 34.1 | 48.4 | 29.9 | 18.0 | 73.4 | 59.3 | 36.6 | 12.3 | 60.5 | **68.5** |
| | HSA | 70.8 | 30.8 | **38.0** | 50.0 | 29.9 | **19.0** | 73.4 | 59.6 | **37.6** | 12.7 | 60.7 | 68.2 |
| | RocketKV | 39.6 | 10.8 | 37.6 | 33.0 | 25.5 | 18.7 | 71.7 | 58.9 | 37.3 | 12.2 | **60.9** | 28.3 |
| | RocketKV-MT | **74.6** | **37.2** | **38.0** | 46.1 | 30.1 | 18.9 | 73.4 | 59.6 | **37.6** | 11.7 | 60.8 | 67.8 |

Table 6: Results of RULER-16K for Llama3.1-8B-Ins

| Token Budget | Method | single1 | single2 | single3 | multikey1 | multikey2 | multikey3 | multivalue | multiquery | vt | cwe | fwe | qa-1 | qa-2 | avg |
|---|---|---|---|---|---|---|---|---|---|---|---|---|---|---|---|
| N/A | Full-KV | 100.0 | 100.0 | 100.0 | 100.0 | 100.0 | 100.0 | 100.0 | 100.0 | 99.2 | 60.6 | 92.7 | 84.0 | 50.0 | 91.3 |
| 256 | Exact-TopK | 100.0 | 100.0 | 100.0 | 100.0 | 100.0 | 98.0 | 99.0 | 100.0 | 94.8 | 65.8 | 87.3 | 80.0 | 50.0 | 90.4 |
| | DuoAttention | 0.0 | 2.0 | 2.0 | 2.0 | 0.0 | 0.0 | 6.0 | 5.0 | 1.6 | 2.8 | 67.3 | 42.0 | 28.0 | 12.2 |
| | SnapKV | 86.0 | 52.0 | 0.0 | 30.0 | 20.0 | 0.0 | 1.5 | 1.0 | 0.8 | 0.0 | 32.7 | 78.0 | 40.0 | 26.3 |
| | Quest | 2.0 | 4.0 | 0.0 | 12.0 | 0.0 | 0.0 | 1.0 | 2.5 | 0.0 | 0.6 | 33.3 | 6.0 | 8.0 | 5.3 |
| | SparQ | 4.0 | 0.0 | 0.0 | 0.0 | 2.0 | 0.0 | 1.0 | 0.0 | 5.2 | 5.6 | 53.3 | 26.0 | 16.0 | 8.7 |
| | HSA | **100.0** | 96.0 | 74.0 | 96.0 | 66.0 | 0.0 | 80.5 | 85.0 | 82.8 | **11.8** | 80.7 | 80.0 | 44.0 | 69.0 |
| | RocketKV | **100.0** | **100.0** | **100.0** | **100.0** | **100.0** | **100.0** | **98.0** | **100.0** | **92.0** | 9.6 | **92.0** | **82.0** | **48.0** | **86.3** |
| 512 | Exact-TopK | 100.0 | 100.0 | 100.0 | 100.0 | 100.0 | 100.0 | 99.0 | 100.0 | 96.8 | 69.8 | 86.7 | 80.0 | 50.0 | 90.9 |
| | DuoAttention | 2.0 | 2.0 | 4.0 | 2.0 | 4.0 | 0.0 | 6.0 | 5.0 | 3.2 | 0.4 | 90.0 | 42.0 | 20.0 | 13.9 |
| | SnapKV | **100.0** | 88.0 | 0.0 | 68.0 | 42.0 | 0.0 | 19.5 | 31.5 | 12.0 | 0.6 | 66.0 | 76.0 | 42.0 | 42.0 |
| | Quest | 98.0 | 74.0 | 0.0 | **100.0** | 26.0 | 0.0 | 22.0 | 16.0 | 60.4 | 4.0 | 52.7 | 34.0 | 22.0 | 39.2 |
| | SparQ | 60.0 | 54.0 | 0.0 | 28.0 | 2.0 | 0.0 | 23.0 | 11.0 | 21.2 | 13.0 | 73.3 | 36.0 | 30.0 | 27.0 |
| | HSA | **100.0** | **100.0** | 98.0 | **100.0** | 90.0 | 66.0 | 95.0 | 99.5 | 90.4 | 24.0 | 88.7 | **82.0** | 52.0 | 83.5 |
| | RocketKV | **100.0** | **100.0** | **100.0** | **100.0** | **100.0** | **100.0** | **99.5** | **100.0** | **92.8** | **31.0** | **92.7** | **82.0** | **54.0** | **88.6** |
| 1024 | Exact-TopK | 100.0 | 100.0 | 100.0 | 100.0 | 100.0 | 100.0 | 99.5 | 100.0 | 98.8 | 71.4 | 89.3 | 80.0 | 50.0 | 91.5 |
| | DuoAttention | 2.0 | 2.0 | 4.0 | 2.0 | 6.0 | 2.0 | 6.0 | 4.5 | 2.8 | 0.0 | 90.0 | 34.0 | 22.0 | 13.6 |
| | SnapKV | **100.0** | 92.0 | 0.0 | 82.0 | 64.0 | 0.0 | 41.0 | 68.0 | 78.8 | 3.2 | 84.0 | 80.0 | 46.0 | 56.8 |
| | Quest | 94.0 | **100.0** | 38.0 | **100.0** | 84.0 | 0.0 | 66.5 | 79.5 | 80.4 | 9.2 | 70.0 | 68.0 | 50.0 | 64.6 |
| | SparQ | **100.0** | **100.0** | **100.0** | **100.0** | 98.0 | 54.0 | 97.0 | 99.0 | 85.2 | **66.0** | 86.7 | **84.0** | **54.0** | 86.5 |
| | HSA | **100.0** | **100.0** | **100.0** | **100.0** | **100.0** | 98.0 | **99.5** | **100.0** | 94.4 | 51.8 | 88.0 | 82.0 | 50.0 | 89.5 |
| | RocketKV | **100.0** | **100.0** | **100.0** | **100.0** | **100.0** | **100.0** | 99.0 | **100.0** | **95.2** | 56.2 | **93.3** | 82.0 | 52.0 | **90.6** |
| 2048 | Exact-TopK | 100.0 | 100.0 | 100.0 | 100.0 | 100.0 | 100.0 | 99.5 | 100.0 | 99.2 | 64.2 | 90.0 | 82.0 | 50.0 | 91.1 |
| | DuoAttention | 4.0 | 20.0 | 6.0 | 20.0 | 24.0 | 4.0 | 6.0 | 8.0 | 2.4 | 2.6 | 89.3 | 52.0 | 34.0 | 20.9 |
| | SnapKV | **100.0** | 98.0 | 0.0 | 92.0 | 80.0 | 2.0 | 61.0 | 86.5 | 92.8 | 13.0 | 87.3 | 76.0 | **50.0** | 64.5 |
| | Quest | 94.0 | 96.0 | 60.0 | 98.0 | 92.0 | 28.0 | 91.5 | 90.5 | 80.8 | 19.8 | 86.7 | 80.0 | **50.0** | 74.4 |
| | SparQ | **100.0** | **100.0** | **100.0** | **100.0** | **100.0** | **100.0** | **100.0** | **100.0** | **99.2** | **77.2** | 88.0 | **82.0** | **50.0** | **92.0** |
| | HSA | **100.0** | **100.0** | **100.0** | **100.0** | **100.0** | **100.0** | 99.5 | **100.0** | 97.6 | 66.8 | 86.7 | 80.0 | **50.0** | 90.8 |
| | RocketKV | **100.0** | **100.0** | **100.0** | **100.0** | **100.0** | **100.0** | 99.5 | **100.0** | 95.6 | 67.6 | **95.3** | 80.0 | **50.0** | 91.4 |
| 4096 | Exact-TopK | 100.0 | 100.0 | 100.0 | 100.0 | 100.0 | 100.0 | 98.5 | 100.0 | 99.2 | 57.4 | 90.7 | 84.0 | 50.0 | 90.8 |
| | DuoAttention | 20.0 | 88.0 | 86.0 | 98.0 | **100.0** | 46.0 | 71.5 | 77.0 | 58.0 | 14.2 | 88.0 | **82.0** | 52.0 | 67.7 |
| | SnapKV | **100.0** | **100.0** | 10.0 | 96.0 | 96.0 | 4.0 | 76.5 | 94.5 | 94.4 | 23.4 | 92.0 | 78.0 | **54.0** | 70.7 |
| | Quest | **100.0** | 94.0 | 96.0 | **100.0** | 96.0 | 98.0 | 98.0 | 98.0 | 98.8 | 54.6 | 88.0 | **82.0** | 50.0 | 88.7 |
| | SparQ | **100.0** | **100.0** | **100.0** | **100.0** | **100.0** | **100.0** | **100.0** | **100.0** | **99.2** | 75.2 | 92.0 | **82.0** | 50.0 | 92.2 |
| | HSA | **100.0** | **100.0** | **100.0** | **100.0** | **100.0** | **100.0** | **100.0** | **100.0** | **99.2** | **77.0** | 91.3 | **82.0** | 50.0 | **92.3** |
| | RocketKV | **100.0** | **100.0** | **100.0** | **100.0** | **100.0** | **100.0** | **100.0** | **100.0** | **99.2** | 72.8 | **93.3** | 80.0 | 50.0 | 91.9 |

Table 7: Results of RULER-32K for Llama3.1-8B-Ins

| Token Budget | Method | single1 | single2 | single3 | multikey1 | multikey2 | multikey3 | multivalue | multiquery | vt | cwe | fwe | qa-1 | qa-2 | avg |
|---|---|---|---|---|---|---|---|---|---|---|---|---|---|---|---|
| N/A | Full-KV | 100.0 | 100.0 | 100.0 | 100.0 | 100.0 | 100.0 | 99.0 | 99.5 | 96.4 | 6.0 | 94.0 | 74.0 | 54.0 | 86.4 |
| 256 | Exact-TopK | 100.0 | 100.0 | 100.0 | 100.0 | 100.0 | 90.0 | 99.0 | 99.5 | 91.6 | 24.6 | 81.3 | 76.0 | 54.0 | 85.8 |
| | DuoAttention | 0.0 | 4.0 | 4.0 | 4.0 | 2.0 | 0.0 | 4.5 | 4.5 | 2.4 | 1.0 | 52.0 | 38.0 | 32.0 | 11.4 |
| | SnapKV | 96.0 | 68.0 | 0.0 | 24.0 | 26.0 | 0.0 | 2.5 | 0.0 | 0.4 | 0.0 | 18.0 | 62.0 | 44.0 | 26.2 |
| | Quest | 0.0 | 0.0 | 0.0 | 0.0 | 0.0 | 0.0 | 0.0 | 0.0 | 0.0 | **1.2** | 22.0 | 0.0 | 0.0 | 1.8 |
| | SparQ | 44.0 | 0.0 | 0.0 | 0.0 | 0.0 | 0.0 | 0.0 | 0.0 | 2.8 | 0.4 | 18.7 | 4.0 | 8.0 | 6.0 |
| | HSA | 98.0 | 96.0 | 38.0 | 94.0 | 16.0 | 0.0 | 52.0 | 57.5 | 70.4 | 0.4 | 48.0 | 62.0 | 38.0 | 51.6 |
| | RocketKV | **100.0** | **100.0** | **100.0** | **100.0** | **100.0** | **100.0** | **96.5** | **98.0** | **87.6** | 0.8 | **77.3** | **78.0** | **54.0** | **84.0** |
| 512 | Exact-TopK | 100.0 | 100.0 | 100.0 | 100.0 | 100.0 | 100.0 | 99.0 | 99.0 | 90.8 | 17.2 | 82.0 | 78.0 | 52.0 | 86.0 |
| | DuoAttention | 0.0 | 4.0 | 8.0 | 4.0 | 2.0 | 0.0 | 4.5 | 4.5 | 3.6 | 0.0 | 70.7 | 40.0 | 30.0 | 13.2 |
| | SnapKV | **100.0** | 86.0 | 0.0 | 64.0 | 50.0 | 0.0 | 15.0 | 23.0 | 9.6 | 0.0 | 42.0 | 72.0 | 44.0 | 38.9 |
| | Quest | **100.0** | 26.0 | 16.0 | 4.0 | 2.0 | 0.0 | 1.5 | 1.5 | 7.2 | 1.2 | 45.3 | 10.0 | 16.0 | 17.7 |
| | SparQ | 62.0 | 0.0 | 0.0 | 0.0 | 0.0 | 0.0 | 0.5 | 0.5 | 3.2 | **1.6** | 26.7 | 24.0 | 26.0 | 11.1 |
| | HSA | **100.0** | 96.0 | 90.0 | **100.0** | 72.0 | 6.0 | 89.0 | 96.0 | 83.2 | 1.2 | 60.7 | **80.0** | 52.0 | 71.2 |
| | RocketKV | **100.0** | **100.0** | **100.0** | **100.0** | **100.0** | **100.0** | **98.0** | **100.0** | **89.2** | **1.6** | 78.0 | 76.0 | **56.0** | **84.5** |
| 1024 | Exact-TopK | 100.0 | 100.0 | 100.0 | 100.0 | 100.0 | 100.0 | 99.0 | 98.5 | 91.6 | 10.6 | 87.3 | 74.0 | 52.0 | 85.6 |
| | DuoAttention | 0.0 | 4.0 | 8.0 | 4.0 | 2.0 | 0.0 | 4.5 | 4.0 | 4.0 | 0.0 | 71.3 | 42.0 | 34.0 | 13.7 |
| | SnapKV | **100.0** | 92.0 | 0.0 | 84.0 | 60.0 | 0.0 | 48.5 | 68.5 | 72.0 | 1.4 | 56.7 | 72.0 | 52.0 | 54.4 |
| | Quest | 90.0 | 78.0 | 18.0 | 76.0 | 46.0 | 0.0 | 34.0 | 31.0 | 38.8 | 1.0 | 50.0 | 20.0 | 22.0 | 38.8 |
| | SparQ | 84.0 | 54.0 | 0.0 | 46.0 | 8.0 | 0.0 | 27.0 | 23.0 | 21.6 | 4.2 | 53.3 | 44.0 | 38.0 | 31.0 |
| | HSA | **100.0** | **100.0** | **100.0** | **100.0** | **100.0** | 82.0 | 97.5 | 99.5 | 88.0 | 6.0 | 72.7 | **80.0** | **58.0** | 83.4 |
| | RocketKV | **100.0** | **100.0** | **100.0** | **100.0** | **100.0** | **100.0** | **98.0** | **100.0** | **90.4** | **6.2** | **78.7** | 78.0 | 52.0 | **84.9** |
| 2048 | Exact-TopK | 100.0 | 100.0 | 100.0 | 100.0 | 100.0 | 100.0 | 99.0 | 98.5 | 95.6 | 7.0 | 90.7 | 74.0 | 54.0 | 86.1 |
| | DuoAttention | 0.0 | 4.0 | 8.0 | 8.0 | 8.0 | 0.0 | 4.5 | 3.5 | 3.2 | 0.0 | 72.0 | 40.0 | 36.0 | 14.2 |
| | SnapKV | **100.0** | 92.0 | 2.0 | 92.0 | 70.0 | 2.0 | 68.0 | 85.5 | 85.2 | 6.6 | 62.0 | 70.0 | 50.0 | 60.4 |
| | Quest | 86.0 | **100.0** | 6.0 | 86.0 | 76.0 | 0.0 | 43.5 | 66.0 | 54.4 | 2.2 | 68.0 | 60.0 | 42.0 | 53.1 |
| | SparQ | **100.0** | **100.0** | 98.0 | 98.0 | 96.0 | 36.0 | 98.0 | 98.0 | 82.0 | **34.2** | 74.7 | **76.0** | **58.0** | 80.7 |
| | HSA | **100.0** | **100.0** | **100.0** | **100.0** | **100.0** | **100.0** | 99.0 | 99.0 | 92.0 | 14.6 | 76.7 | **76.0** | 56.0 | 85.6 |
| | RocketKV | **100.0** | **100.0** | **100.0** | **100.0** | **100.0** | **100.0** | 99.0 | **100.0** | **93.2** | 17.4 | **80.0** | **76.0** | 52.0 | **86.0** |
| 4096 | Exact-TopK | 100.0 | 100.0 | 100.0 | 100.0 | 100.0 | 100.0 | 99.0 | 98.5 | 96.8 | 7.0 | 93.3 | 74.0 | 54.0 | 86.4 |
| | DuoAttention | 4.0 | 26.0 | 14.0 | 36.0 | 16.0 | 2.0 | 5.5 | 10.5 | 3.2 | 0.0 | 76.7 | 50.0 | 48.0 | 22.5 |
| | SnapKV | **100.0** | 96.0 | 14.0 | 98.0 | 80.0 | 2.0 | 78.5 | 94.5 | 88.8 | 8.8 | 69.3 | 74.0 | **54.0** | 66.0 |
| | Quest | 82.0 | **100.0** | 70.0 | 90.0 | 92.0 | 38.0 | 87.0 | 87.5 | 88.4 | 5.6 | 78.7 | **78.0** | 52.0 | 73.0 |
| | SparQ | **100.0** | **100.0** | **100.0** | **100.0** | **100.0** | **100.0** | **99.5** | **99.0** | **95.2** | **20.6** | 86.7 | 74.0 | 52.0 | **86.7** |
| | HSA | **100.0** | **100.0** | **100.0** | **100.0** | **100.0** | **100.0** | **99.5** | **99.0** | 93.2 | 19.8 | 79.3 | 74.0 | 52.0 | 85.9 |
| | RocketKV | **100.0** | **100.0** | **100.0** | **100.0** | **100.0** | **100.0** | **99.5** | **99.0** | 93.6 | 17.2 | **87.3** | 76.0 | 52.0 | 86.5 |

Table 8: Results of RULER-64K for Llama3.1-8B-Ins

| Token Budget | Method | single1 | single2 | single3 | multikey1 | multikey2 | multikey3 | multivalue | multiquery | vt | cwe | fwe | qa-1 | qa-2 | avg |
|---|---|---|---|---|---|---|---|---|---|---|---|---|---|---|---|
| N/A | Full-KV | 100.0 | 100.0 | 100.0 | 100.0 | 100.0 | 100.0 | 95.5 | 97.5 | 94.8 | 0.0 | 83.3 | 76.0 | 56.0 | 84.9 |
| 256 | Exact-TopK | 100.0 | 100.0 | 100.0 | 100.0 | 100.0 | 58.0 | 94.5 | 97.5 | 93.6 | 0.4 | 63.3 | 72.0 | 56.0 | 79.6 |
| | DuoAttention | 0.0 | 2.0 | 0.0 | 8.0 | 0.0 | 0.0 | 5.0 | 3.0 | 0.0 | **0.4** | **84.0** | 44.0 | 32.0 | 13.7 |
| | SnapKV | **100.0** | 66.0 | 0.0 | 20.0 | 12.0 | 0.0 | 0.5 | 0.0 | 0.4 | 0.0 | 4.0 | 58.0 | 38.0 | 23.0 |
| | Quest | 0.0 | 0.0 | 0.0 | 0.0 | 0.0 | 0.0 | 0.0 | 0.0 | 0.0 | **0.4** | 25.3 | 0.0 | 2.0 | 2.1 |
| | SparQ | 0.0 | 0.0 | 0.0 | 0.0 | 0.0 | 0.0 | 0.0 | 0.0 | 0.0 | 0.0 | 0.0 | 0.0 | 0.0 | 0.0 |
| | HSA | **100.0** | 24.0 | 0.0 | 28.0 | 2.0 | 0.0 | 4.5 | 6.0 | 48.8 | 0.2 | 22.7 | 36.0 | 24.0 | 22.8 |
| | RocketKV | **100.0** | **100.0** | **100.0** | **100.0** | **94.0** | **94.0** | **92.0** | **98.5** | 84.8 | 0.0 | 70.0 | **68.0** | **54.0** | **81.2** |
| 512 | Exact-TopK | 100.0 | 100.0 | 100.0 | 100.0 | 100.0 | 80.0 | 98.0 | 98.0 | 92.8 | 0.4 | 67.3 | 76.0 | 54.0 | 82.0 |
| | DuoAttention | 0.0 | 2.0 | 0.0 | 8.0 | 0.0 | 0.0 | 5.0 | 3.5 | 0.4 | 0.8 | **96.0** | 38.0 | 28.0 | 14.0 |
| | SnapKV | **100.0** | 78.0 | 0.0 | 62.0 | 24.0 | 0.0 | 5.5 | 12.5 | 13.6 | 0.0 | 44.7 | 60.0 | 46.0 | 34.3 |
| | Quest | 0.0 | 0.0 | 0.0 | 0.0 | 0.0 | 0.0 | 0.0 | 0.5 | 0.0 | 0.4 | 34.7 | 4.0 | 6.0 | 3.5 |
| | SparQ | 20.0 | 0.0 | 0.0 | 0.0 | 0.0 | 0.0 | 0.0 | 0.0 | 2.0 | **1.0** | 15.3 | 4.0 | 14.0 | 4.3 |
| | HSA | **100.0** | 92.0 | 26.0 | 82.0 | 22.0 | 0.0 | 52.5 | 71.5 | 74.8 | 0.2 | 56.7 | 62.0 | 46.0 | 52.7 |
| | RocketKV | **100.0** | **100.0** | **100.0** | **100.0** | **92.0** | **100.0** | **96.5** | **99.0** | 88.8 | 0.2 | 76.0 | **74.0** | **56.0** | **83.3** |
| 1024 | Exact-TopK | 100.0 | 100.0 | 100.0 | 100.0 | 100.0 | 96.0 | 98.5 | 98.5 | 93.6 | 0.2 | 74.7 | 74.0 | 54.0 | 83.8 |
| | DuoAttention | 0.0 | 2.0 | 0.0 | 8.0 | 0.0 | 2.0 | 5.0 | 3.5 | 0.4 | 0.8 | **96.0** | 40.0 | 30.0 | 14.4 |
| | SnapKV | **100.0** | 82.0 | 0.0 | 88.0 | 38.0 | 0.0 | 28.0 | 47.0 | 55.2 | 0.0 | 56.0 | 66.0 | 46.0 | 46.6 |
| | Quest | **100.0** | 32.0 | 10.0 | 24.0 | 2.0 | 0.0 | 1.5 | 6.0 | 35.6 | 0.4 | 49.3 | 12.0 | 12.0 | 21.9 |
| | SparQ | 48.0 | 0.0 | 0.0 | 2.0 | 0.0 | 0.0 | 0.5 | 0.0 | 12.4 | 0.8 | 18.7 | 20.0 | 24.0 | 9.7 |
| | HSA | **100.0** | **100.0** | **100.0** | **100.0** | 70.0 | 2.0 | 92.0 | 98.0 | 84.4 | 0.2 | 70.0 | **72.0** | 54.0 | 72.5 |
| | RocketKV | **100.0** | **100.0** | **100.0** | **100.0** | **98.0** | **100.0** | **96.0** | **99.5** | 89.2 | 1.0 | 72.7 | **72.0** | **56.0** | **83.4** |
| 2048 | Exact-TopK | 100.0 | 100.0 | 100.0 | 100.0 | 100.0 | 98.0 | 97.0 | 99.0 | 94.0 | 0.0 | 78.7 | 74.0 | 52.0 | 84.1 |
| | DuoAttention | 0.0 | 2.0 | 0.0 | 8.0 | 0.0 | 2.0 | 5.0 | 3.5 | 0.4 | 0.8 | **94.0** | 44.0 | 28.0 | 14.4 |
| | SnapKV | **100.0** | 88.0 | 2.0 | 96.0 | 52.0 | 0.0 | 44.5 | 79.0 | 82.0 | 0.6 | 56.0 | 72.0 | 50.0 | 55.5 |
| | Quest | **100.0** | 78.0 | 70.0 | 88.0 | 48.0 | 0.0 | 19.5 | 39.5 | 76.8 | 0.0 | 63.3 | 38.0 | 38.0 | 50.7 |
| | SparQ | 98.0 | 46.0 | 0.0 | 54.0 | 10.0 | 0.0 | 31.0 | 27.0 | 41.2 | **5.6** | 52.0 | 36.0 | 36.0 | 33.6 |
| | HSA | **100.0** | **100.0** | 98.0 | **100.0** | 96.0 | 54.0 | **96.0** | 99.0 | 85.6 | 0.8 | 68.7 | **78.0** | **56.0** | 79.4 |
| | RocketKV | **100.0** | **100.0** | **100.0** | **100.0** | **100.0** | **98.0** | 94.5 | **99.5** | 91.6 | 0.8 | 72.0 | 74.0 | **56.0** | **83.6** |
| 4096 | Exact-TopK | 100.0 | 100.0 | 100.0 | 100.0 | 100.0 | 100.0 | 96.5 | 99.0 | 95.6 | 0.0 | 80.0 | 76.0 | 52.0 | 84.5 |
| | DuoAttention | 0.0 | 4.0 | 0.0 | 8.0 | 0.0 | 2.0 | 4.5 | 3.5 | 0.4 | 0.8 | **91.3** | 44.0 | 32.0 | 14.7 |
| | SnapKV | **100.0** | 90.0 | 20.0 | 96.0 | 54.0 | 2.0 | 67.0 | 90.5 | 91.2 | 0.8 | 60.0 | **76.0** | 50.0 | 61.3 |
| | Quest | **100.0** | 76.0 | 20.0 | **100.0** | 76.0 | 0.0 | 48.0 | 77.5 | 78.8 | 0.0 | 72.0 | **76.0** | 52.0 | 59.7 |
| | SparQ | **100.0** | **100.0** | **100.0** | **100.0** | 94.0 | 32.0 | 94.0 | 96.5 | 86.4 | **2.4** | 73.3 | 70.0 | 52.0 | 77.0 |
| | HSA | **100.0** | **100.0** | **100.0** | **100.0** | 98.0 | 96.0 | **96.5** | 98.0 | 90.8 | 0.2 | 70.0 | **76.0** | **56.0** | 83.2 |
| | RocketKV | **100.0** | **100.0** | **100.0** | **100.0** | **100.0** | **100.0** | **96.5** | **100.0** | 92.8 | 1.2 | 76.0 | **76.0** | **56.0** | **84.5** |

Table 9: Results of RULER-96K for Llama3.1-8B-Ins

| Token Budget | Method | single1 | single2 | single3 | multikey1 | multikey2 | multikey3 | multivalue | multiquery | vt | cwe | fwe | qa-1 | qa-2 | avg |
|---|---|---|---|---|---|---|---|---|---|---|---|---|---|---|---|
| N/A | Full-KV | 100.0 | 100.0 | 100.0 | 100.0 | 94.0 | 88.0 | 98.5 | 100.0 | 66.8 | 0.0 | 74.7 | 78.0 | 50.0 | 80.8 |
| 256 | Exact-TopK | 100.0 | 100.0 | 100.0 | 100.0 | 84.0 | 14.0 | 95.5 | 99.0 | 67.2 | 0.4 | 56.0 | 82.0 | 50.0 | 72.9 |
| | DuoAttention | 0.0 | 2.0 | 4.0 | 6.0 | 0.0 | 0.0 | 3.5 | 5.0 | 1.6 | 0.0 | 38.7 | 40.0 | 32.0 | 10.2 |
| | SnapKV | 90.0 | 66.0 | 0.0 | 24.0 | 0.0 | 0.0 | 0.0 | 0.0 | 0.8 | 0.0 | 17.3 | 58.0 | 40.0 | 22.8 |
| | Quest | 0.0 | 0.0 | 0.0 | 0.0 | 0.0 | 0.0 | 0.0 | 0.0 | 0.0 | **0.6** | 19.3 | 0.0 | 2.0 | 1.7 |
| | SparQ | 0.0 | 0.0 | 0.0 | 0.0 | 0.0 | 0.0 | 0.0 | 0.0 | 0.0 | 0.0 | 0.0 | 0.0 | 0.0 | 0.0 |
| | HSA | 0.0 | 0.0 | 0.0 | 0.0 | 0.0 | 0.0 | 0.0 | 0.0 | 4.8 | 0.0 | 11.3 | 4.0 | 4.0 | 1.9 |
| | RocketKV | **98.0** | **100.0** | **94.0** | **98.0** | **84.0** | **52.0** | **91.0** | **95.5** | 53.6 | 0.0 | 59.3 | 80.0 | 46.0 | 73.2 |
| 512 | Exact-TopK | 100.0 | 100.0 | 100.0 | 100.0 | 84.0 | 32.0 | 96.5 | 100.0 | 67.6 | 0.4 | 59.3 | 80.0 | 50.0 | 74.6 |
| | DuoAttention | 0.0 | 2.0 | 6.0 | 6.0 | 0.0 | 0.0 | 3.5 | 5.5 | 1.6 | 0.0 | **70.0** | 38.0 | 32.0 | 12.7 |
| | SnapKV | **98.0** | 70.0 | 0.0 | 50.0 | 6.0 | 0.0 | 2.5 | 5.5 | 6.0 | 0.0 | 25.3 | 54.0 | 36.0 | 27.2 |
| | Quest | 0.0 | 0.0 | 0.0 | 0.0 | 0.0 | 0.0 | 0.0 | 0.0 | 0.0 | 0.0 | 26.0 | 0.0 | 4.0 | 2.3 |
| | SparQ | 0.0 | 0.0 | 0.0 | 0.0 | 0.0 | 0.0 | 0.0 | 0.0 | 0.0 | 0.0 | 0.0 | 0.0 | 0.0 | 0.0 |
| | HSA | 94.0 | 46.0 | 0.0 | 18.0 | 0.0 | 0.0 | 4.0 | 7.5 | 42.0 | 0.0 | 28.0 | 20.0 | 16.0 | 21.2 |
| | RocketKV | **98.0** | **100.0** | **98.0** | **100.0** | **86.0** | **70.0** | **95.5** | **98.0** | 64.0 | 0.4 | 61.3 | 78.0 | 44.0 | 76.4 |
| 1024 | Exact-TopK | 100.0 | 100.0 | 100.0 | 100.0 | 90.0 | 58.0 | 96.0 | 100.0 | 68.4 | 0.0 | 61.3 | 80.0 | 48.0 | 77.1 |
| | DuoAttention | 0.0 | 2.0 | 6.0 | 6.0 | 0.0 | 0.0 | 3.5 | 5.5 | 1.6 | 0.0 | **70.7** | 40.0 | 32.0 | 12.9 |
| | SnapKV | **100.0** | 76.0 | 0.0 | 66.0 | 20.0 | 0.0 | 14.5 | 20.5 | 36.0 | 0.0 | 36.0 | 62.0 | 40.0 | 36.2 |
| | Quest | 94.0 | 0.0 | 0.0 | 10.0 | 0.0 | 0.0 | 2.5 | 2.5 | 0.4 | 0.2 | 58.0 | 2.0 | 8.0 | 13.7 |
| | SparQ | 36.0 | 0.0 | 0.0 | 0.0 | 0.0 | 0.0 | 0.0 | 0.0 | 1.2 | 0.6 | 15.3 | 2.0 | 0.0 | 4.2 |
| | HSA | **100.0** | 88.0 | 2.0 | 84.0 | 12.0 | 0.0 | 61.5 | 72.0 | **66.0** | **1.2** | 43.3 | 58.0 | 42.0 | 48.5 |
| | RocketKV | **100.0** | **100.0** | **100.0** | **100.0** | **88.0** | **76.0** | **95.0** | **99.0** | 62.0 | 0.4 | 61.3 | **76.0** | **48.0** | 77.4 |
| 2048 | Exact-TopK | 100.0 | 100.0 | 100.0 | 100.0 | 92.0 | 82.0 | 95.5 | 100.0 | 64.4 | 0.0 | 66.7 | 80.0 | 48.0 | 79.1 |
| | DuoAttention | 0.0 | 2.0 | 6.0 | 6.0 | 0.0 | 0.0 | 3.5 | 5.0 | 1.6 | 0.0 | **70.7** | 44.0 | 28.0 | 12.8 |
| | SnapKV | **100.0** | 94.0 | 0.0 | 90.0 | 22.0 | 0.0 | 28.0 | 57.0 | 50.8 | **0.4** | 39.3 | 64.0 | 42.0 | 45.2 |
| | Quest | **100.0** | 56.0 | 0.0 | 62.0 | 4.0 | 0.0 | 11.5 | 9.5 | 54.0 | 0.2 | 59.3 | 32.0 | 22.0 | 31.6 |
| | SparQ | 12.0 | 0.0 | 0.0 | 0.0 | 0.0 | 0.0 | 0.0 | 2.0 | 3.6 | 0.2 | 32.7 | 8.0 | 14.0 | 5.6 |
| | HSA | **100.0** | 96.0 | 74.0 | 98.0 | 54.0 | 0.0 | 89.0 | 96.5 | **77.2** | **0.4** | 52.0 | 78.0 | 48.0 | 66.4 |
| | RocketKV | **100.0** | **100.0** | **100.0** | **100.0** | **94.0** | **74.0** | **98.5** | **100.0** | 62.0 | **0.4** | 56.7 | **80.0** | **50.0** | 78.1 |
| 4096 | Exact-TopK | 100.0 | 100.0 | 100.0 | 100.0 | 96.0 | 88.0 | 96.5 | 100.0 | 65.6 | 0.0 | 71.3 | 78.0 | 46.0 | 80.1 |
| | DuoAttention | 0.0 | 2.0 | 6.0 | 8.0 | 0.0 | 0.0 | 3.5 | 5.0 | 1.2 | 0.0 | **72.0** | 44.0 | 32.0 | 13.4 |
| | SnapKV | **100.0** | **100.0** | 6.0 | **100.0** | 38.0 | 0.0 | 47.0 | 80.0 | 62.8 | 0.2 | 50.7 | 72.0 | 44.0 | 53.9 |
| | Quest | 98.0 | **100.0** | 0.0 | 94.0 | 36.0 | 2.0 | 40.5 | 63.0 | 45.2 | 0.0 | 58.7 | 36.0 | 36.0 | 46.9 |
| | SparQ | **100.0** | 62.0 | 2.0 | 60.0 | 12.0 | 0.0 | 68.5 | 62.5 | 46.0 | **0.4** | 46.7 | 48.0 | 42.0 | 42.3 |
| | HSA | **100.0** | **100.0** | 98.0 | **100.0** | 88.0 | 16.0 | 96.5 | 99.0 | **70.4** | 0.2 | 54.7 | 76.0 | 48.0 | 72.8 |
| | RocketKV | **100.0** | **100.0** | **100.0** | **100.0** | **94.0** | **82.0** | **97.5** | **100.0** | 66.8 | **0.4** | 56.7 | **78.0** | **50.0** | 78.9 |

Table 10: Average results of LongBench and NIAH for Mistral-7B-Ins-v0.2

| Token Budget | Method | Single QA | Multi. QA | Summ. | Few-shot | Synthetic | Code | LB Avg. | NIAH |
|---|---|---|---|---|---|---|---|---|---|
| N/A | Full-KV | 36.4 | 29.6 | 28.1 | 66.7 | 44.9 | 55.6 | 45.4 | 99.0 |
| 256 | Exact-TopK | 35.5 | 28.6 | 28.2 | 67.6 | 42.6 | 58.2 | 45.3 | 98.7 |
| | DuoAttention | 17.7 | 19.6 | 18.5 | 48.5 | 5.2 | 52.3 | 28.3 | 22.0 |
| | SnapKV | 23.9 | 21.1 | 19.3 | 50.6 | 31.6 | 50.4 | 33.6 | 34.0 |
| | Quest | 8.7 | 5.3 | 15.3 | 12.6 | 3.2 | 32.6 | 13.1 | 14.0 |
| | SparQ | 1.6 | 0.9 | 8.6 | 2.5 | 6.4 | 26.0 | 7.0 | 10.0 |
| | HSA | 29.7 | 21.9 | **25.5** | 58.5 | 25.0 | 50.9 | 37.0 | 56.7 |
| | RocketKV | **34.0** | **27.0** | 24.7 | **65.7** | **38.0** | **56.2** | **42.7** | **97.3** |
| 512 | Exact-TopK | 35.8 | 29.2 | 28.2 | 67.1 | 42.9 | 57.4 | 45.3 | 99.3 |
| | DuoAttention | 19.8 | 19.2 | 21.1 | 53.9 | 7.1 | 53.8 | 30.8 | 24.0 |
| | SnapKV | 27.3 | 23.7 | 21.1 | 56.2 | 37.3 | 53.3 | 37.6 | 45.7 |
| | Quest | 17.4 | 12.3 | 22.4 | 35.8 | 23.0 | 44.8 | 26.4 | 39.7 |
| | SparQ | 5.1 | 1.7 | 13.0 | 8.2 | 2.3 | 35.3 | 10.6 | 11.7 |
| | HSA | 34.2 | 27.4 | **27.2** | 65.9 | 38.3 | **57.2** | 43.5 | 85.7 |
| | RocketKV | **35.2** | **28.1** | 26.4 | **66.6** | **42.3** | 56.2 | **44.2** | **98.3** |
| 1024 | Exact-TopK | 35.8 | 29.5 | 28.2 | 67.0 | 42.8 | 56.7 | 45.2 | 99.3 |
| | DuoAttention | 21.2 | 19.4 | 22.4 | 59.1 | 8.1 | 55.0 | 32.6 | 30.0 |
| | SnapKV | 30.4 | 25.6 | 22.7 | 61.2 | **42.3** | 54.9 | 40.7 | 54.7 |
| | Quest | 25.4 | 18.6 | 25.5 | 56.2 | 36.3 | 53.3 | 36.9 | 79.3 |
| | SparQ | 13.1 | 4.1 | 16.9 | 18.1 | 0.5 | 43.5 | 16.3 | 20.0 |
| | HSA | 35.3 | 28.0 | **27.7** | **67.3** | 42.1 | **57.7** | **44.8** | 97.7 |
| | RocketKV | **36.0** | **28.5** | 27.5 | 66.8 | 42.2 | 55.9 | 44.7 | **98.7** |
| 2048 | Exact-TopK | 36.2 | 29.7 | 28.2 | 66.6 | 42.9 | 56.8 | 45.2 | 99.3 |
| | DuoAttention | 27.1 | 21.5 | 23.5 | 62.4 | 12.1 | 55.6 | 35.7 | 41.0 |
| | SnapKV | 33.3 | 26.8 | 24.4 | 64.1 | 42.6 | 55.7 | 42.6 | 65.7 |
| | Quest | 32.2 | 26.9 | 27.0 | 62.5 | 39.3 | 56.0 | 42.3 | 90.3 |
| | SparQ | 22.0 | 10.3 | 20.1 | 34.1 | 4.0 | 50.5 | 24.5 | 27.0 |
| | HSA | **36.2** | **29.0** | **28.2** | 67.2 | 42.2 | **57.2** | **45.2** | **99.0** |
| | RocketKV | 35.9 | 28.9 | 27.9 | 67.0 | **42.8** | 55.7 | 44.9 | **99.0** |
| 4096 | Exact-TopK | 36.2 | 29.6 | 28.1 | 66.5 | 43.0 | 56.5 | 45.2 | 99.3 |
| | DuoAttention | 32.1 | 25.3 | 25.6 | 65.4 | 26.7 | 56.4 | 40.6 | 62.0 |
| | SnapKV | 34.2 | 28.0 | 25.8 | 64.8 | **42.8** | 55.7 | 43.5 | 76.7 |
| | Quest | 35.1 | 29.1 | 27.9 | 66.3 | 42.7 | 56.3 | 44.7 | 96.0 |
| | SparQ | 30.6 | 20.3 | 24.8 | 59.4 | 37.7 | 53.3 | 38.9 | 50.3 |
| | HSA | **36.4** | **29.5** | 28.1 | **66.8** | **42.8** | **57.0** | **45.3** | **99.3** |
| | RocketKV | 36.2 | **29.5** | **28.3** | **66.8** | **42.8** | 55.0 | 45.0 | **99.3** |

Table 11: Individual results of LongBench for Mistral-7B-Ins-v0.2

| Token Budget | Method | narrativeqa | qasper | multifieldqa | hotpotqa | 2wikimqa | musique | gov-report | qmsum | multi-news | trec | triviaqa | samsum | pass-ret. | lcc | repobench-p | passagecnt |
|---|---|---|---|---|---|---|---|---|---|---|---|---|---|---|---|---|---|
| N/A | Full-KV | 26.8 | 33.1 | 49.3 | 42.8 | 27.3 | 18.8 | 33.0 | 24.2 | 27.1 | 71.0 | 86.2 | 42.8 | 87.0 | 56.9 | 54.3 | 2.8 |
| | Exact-TopK | 26.2 | 31.3 | 48.9 | 40.0 | 26.2 | 19.6 | 33.3 | 24.2 | 27.1 | 73.0 | 86.6 | 43.3 | 82.9 | 59.0 | 57.3 | 2.3 |
| | DuoAttention | 13.0 | 11.7 | 28.5 | 29.5 | 20.1 | 9.3 | 17.6 | 19.4 | 18.6 | 26.0 | 81.8 | 37.7 | 7.1 | 50.4 | 54.1 | 3.2 |
| | SnapKV | 18.3 | 15.1 | 38.3 | 30.4 | 19.2 | 13.8 | 16.5 | 21.4 | 19.9 | 32.0 | 81.7 | 38.1 | 59.1 | 49.0 | 51.8 | **4.0** |
| 256 | Quest | 2.1 | 9.1 | 14.9 | 6.0 | 7.8 | 2.1 | 13.0 | 10.8 | 22.2 | 14.0 | 14.1 | 9.6 | 5.9 | 39.6 | 25.6 | 0.4 |
| | SparQ | 0.3 | 1.7 | 2.9 | 0.9 | 1.5 | 0.1 | 5.3 | 6.7 | 13.8 | 0.0 | 7.1 | 0.4 | 12.7 | 30.0 | 22.0 | 0.2 |
| | HSA | 17.1 | 28.1 | 43.9 | 30.2 | 23.6 | 11.8 | **27.3** | 22.4 | **26.8** | 63.0 | 78.8 | 33.6 | 46.9 | 52.1 | 49.7 | 3.2 |
| | RocketKV | **24.7** | **28.4** | **48.9** | **37.9** | **24.7** | **18.5** | 25.9 | **23.6** | 24.7 | **69.0** | **86.1** | **42.1** | **73.5** | **55.0** | **57.4** | 2.4 |
| | Exact-TopK | 27.2 | 31.6 | 48.6 | 40.5 | 27.5 | 19.7 | 33.2 | 24.2 | 27.1 | 72.0 | 86.6 | 42.8 | 83.1 | 58.3 | 56.5 | 2.6 |
| | DuoAttention | 14.7 | 13.8 | 30.8 | 27.2 | 20.5 | 10.0 | 20.6 | 19.9 | 22.8 | 40.0 | 81.8 | 39.8 | 11.7 | 53.5 | 54.0 | 2.5 |
| | SnapKV | 19.6 | 19.5 | 42.8 | 34.6 | 21.9 | 14.7 | 19.6 | 21.7 | 21.9 | 46.0 | 83.5 | 39.0 | 72.1 | 52.3 | 54.2 | 2.5 |
| 512 | Quest | 6.0 | 17.0 | 29.1 | 18.1 | 13.8 | 5.1 | 22.8 | 18.9 | 25.5 | 35.0 | 50.7 | 21.8 | 43.1 | 48.3 | 41.2 | 2.9 |
| | SparQ | 0.1 | 4.1 | 10.9 | 1.8 | 2.8 | 0.5 | 6.8 | 11.9 | 20.3 | 4.5 | 16.2 | 3.9 | 4.4 | 45.2 | 25.4 | 0.2 |
| | HSA | 23.2 | **31.2** | 48.4 | **38.6** | 26.6 | 17.0 | **31.0** | 23.3 | **27.2** | **71.5** | 85.0 | 41.2 | 74.7 | **57.1** | **57.3** | 1.9 |
| | RocketKV | **25.9** | 30.9 | **48.9** | 38.0 | **27.0** | **19.2** | 29.0 | **24.2** | 25.9 | 70.5 | **86.9** | **42.3** | **81.6** | 56.4 | 56.0 | **3.0** |
| | Exact-TopK | 27.2 | 31.3 | 48.8 | 41.7 | 27.4 | 19.3 | 33.4 | 24.1 | 27.2 | 72.0 | 86.4 | 42.5 | 82.8 | 57.7 | 55.7 | 2.9 |
| | DuoAttention | 15.7 | 15.7 | 32.3 | 29.4 | 19.3 | 9.4 | 22.0 | 20.4 | 24.8 | 53.0 | 83.8 | 40.5 | 13.2 | 55.1 | 54.9 | 3.1 |
| | SnapKV | 21.7 | 23.1 | 46.3 | 35.7 | 23.4 | 17.7 | 21.8 | 22.8 | 23.5 | 57.5 | 85.2 | 40.8 | 81.4 | 53.8 | 55.9 | **3.2** |
| 1024 | Quest | 10.1 | 26.7 | 39.6 | 26.5 | 17.9 | 11.4 | 27.5 | 22.9 | 26.3 | 66.0 | 70.5 | 32.3 | 69.8 | 55.7 | 50.9 | 2.8 |
| | SparQ | 0.2 | 14.5 | 24.5 | 5.8 | 6.0 | 0.5 | 9.3 | 16.3 | 25.1 | 13.0 | 31.1 | 10.1 | 1.0 | 54.3 | 32.8 | 0.0 |
| | HSA | 26.1 | 31.6 | 48.2 | 38.4 | **26.8** | 18.8 | **32.1** | 23.5 | **27.3** | **73.0** | **86.5** | 42.5 | 81.8 | **58.8** | **56.7** | 2.5 |
| | RocketKV | **27.1** | **32.7** | 48.4 | 39.7 | 26.2 | **19.6** | 31.2 | **24.0** | 27.2 | 71.5 | 86.2 | **42.7** | 81.7 | 56.3 | 55.5 | 2.8 |
| | Exact-TopK | 27.3 | 32.8 | 48.6 | 41.8 | 27.9 | 19.4 | 33.3 | 24.0 | 27.4 | 71.0 | 86.4 | 42.4 | 82.8 | 57.9 | 55.6 | 3.0 |
| | DuoAttention | 17.2 | 24.5 | 39.5 | 32.9 | 21.8 | 9.9 | 23.5 | 20.6 | 26.6 | 61.5 | 84.3 | 41.3 | 20.7 | 56.1 | 55.0 | **3.4** |
| | SnapKV | 24.8 | 27.0 | 48.2 | 38.1 | 23.9 | 18.2 | 23.8 | 23.5 | 25.9 | 66.0 | 85.8 | 40.6 | 81.9 | 56.0 | 55.3 | 3.2 |
| 2048 | Quest | 18.3 | 31.5 | 46.7 | 36.9 | 25.5 | 18.3 | 31.0 | 23.1 | 27.0 | 70.5 | 79.6 | 37.4 | 76.3 | **58.0** | 54.1 | 2.3 |
| | SparQ | 2.9 | 29.3 | 33.8 | 9.0 | 21.4 | 0.6 | 16.7 | 16.9 | 26.6 | 35.5 | 47.6 | 19.2 | 6.9 | 57.2 | 43.8 | 1.2 |
| | HSA | **27.2** | 32.2 | **49.1** | 40.8 | **27.0** | 19.2 | **33.2** | 24.1 | **27.4** | 72.0 | **86.8** | **42.9** | 81.3 | 57.7 | **56.8** | 3.1 |
| | RocketKV | 27.1 | **32.3** | 48.5 | 40.4 | 26.7 | **19.7** | 32.1 | **24.4** | 27.1 | 72.0 | 86.4 | 42.5 | **82.7** | 56.0 | 55.3 | 2.8 |
| | Exact-TopK | 27.7 | 32.7 | 48.2 | 41.5 | 27.9 | 19.4 | 33.1 | 24.1 | 27.2 | 71.0 | 86.2 | 42.4 | 83.3 | 57.8 | 55.3 | 2.8 |
| | DuoAttention | 19.1 | 31.6 | 45.6 | 36.6 | 26.8 | 12.4 | 27.7 | 21.8 | **27.2** | 70.0 | 85.0 | 41.2 | 51.2 | 56.8 | 56.1 | 2.2 |
| | SnapKV | 25.7 | 29.4 | 47.6 | 39.7 | 25.2 | 19.0 | 26.5 | 24.2 | 26.7 | 67.0 | 86.0 | 41.4 | 82.8 | 56.4 | 55.1 | 2.8 |
| 4096 | Quest | 24.6 | 32.1 | 48.7 | 40.4 | 27.1 | **19.7** | 32.8 | 23.5 | **27.2** | 72.5 | 86.3 | 40.3 | 82.7 | 57.7 | 54.9 | 2.8 |
| | SparQ | 12.3 | 31.7 | 47.9 | 26.1 | 25.8 | 9.0 | 26.5 | 20.7 | 27.1 | 71.0 | 69.6 | 37.5 | 71.5 | 57.4 | 49.1 | **3.9** |
| | HSA | **27.8** | 32.2 | **49.0** | 42.0 | 27.4 | 19.2 | 33.0 | 23.9 | 27.2 | 71.5 | 86.2 | **42.7** | 83.3 | **57.9** | **56.2** | 2.3 |
| | RocketKV | **27.8** | 32.1 | 48.7 | 41.8 | **27.5** | 19.2 | **33.4** | **24.5** | 27.1 | 71.5 | 86.2 | 42.5 | 82.8 | 56.0 | 54.0 | 2.9 |

Table 12: Results of RULER-8K for Mistral-7B-Ins-v0.2

| Token Budget | Method | single1 | single2 | single3 | multikey1 | multikey2 | multikey3 | multivalue | multiquery | vt | cwe | fwe | qa-1 | qa-2 | avg |
|---|---|---|---|---|---|---|---|---|---|---|---|---|---|---|---|
| N/A | Full-KV | 100.0 | 100.0 | 100.0 | 98.0 | 100.0 | 82.0 | 79.5 | 94.0 | 97.6 | 90.6 | 71.3 | 82.0 | 64.0 | 89.2 |
| 256 | Exact-TopK | 100.0 | 100.0 | 100.0 | 98.0 | 100.0 | 84.0 | 80.5 | 98.0 | 96.0 | 88.0 | 68.7 | 84.0 | 66.0 | 89.5 |
| | DuoAttention | 4.0 | 0.0 | 2.0 | 2.0 | 0.0 | 0.0 | 2.0 | 2.0 | 2.4 | 26.8 | 48.7 | 32.0 | 40.0 | 12.5 |
| | SnapKV | 16.0 | 24.0 | 0.0 | 4.0 | 32.0 | 0.0 | 0.0 | 0.0 | 2.4 | 5.8 | 28.0 | 68.0 | 52.0 | 17.9 |
| | Quest | 92.0 | 32.0 | 0.0 | 24.0 | 6.0 | 0.0 | 3.0 | 6.0 | 15.2 | 19.6 | 38.0 | 22.0 | 28.0 | 22.0 |
| | SparQ | 0.0 | 0.0 | 0.0 | 0.0 | 0.0 | 0.0 | 0.0 | 0.0 | 0.0 | 2.2 | 9.3 | 2.0 | 10.0 | 1.8 |
| | HSA | 98.0 | 82.0 | 50.0 | 84.0 | 74.0 | 12.0 | 42.5 | 60.5 | 88.4 | 30.4 | **73.3** | 84.0 | **66.0** | 65.0 |
| | RocketKV | **100.0** | **100.0** | **96.0** | **98.0** | **100.0** | 78.0 | **83.5** | **95.5** | **94.4** | **43.6** | 72.7 | **88.0** | 64.0 | **85.7** |
| 512 | Exact-TopK | 100.0 | 100.0 | 100.0 | 98.0 | 100.0 | 82.0 | 80.5 | 98.0 | 96.0 | 89.6 | 72.7 | 82.0 | 66.0 | 89.6 |
| | DuoAttention | 8.0 | 12.0 | 2.0 | 4.0 | 4.0 | 8.0 | 9.5 | 8.0 | 6.0 | 2.4 | 69.3 | 26.0 | 42.0 | 15.5 |
| | SnapKV | 82.0 | 56.0 | 0.0 | 30.0 | 56.0 | 0.0 | 1.0 | 1.0 | 18.0 | 10.4 | 72.7 | 78.0 | 56.0 | 35.5 |
| | Quest | 94.0 | 76.0 | 2.0 | 92.0 | 52.0 | 0.0 | 53.5 | 59.5 | 76.4 | 29.8 | 65.3 | 64.0 | 54.0 | 55.3 |
| | SparQ | 0.0 | 0.0 | 0.0 | 0.0 | 0.0 | 6.0 | 0.0 | 0.0 | 0.0 | 2.6 | 67.3 | 0.0 | 32.0 | 8.3 |
| | HSA | **100.0** | 98.0 | **96.0** | **98.0** | 96.0 | 80.0 | 75.5 | 93.0 | 95.2 | 58.0 | 71.3 | **84.0** | **66.0** | 85.5 |
| | RocketKV | **100.0** | **100.0** | **96.0** | **98.0** | **100.0** | **84.0** | **87.5** | **95.5** | **96.4** | **60.6** | **74.0** | **84.0** | 64.0 | **87.7** |
| 1024 | Exact-TopK | 100.0 | 100.0 | 100.0 | 98.0 | 100.0 | 82.0 | 80.0 | 96.0 | 96.8 | 90.8 | 73.3 | 80.0 | 66.0 | 89.5 |
| | DuoAttention | 12.0 | 12.0 | 2.0 | 4.0 | 14.0 | 10.0 | 9.5 | 8.0 | 9.2 | 14.2 | 72.0 | 42.0 | 48.0 | 19.8 |
| | SnapKV | 96.0 | 70.0 | 0.0 | 60.0 | 78.0 | 0.0 | 3.5 | 11.0 | 86.4 | 15.0 | **75.3** | 82.0 | 58.0 | 48.9 |
| | Quest | 92.0 | 84.0 | 26.0 | 94.0 | 76.0 | 22.0 | 74.5 | 78.0 | 89.6 | 51.8 | 70.0 | **84.0** | 64.0 | 69.7 |
| | SparQ | 0.0 | 62.0 | 6.0 | 72.0 | 0.0 | 72.0 | 70.0 | 61.0 | 0.0 | 8.6 | 68.0 | 30.0 | 44.0 | 38.0 |
| | HSA | **100.0** | **100.0** | **100.0** | 98.0 | **100.0** | **84.0** | 79.5 | 98.0 | 96.4 | **80.0** | 68.7 | 82.0 | 66.0 | 88.7 |
| | RocketKV | **100.0** | **100.0** | 98.0 | **98.0** | **100.0** | 78.0 | **89.5** | **98.5** | **97.6** | 79.2 | 72.0 | **84.0** | **68.0** | **89.4** |
| 2048 | Exact-TopK | 100.0 | 100.0 | 100.0 | 98.0 | 100.0 | 82.0 | 79.5 | 95.5 | 97.6 | 91.2 | 74.0 | 80.0 | 66.0 | 89.5 |
| | DuoAttention | 38.0 | 40.0 | 30.0 | 34.0 | 60.0 | 24.0 | 15.0 | 22.0 | 24.0 | 48.0 | 70.0 | 62.0 | 62.0 | 40.7 |
| | SnapKV | **100.0** | 90.0 | 0.0 | 82.0 | 88.0 | 0.0 | 8.5 | 30.0 | 95.2 | 29.6 | **74.0** | 80.0 | 64.0 | 57.0 |
| | Quest | **100.0** | 96.0 | 98.0 | 94.0 | 98.0 | 80.0 | 80.5 | 95.5 | 97.2 | 74.2 | 70.7 | 82.0 | **66.0** | 87.1 |
| | SparQ | 96.0 | 90.0 | **100.0** | 96.0 | **100.0** | **84.0** | 78.5 | 95.0 | 81.6 | **90.0** | 70.7 | 82.0 | **66.0** | 86.9 |
| | HSA | **100.0** | **100.0** | **100.0** | **98.0** | **100.0** | 82.0 | 80.5 | 96.0 | 96.4 | 88.4 | 69.3 | 80.0 | **66.0** | 89.0 |
| | RocketKV | **100.0** | **100.0** | 98.0 | **98.0** | **100.0** | 82.0 | **91.5** | **98.0** | **98.8** | 86.2 | 73.3 | **84.0** | **66.0** | **90.4** |
| 4096 | Exact-TopK | 100.0 | 100.0 | 100.0 | 98.0 | 100.0 | 82.0 | 79.5 | 94.5 | 97.6 | 90.8 | 73.3 | 82.0 | 64.0 | 89.4 |
| | DuoAttention | **100.0** | 98.0 | **100.0** | 96.0 | **100.0** | 84.0 | 71.0 | 90.5 | 96.0 | 85.4 | 70.0 | **86.0** | 66.0 | 87.9 |
| | SnapKV | **100.0** | 96.0 | 2.0 | 94.0 | 92.0 | 12.0 | 27.5 | 52.5 | 98.0 | 57.8 | 73.3 | 80.0 | 66.0 | 65.5 |
| | Quest | **100.0** | **100.0** | **100.0** | 98.0 | **100.0** | 84.0 | 80.0 | 96.0 | 96.8 | 87.8 | 71.3 | 82.0 | **66.0** | 89.4 |
| | SparQ | **100.0** | **100.0** | **100.0** | 98.0 | **100.0** | 82.0 | 79.0 | 95.5 | 97.6 | 89.8 | 72.7 | 82.0 | 64.0 | 89.3 |
| | HSA | **100.0** | **100.0** | **100.0** | 98.0 | **100.0** | 82.0 | 79.5 | 96.0 | 97.6 | **90.6** | **74.0** | 80.0 | 64.0 | 89.4 |
| | RocketKV | **100.0** | **100.0** | 94.0 | **98.0** | **100.0** | 82.0 | **91.0** | **98.0** | **99.2** | 90.4 | 73.3 | 84.0 | **66.0** | **90.5** |

Table 13: Results of RULER-16K for Mistral-7B-Ins-v0.2

| Token Budget | Method | single1 | single2 | single3 | multikey1 | multikey2 | multikey3 | multivalue | multiquery | vt | cwe | fwe | qa-1 | qa-2 | avg |
|---|---|---|---|---|---|---|---|---|---|---|---|---|---|---|---|
| N/A | Full-KV | 100.0 | 100.0 | 100.0 | 86.0 | 90.0 | 74.0 | 84.0 | 85.5 | 87.6 | 74.2 | 86.7 | 78.0 | 54.0 | 84.6 |
| 256 | Exact-TopK | 100.0 | 100.0 | 100.0 | 84.0 | 92.0 | 82.0 | 87.0 | 80.0 | 85.2 | 70.6 | 76.7 | 80.0 | 58.0 | 84.3 |
| | DuoAttention | 2.0 | 0.0 | 2.0 | 2.0 | 0.0 | 0.0 | 2.5 | 3.5 | 0.8 | 12.6 | 48.7 | 30.0 | 36.0 | 10.8 |
| | SnapKV | 16.0 | 14.0 | 0.0 | 6.0 | 8.0 | 0.0 | 0.0 | 0.0 | 2.0 | 1.6 | 20.0 | 56.0 | 48.0 | 13.2 |
| | Quest | 0.0 | 2.0 | 0.0 | 0.0 | 0.0 | 0.0 | 0.0 | 0.0 | 0.0 | 3.6 | 18.7 | 8.0 | 12.0 | 3.4 |
| | SparQ | 0.0 | 0.0 | 0.0 | 0.0 | 0.0 | 0.0 | 0.0 | 0.0 | 0.0 | 0.4 | 0.0 | 2.0 | 0.0 | 0.2 |
| | HSA | 96.0 | 52.0 | 2.0 | 30.0 | 24.0 | 0.0 | 16.0 | 10.0 | 72.0 | 16.0 | 70.7 | 74.0 | **54.0** | 39.7 |
| | RocketKV | **100.0** | **100.0** | **98.0** | **86.0** | **92.0** | **36.0** | 82.5 | 92.0 | 84.4 | 33.2 | **82.0** | 76.0 | 52.0 | **78.0** |
| 512 | Exact-TopK | 100.0 | 100.0 | 100.0 | 84.0 | 92.0 | 80.0 | 85.0 | 84.0 | 86.0 | 72.6 | 79.3 | 80.0 | 58.0 | 84.7 |
| | DuoAttention | 2.0 | 2.0 | 4.0 | 2.0 | 0.0 | 0.0 | 6.5 | 4.5 | 2.8 | 0.4 | 80.7 | 32.0 | 32.0 | 13.0 |
| | SnapKV | 84.0 | 44.0 | 0.0 | 22.0 | 22.0 | 0.0 | 0.0 | 0.5 | 16.0 | 5.0 | 63.3 | 60.0 | 46.0 | 27.9 |
| | Quest | **100.0** | 72.0 | 4.0 | 56.0 | 12.0 | 0.0 | 18.5 | 16.5 | 41.2 | 13.4 | 62.7 | 22.0 | 36.0 | 34.9 |
| | SparQ | 0.0 | 0.0 | 0.0 | 0.0 | 0.0 | 0.0 | 0.0 | 0.0 | 0.0 | 1.2 | 2.7 | 0.0 | 6.0 | 0.8 |
| | HSA | **100.0** | 94.0 | 86.0 | 80.0 | 76.0 | 18.0 | 47.5 | 59.0 | 81.6 | 34.4 | 75.3 | **80.0** | **58.0** | 68.4 |
| | RocketKV | **100.0** | **100.0** | **98.0** | 82.0 | 86.0 | 42.0 | 88.5 | 95.5 | 85.6 | 47.0 | 81.3 | **80.0** | 52.0 | 79.8 |
| 1024 | Exact-TopK | 100.0 | 100.0 | 100.0 | 84.0 | 92.0 | 78.0 | 83.0 | 85.5 | 88.4 | 74.4 | 84.0 | 78.0 | 58.0 | 85.0 |
| | DuoAttention | 6.0 | 8.0 | 4.0 | 2.0 | 0.0 | 0.0 | 8.0 | 5.0 | 4.4 | 0.2 | 80.7 | 38.0 | 46.0 | 15.6 |
| | SnapKV | 94.0 | 62.0 | 0.0 | 40.0 | 40.0 | 0.0 | 5.5 | 9.0 | 78.0 | 11.8 | 78.0 | 72.0 | 48.0 | 41.4 |
| | Quest | 84.0 | 94.0 | 10.0 | 82.0 | 44.0 | 0.0 | 61.5 | 54.5 | 73.6 | 28.8 | 70.0 | 68.0 | **56.0** | 55.9 |
| | SparQ | 0.0 | 0.0 | 0.0 | 0.0 | 0.0 | 0.0 | 0.0 | 0.0 | 0.0 | 6.6 | 60.0 | 0.0 | 2.0 | 5.3 |
| | HSA | **100.0** | **100.0** | 98.0 | **86.0** | 90.0 | 72.0 | 76.5 | 77.5 | 82.4 | 57.8 | 78.0 | **78.0** | 56.0 | 80.9 |
| | RocketKV | **100.0** | **100.0** | **100.0** | 82.0 | **90.0** | 58.0 | **90.5** | **97.5** | **88.0** | 62.0 | **84.0** | 76.0 | 56.0 | **83.4** |
| 2048 | Exact-TopK | 100.0 | 100.0 | 100.0 | 84.0 | 92.0 | 76.0 | 83.0 | 85.5 | 88.8 | 74.2 | 86.7 | 80.0 | 58.0 | 85.2 |
| | DuoAttention | 10.0 | 8.0 | 6.0 | 6.0 | 6.0 | 0.0 | 8.0 | 6.0 | 4.8 | 2.2 | 79.3 | 40.0 | 48.0 | 17.3 |
| | SnapKV | 98.0 | 76.0 | 0.0 | 54.0 | 46.0 | 0.0 | 10.5 | 21.0 | 85.6 | 22.2 | 81.3 | 70.0 | 52.0 | 47.4 |
| | Quest | 92.0 | 88.0 | 44.0 | 76.0 | 72.0 | 10.0 | 68.0 | 73.0 | 84.0 | 51.6 | 74.7 | **80.0** | 58.0 | 67.0 |
| | SparQ | 0.0 | 6.0 | 0.0 | 16.0 | 0.0 | 4.0 | 22.0 | 6.0 | 0.4 | 12.8 | 79.3 | 4.0 | 20.0 | 13.1 |
| | HSA | **100.0** | **100.0** | **100.0** | 84.0 | **92.0** | **76.0** | 82.5 | 82.0 | 85.2 | 70.8 | 78.0 | 76.0 | **58.0** | 83.4 |
| | RocketKV | **100.0** | **100.0** | 98.0 | 82.0 | 88.0 | 64.0 | **91.5** | **96.5** | **87.2** | **72.0** | **84.7** | 74.0 | 56.0 | **84.1** |
| 4096 | Exact-TopK | 100.0 | 100.0 | 100.0 | 84.0 | 92.0 | 74.0 | 83.0 | 86.0 | 88.0 | 75.2 | 86.7 | 80.0 | 56.0 | 85.0 |
| | DuoAttention | 48.0 | 46.0 | 40.0 | 32.0 | 48.0 | 14.0 | 24.0 | 20.0 | 22.8 | 16.6 | 80.7 | 62.0 | **58.0** | 39.4 |
| | SnapKV | **100.0** | 92.0 | 2.0 | 74.0 | 60.0 | 0.0 | 20.5 | 42.0 | **92.0** | 37.6 | 81.3 | 72.0 | 54.0 | 56.0 |
| | Quest | **100.0** | **100.0** | 96.0 | **86.0** | **92.0** | **82.0** | 82.5 | 83.0 | 86.0 | 65.8 | 80.7 | **80.0** | **58.0** | 84.0 |
| | SparQ | 82.0 | 86.0 | 84.0 | 82.0 | 80.0 | 78.0 | 67.5 | 64.5 | 31.2 | 73.2 | 83.3 | 76.0 | 52.0 | 72.3 |
| | HSA | **100.0** | **100.0** | **100.0** | 84.0 | **92.0** | 78.0 | 84.0 | 86.5 | 84.4 | 74.6 | 81.3 | 78.0 | **58.0** | 84.7 |
| | RocketKV | **100.0** | **100.0** | **100.0** | 82.0 | 90.0 | 70.0 | **94.0** | **98.0** | 89.6 | **76.4** | **86.7** | 78.0 | 56.0 | **86.2** |

Table 14: Results of RULER-24K for Mistral-7B-Ins-v0.2

| Token Budget | Method | single1 | single2 | single3 | multikey1 | multikey2 | multikey3 | multivalue | multiquery | vt | cwe | fwe | qa-1 | qa-2 | avg |
|---|---|---|---|---|---|---|---|---|---|---|---|---|---|---|---|
| N/A | Full-KV | 100.0 | 100.0 | 92.0 | 86.0 | 94.0 | 78.0 | 77.5 | 83.5 | 93.2 | 40.6 | 86.7 | 68.0 | 58.0 | 81.3 |
| 256 | Exact-TopK | 100.0 | 98.0 | 88.0 | 90.0 | 92.0 | 78.0 | 78.5 | 79.0 | 92.8 | 62.4 | 80.0 | 70.0 | 58.0 | 82.1 |
| | DuoAttention | 2.0 | 0.0 | 2.0 | 6.0 | 2.0 | 0.0 | 1.5 | 3.0 | 0.8 | 1.2 | 56.7 | 24.0 | 34.0 | 10.2 |
| | SnapKV | 32.0 | 14.0 | 0.0 | 4.0 | 8.0 | 0.0 | 0.0 | 0.0 | 2.8 | 2.6 | 22.0 | 48.0 | 40.0 | 13.3 |
| | Quest | 0.0 | 2.0 | 0.0 | 0.0 | 0.0 | 0.0 | 0.0 | 0.0 | 0.0 | 1.8 | 15.3 | 2.0 | 2.0 | 1.8 |
| | SparQ | 0.0 | 0.0 | 0.0 | 0.0 | 0.0 | 0.0 | 0.0 | 0.0 | 0.0 | 0.0 | 0.0 | 6.0 | 0.0 | 0.5 |
| | HSA | 90.0 | 34.0 | 0.0 | 24.0 | 0.0 | 0.0 | 8.0 | 7.5 | 73.6 | 6.4 | 56.0 | 52.0 | 44.0 | 30.4 |
| | RocketKV | **100.0** | **100.0** | **94.0** | **86.0** | **96.0** | **44.0** | **78.0** | **87.0** | **93.2** | 12.2 | **69.3** | **70.0** | **54.0** | **75.7** |
| 512 | Exact-TopK | 100.0 | 100.0 | 90.0 | 88.0 | 92.0 | 80.0 | 77.0 | 81.0 | 93.2 | 59.0 | 86.0 | 70.0 | 58.0 | 82.6 |
| | DuoAttention | 4.0 | 4.0 | 4.0 | 6.0 | 6.0 | 2.0 | 4.5 | 4.5 | 1.6 | 0.4 | 68.7 | 26.0 | 28.0 | 12.3 |
| | SnapKV | 90.0 | 32.0 | 0.0 | 16.0 | 22.0 | 0.0 | 0.5 | 1.0 | 11.2 | 3.4 | 56.0 | 56.0 | 44.0 | 25.5 |
| | Quest | **100.0** | 12.0 | 32.0 | 8.0 | 6.0 | 0.0 | 2.0 | 7.0 | 7.2 | 3.0 | 56.7 | 8.0 | 16.0 | 19.8 |
| | SparQ | 0.0 | 0.0 | 0.0 | 0.0 | 0.0 | 0.0 | 0.0 | 0.0 | 0.0 | 0.6 | 0.0 | 2.0 | 0.0 | 0.2 |
| | HSA | **100.0** | 74.0 | 48.0 | 54.0 | 42.0 | 0.0 | 33.0 | 33.5 | 86.0 | 18.0 | 66.7 | 68.0 | **52.0** | 51.9 |
| | RocketKV | **100.0** | **100.0** | **96.0** | **82.0** | **94.0** | **50.0** | **80.5** | **91.5** | **94.4** | **26.2** | **72.7** | **72.0** | **52.0** | **77.8** |
| 1024 | Exact-TopK | 100.0 | 100.0 | 90.0 | 86.0 | 92.0 | 80.0 | 78.0 | 84.0 | 93.6 | 55.4 | 86.7 | 70.0 | 58.0 | 82.6 |
| | DuoAttention | 4.0 | 4.0 | 4.0 | 6.0 | 8.0 | 2.0 | 4.5 | 4.5 | 2.0 | 0.6 | 70.0 | 28.0 | 32.0 | 13.0 |
| | SnapKV | 96.0 | 60.0 | 0.0 | 36.0 | 32.0 | 0.0 | 3.0 | 8.0 | 76.4 | 7.8 | 66.0 | 62.0 | 46.0 | 37.9 |
| | Quest | **100.0** | 82.0 | 2.0 | 78.0 | 20.0 | 0.0 | 32.0 | 34.5 | 45.2 | 9.6 | 66.0 | 36.0 | 32.0 | 41.3 |
| | SparQ | 0.0 | 0.0 | 0.0 | 0.0 | 0.0 | 0.0 | 0.0 | 0.0 | 0.0 | 2.2 | 1.3 | 0.0 | 4.0 | 0.6 |
| | HSA | **100.0** | 98.0 | 86.0 | 80.0 | 72.0 | 36.0 | 62.0 | 71.5 | 93.2 | 36.4 | 72.7 | **72.0** | **56.0** | 72.0 |
| | RocketKV | **100.0** | **100.0** | **98.0** | **84.0** | **92.0** | **48.0** | **84.0** | **93.5** | **97.2** | **41.6** | **76.7** | 66.0 | 54.0 | **79.6** |
| 2048 | Exact-TopK | 100.0 | 100.0 | 90.0 | 86.0 | 94.0 | 80.0 | 78.0 | 83.0 | 93.6 | 50.2 | 88.7 | 70.0 | 58.0 | 82.4 |
| | DuoAttention | 4.0 | 4.0 | 4.0 | 6.0 | 10.0 | 2.0 | 4.5 | 4.5 | 2.4 | 0.4 | 69.3 | 26.0 | 30.0 | 12.9 |
| | SnapKV | 96.0 | 80.0 | 0.0 | 56.0 | 40.0 | 2.0 | 10.0 | 20.5 | 96.8 | 14.4 | 70.7 | 62.0 | 46.0 | 45.7 |
| | Quest | 96.0 | 88.0 | 2.0 | 80.0 | 64.0 | 0.0 | 65.0 | 66.0 | 88.0 | 18.0 | 75.3 | 62.0 | 50.0 | 58.0 |
| | SparQ | 0.0 | 0.0 | 0.0 | 0.0 | 0.0 | 34.0 | 0.0 | 0.0 | 0.0 | 27.2 | 80.0 | 2.0 | 4.0 | 11.3 |
| | HSA | **100.0** | 98.0 | 88.0 | **88.0** | 90.0 | **68.0** | 73.5 | 80.5 | 94.8 | **53.6** | 74.7 | **68.0** | **60.0** | 79.8 |
| | RocketKV | **100.0** | **100.0** | **98.0** | 84.0 | **94.0** | 62.0 | **87.0** | **94.0** | **98.4** | **53.6** | **81.3** | 62.0 | 54.0 | **82.2** |
| 4096 | Exact-TopK | 100.0 | 100.0 | 90.0 | 86.0 | 94.0 | 78.0 | 78.0 | 84.0 | 93.6 | 41.2 | 88.7 | 70.0 | 58.0 | 81.7 |
| | DuoAttention | 24.0 | 18.0 | 10.0 | 18.0 | 20.0 | 2.0 | 6.0 | 6.0 | 6.4 | 0.8 | 71.3 | 52.0 | 44.0 | 21.4 |
| | SnapKV | **100.0** | 84.0 | 0.0 | 58.0 | 52.0 | 2.0 | 19.5 | 40.5 | 96.8 | 22.2 | 75.3 | 68.0 | 48.0 | 51.3 |
| | Quest | 90.0 | 86.0 | 72.0 | 72.0 | 72.0 | 38.0 | 71.5 | 75.0 | 90.4 | 43.4 | 80.0 | **70.0** | 56.0 | 70.5 |
| | SparQ | 0.0 | 6.0 | 0.0 | 4.0 | 0.0 | 8.0 | 2.0 | 10.5 | 0.0 | 6.2 | **86.0** | 0.0 | 18.0 | 10.8 |
| | HSA | **100.0** | 98.0 | 90.0 | **88.0** | **92.0** | **78.0** | 73.5 | 83.0 | 94.8 | **62.0** | 82.7 | **70.0** | **58.0** | 82.3 |
| | RocketKV | **100.0** | **100.0** | **98.0** | 82.0 | **92.0** | 76.0 | **89.5** | **94.5** | **97.2** | 48.2 | 84.7 | 60.0 | 56.0 | **82.9** |

Table 15: Results of RULER-32K for Mistral-7B-Ins-v0.2

| Token Budget | Method | single1 | single2 | single3 | multikey1 | multikey2 | multikey3 | multivalue | multiquery | vt | cwe | fwe | qa-1 | qa-2 | avg |
|---|---|---|---|---|---|---|---|---|---|---|---|---|---|---|---|
| N/A | Full-KV | 100.0 | 98.0 | 86.0 | 82.0 | 88.0 | 68.0 | 83.0 | 81.0 | 90.0 | 29.6 | 94.7 | 70.0 | 48.0 | 78.3 |
| 256 | Exact-TopK | 100.0 | 92.0 | 68.0 | 78.0 | 80.0 | 60.0 | 78.0 | 58.5 | 91.2 | 40.4 | 86.0 | 66.0 | 48.0 | 72.8 |
| | DuoAttention | 0.0 | 2.0 | 4.0 | 4.0 | 0.0 | 0.0 | 1.0 | 3.0 | 0.0 | 2.0 | 34.0 | 24.0 | 30.0 | 8.0 |
| | SnapKV | 36.0 | 6.0 | 0.0 | 6.0 | 4.0 | 0.0 | 0.0 | 0.0 | 0.0 | 0.2 | 8.0 | 44.0 | 38.0 | 10.9 |
| | Quest | 0.0 | 0.0 | 0.0 | 0.0 | 0.0 | 0.0 | 0.0 | 0.0 | 0.0 | 0.6 | 22.0 | 0.0 | 2.0 | 1.9 |
| | SparQ | 0.0 | 0.0 | 0.0 | 0.0 | 0.0 | 0.0 | 0.0 | 0.0 | 0.0 | 0.0 | 0.0 | 2.0 | 0.0 | 0.2 |
| | HSA | 76.0 | 14.0 | 0.0 | 4.0 | 2.0 | 0.0 | 6.0 | 3.5 | 55.6 | 2.2 | 57.3 | 26.0 | 32.0 | 21.4 |
| | RocketKV | **100.0** | **100.0** | **86.0** | **84.0** | **92.0** | **20.0** | **78.5** | **83.0** | **81.6** | **7.2** | **80.7** | **62.0** | **48.0** | **71.0** |
| 512 | Exact-TopK | 100.0 | 94.0 | 68.0 | 78.0 | 86.0 | 64.0 | 83.0 | 60.5 | 93.2 | 37.4 | 86.0 | 66.0 | 46.0 | 74.1 |
| | DuoAttention | 2.0 | 6.0 | 6.0 | 4.0 | 0.0 | 2.0 | 3.5 | 4.0 | 2.8 | 0.8 | 75.3 | 24.0 | 30.0 | 12.3 |
| | SnapKV | 98.0 | 26.0 | 0.0 | 10.0 | 8.0 | 0.0 | 0.0 | 0.0 | 6.8 | 1.6 | 55.3 | 50.0 | 38.0 | 22.6 |
| | Quest | 88.0 | 2.0 | 4.0 | 2.0 | 0.0 | 0.0 | 1.5 | 1.0 | 0.0 | 4.0 | 64.0 | 8.0 | 16.0 | 14.7 |
| | SparQ | 0.0 | 0.0 | 0.0 | 0.0 | 0.0 | 0.0 | 0.0 | 0.0 | 0.0 | 0.4 | 0.0 | 0.0 | 2.0 | 0.2 |
| | HSA | 96.0 | 28.0 | 4.0 | 28.0 | 26.0 | 0.0 | 16.5 | 12.5 | 77.2 | 9.2 | 77.3 | **72.0** | 44.0 | 37.7 |
| | RocketKV | **100.0** | **100.0** | **86.0** | **82.0** | **90.0** | **32.0** | **84.0** | **87.5** | **93.6** | **12.4** | **84.7** | 70.0 | **46.0** | **74.5** |
| 1024 | Exact-TopK | 100.0 | 96.0 | 72.0 | 80.0 | 90.0 | 64.0 | 81.5 | 66.0 | 92.4 | 37.0 | 91.3 | 68.0 | 50.0 | 76.0 |
| | DuoAttention | 2.0 | 6.0 | 6.0 | 4.0 | 0.0 | 2.0 | 3.5 | 4.5 | 3.6 | 0.6 | 80.7 | 20.0 | 32.0 | 12.7 |
| | SnapKV | **100.0** | 58.0 | 0.0 | 22.0 | 14.0 | 0.0 | 2.5 | 3.5 | 56.0 | 2.6 | 72.0 | 58.0 | 40.0 | 33.0 |
| | Quest | 94.0 | 74.0 | 2.0 | 44.0 | 18.0 | 0.0 | 15.5 | 19.0 | 69.6 | 5.6 | 77.3 | 28.0 | 32.0 | 36.8 |
| | SparQ | 0.0 | 0.0 | 0.0 | 0.0 | 0.0 | 0.0 | 0.0 | 0.0 | 0.0 | 0.6 | 0.0 | 0.0 | 8.0 | 0.7 |
| | HSA | 98.0 | 70.0 | 56.0 | 56.0 | 50.0 | 10.0 | 46.5 | 38.0 | 85.2 | **24.8** | 78.7 | **74.0** | 48.0 | 56.6 |
| | RocketKV | **100.0** | **100.0** | **86.0** | 80.0 | **92.0** | 28.0 | **86.0** | **90.0** | **94.8** | 23.0 | 82.7 | 68.0 | 48.0 | 75.3 |
| 2048 | Exact-TopK | 100.0 | 98.0 | 76.0 | 82.0 | 90.0 | 64.0 | 84.0 | 73.5 | 90.8 | 34.0 | 92.7 | 68.0 | 48.0 | 77.0 |
| | DuoAttention | 2.0 | 6.0 | 6.0 | 4.0 | 0.0 | 2.0 | 3.5 | 5.0 | 3.6 | 0.8 | 81.3 | 28.0 | 30.0 | 13.2 |
| | SnapKV | **100.0** | 66.0 | 0.0 | 40.0 | 24.0 | 0.0 | 5.0 | 11.5 | 86.4 | 8.2 | 76.7 | 64.0 | 42.0 | 40.3 |
| | Quest | 98.0 | 94.0 | 2.0 | 70.0 | 32.0 | 0.0 | 49.5 | 46.0 | 91.2 | 7.0 | **84.7** | 52.0 | 40.0 | 51.3 |
| | SparQ | 0.0 | 0.0 | 0.0 | 0.0 | 0.0 | 0.0 | 0.0 | 0.0 | 0.0 | 4.4 | 67.3 | 0.0 | 4.0 | 5.8 |
| | HSA | **100.0** | 96.0 | 76.0 | 72.0 | 72.0 | 40.0 | 71.5 | 59.5 | 90.8 | **44.2** | 82.0 | **68.0** | **50.0** | 70.9 |
| | RocketKV | **100.0** | **100.0** | **86.0** | **86.0** | **90.0** | **46.0** | **85.0** | **91.0** | **96.4** | 35.6 | 84.0 | 64.0 | 48.0 | **77.8** |
| 4096 | Exact-TopK | 100.0 | 98.0 | 76.0 | 82.0 | 92.0 | 64.0 | 82.5 | 77.0 | 90.4 | 34.8 | 93.3 | 68.0 | 48.0 | 77.4 |
| | DuoAttention | 10.0 | 8.0 | 8.0 | 6.0 | 6.0 | 2.0 | 4.0 | 4.5 | 4.8 | 1.8 | 84.0 | 34.0 | 32.0 | 15.8 |
| | SnapKV | **100.0** | 74.0 | 0.0 | 66.0 | 32.0 | 0.0 | 12.0 | 22.0 | 92.8 | 16.2 | 80.0 | 62.0 | 46.0 | 46.4 |
| | Quest | 92.0 | 78.0 | 32.0 | 68.0 | 64.0 | 8.0 | 67.0 | 59.0 | 90.0 | 24.8 | **89.3** | 68.0 | 46.0 | 60.5 |
| | SparQ | 0.0 | 2.0 | 0.0 | 0.0 | 0.0 | 0.0 | 0.5 | 0.0 | 0.0 | 9.4 | 88.0 | 0.0 | 4.0 | 8.0 |
| | HSA | **100.0** | 98.0 | 76.0 | 78.0 | **84.0** | **60.0** | 74.0 | 68.5 | 94.4 | **43.4** | 85.3 | 66.0 | **48.0** | 75.0 |
| | RocketKV | **100.0** | **100.0** | **86.0** | **88.0** | **84.0** | 54.0 | **89.0** | **92.0** | **96.4** | 36.0 | 88.0 | 64.0 | 48.0 | **78.9** |

Table 16: Average results of LongBench and NIAH for LongChat-7B-v1.5

| Token Budget | Method | Single QA | Multi. QA | Summ. | Few-shot | Synthetic | Code | LB Avg. | NIAH |
|---|---|---|---|---|---|---|---|---|---|
| N/A | Full-KV | 31.1 | 24.1 | 26.7 | 57.7 | 15.2 | 57.0 | 37.5 | 96.3 |
| 256 | Exact-TopK | 30.5 | 23.5 | 26.5 | 53.9 | 14.8 | 58.3 | 36.6 | 96.7 |
| | DuoAttention | 14.7 | 15.8 | 14.4 | 29.8 | 2.4 | 48.5 | 21.7 | 20.7 |
| | SnapKV | 25.9 | 23.0 | 20.8 | 45.1 | **15.2** | 51.1 | 31.7 | 43.3 |
| | Quest | 9.2 | 7.2 | 7.8 | 11.4 | 0.9 | 34.4 | 11.8 | 15.3 |
| | SparQ | 18.9 | 11.2 | 15.2 | 16.8 | 1.5 | 32.4 | 16.9 | 21.0 |
| | HSA | 28.2 | 23.7 | **25.1** | 49.4 | 13.0 | 54.4 | 34.1 | 58.0 |
| | RocketKV | **30.1** | **24.5** | 23.0 | **51.8** | 13.8 | **56.5** | **35.1** | **96.3** |
| 512 | Exact-TopK | 30.8 | 23.9 | 26.7 | 54.4 | 14.5 | 57.6 | 36.8 | 96.3 |
| | DuoAttention | 18.1 | 20.4 | 19.5 | 37.7 | 3.2 | 49.4 | 26.1 | 23.7 |
| | SnapKV | 28.1 | 23.8 | 22.2 | 51.2 | **14.5** | 53.3 | 34.1 | 66.7 |
| | Quest | 20.7 | 15.6 | 16.8 | 32.5 | 7.6 | 46.4 | 24.2 | 48.0 |
| | SparQ | 26.5 | 21.4 | 23.6 | 41.5 | 13.5 | 49.1 | 30.9 | 38.0 |
| | HSA | 30.6 | **24.8** | **26.5** | 52.3 | 14.2 | **57.2** | 36.3 | 90.7 |
| | RocketKV | **30.9** | 24.1 | 24.5 | **52.9** | **14.5** | 56.4 | 35.9 | **96.7** |
| 1024 | Exact-TopK | 30.7 | 23.9 | 26.8 | 54.3 | 14.2 | 57.4 | 36.7 | 96.3 |
| | DuoAttention | 20.8 | 20.7 | 21.0 | 42.0 | 4.0 | 53.6 | 28.5 | 28.0 |
| | SnapKV | 28.9 | 24.0 | 23.6 | 53.0 | 14.5 | 54.1 | 35.0 | 80.0 |
| | Quest | 28.2 | 23.0 | 23.6 | 47.9 | **17.2** | 53.3 | 33.8 | 77.0 |
| | SparQ | 29.7 | 22.9 | 26.4 | 53.3 | **17.2** | 55.9 | 36.1 | 76.7 |
| | HSA | 30.7 | 24.2 | **27.0** | 53.8 | 14.0 | **57.9** | **36.7** | 95.7 |
| | RocketKV | **31.0** | 24.3 | 25.6 | **54.5** | 14.8 | 55.3 | 36.4 | **96.3** |
| 2048 | Exact-TopK | 30.7 | 24.0 | 26.8 | 54.4 | 15.0 | 57.3 | 36.8 | 96.3 |
| | DuoAttention | 26.0 | 20.2 | 23.1 | 46.3 | 3.8 | 55.0 | 30.9 | 42.3 |
| | SnapKV | 30.2 | 23.8 | 24.9 | 54.0 | 14.5 | 54.4 | 35.8 | 85.0 |
| | Quest | 30.0 | 24.2 | 26.5 | 53.3 | **15.5** | 56.3 | 36.3 | 89.7 |
| | SparQ | 30.6 | 24.1 | **26.9** | 54.6 | 14.5 | 56.8 | 36.7 | 96.0 |
| | HSA | 30.7 | 24.1 | 26.8 | **54.6** | 14.5 | **57.3** | **36.8** | 96.3 |
| | RocketKV | **31.1** | 24.3 | 26.3 | **54.6** | 14.5 | 55.0 | 36.5 | 96.3 |
| 4096 | Exact-TopK | 31.0 | 24.0 | 26.6 | 54.4 | 15.5 | 57.2 | 36.9 | 96.3 |
| | DuoAttention | 28.6 | 22.4 | 25.4 | 50.5 | 10.5 | 55.2 | 34.1 | 67.3 |
| | SnapKV | **31.1** | **24.4** | 26.1 | **55.0** | 14.8 | 55.1 | 36.6 | 89.3 |
| | Quest | 30.7 | 24.2 | **27.0** | 54.3 | 14.2 | **57.3** | 36.8 | 96.3 |
| | SparQ | 30.8 | 23.9 | 26.7 | 54.6 | **15.0** | 57.2 | **36.8** | 94.3 |
| | HSA | 31.0 | 23.9 | 26.8 | 54.4 | 14.2 | **57.3** | **36.8** | 96.3 |
| | RocketKV | 30.7 | 24.3 | 26.8 | 54.3 | 14.2 | 55.2 | 36.5 | **96.3** |

Table 17: Individual results of LongBench for LongChat-7B-v1.5

| Token Budget | Method | narrativeqa | qasper | multifieldqa | hotpotqa | 2wikimqa | musique | gov-report | qmsum | multi-news | trec | triviaqa | samsum | pass-ret. | lcc | repobench-p | passagecnt |
|---|---|---|---|---|---|---|---|---|---|---|---|---|---|---|---|---|---|
| N/A | Full-KV | 20.8 | 29.4 | 43.1 | 33.0 | 24.4 | 14.7 | 30.8 | 22.8 | 26.7 | 66.5 | 84.0 | 22.5 | 30.5 | 54.7 | 59.2 | 0.0 |
| | Exact-TopK | 19.6 | 30.2 | 41.6 | 33.7 | 23.5 | 13.5 | 30.6 | 22.5 | 26.3 | 65.5 | 83.5 | 12.7 | 29.5 | 57.7 | 58.9 | 0.0 |
| | DuoAttention | 12.4 | 10.1 | 21.6 | 18.6 | 20.5 | 8.1 | 12.6 | 18.7 | 11.8 | 13.5 | 65.9 | 9.8 | 4.8 | 48.6 | 48.3 | 0.0 |
| | SnapKV | 17.5 | 23.1 | 37.0 | 32.4 | 23.6 | 13.0 | 19.7 | 21.3 | 21.4 | 44.0 | 83.2 | 8.1 | 29.0 | 51.1 | 51.2 | 1.5 |
| 256 | Quest | 1.7 | 13.8 | 12.0 | 8.3 | 12.4 | 0.9 | 2.9 | 4.1 | 16.5 | 18.0 | 14.7 | 1.5 | 1.1 | 43.1 | 25.6 | 0.8 |
| | SparQ | 2.5 | 27.3 | 26.8 | 12.9 | 19.4 | 1.2 | 10.4 | 9.8 | 25.3 | 22.5 | 25.4 | 2.5 | 2.5 | 44.8 | 20.0 | 0.5 |
| | HSA | 14.1 | **29.8** | 40.7 | **35.0** | 23.8 | 12.3 | **27.3** | 21.3 | **26.7** | 62.0 | 78.3 | 7.8 | 24.1 | **57.8** | 50.9 | **2.0** |
| | RocketKV | **18.6** | 29.6 | **42.1** | 34.9 | **24.0** | **14.6** | 24.5 | 21.5 | 23.0 | 63.0 | **85.2** | 7.3 | 26.0 | 57.0 | **55.9** | 1.5 |
| | Exact-TopK | 20.1 | 30.5 | 41.9 | 33.9 | 23.8 | 14.0 | 30.6 | 22.9 | 26.6 | 66.0 | 84.4 | 12.9 | 29.0 | 56.4 | 58.8 | 0.0 |
| | DuoAttention | 13.1 | 15.9 | 25.4 | 28.0 | 24.1 | 9.2 | 19.4 | 19.3 | 19.7 | 31.5 | 72.2 | 9.5 | 5.5 | 50.6 | 48.3 | 1.0 |
| | SnapKV | 17.5 | 27.0 | 39.7 | 34.2 | 23.4 | 13.7 | 21.5 | 21.9 | 23.1 | 59.0 | **84.6** | **10.2** | 29.0 | 52.4 | 54.3 | 0.0 |
| 512 | Quest | 5.0 | 26.8 | 30.2 | 21.3 | 19.8 | 5.8 | 11.4 | 13.9 | 25.2 | 42.5 | 51.4 | 3.5 | 13.8 | 54.0 | 38.8 | **1.5** |
| | SparQ | 10.3 | 29.6 | 39.5 | 30.6 | 24.1 | 9.6 | 23.9 | 20.5 | **26.5** | 62.0 | 58.0 | 4.6 | 26.1 | 54.6 | 43.5 | 1.0 |
| | HSA | 19.1 | **29.8** | 42.8 | **35.1** | 24.3 | **15.1** | 30.8 | 22.4 | 26.4 | 64.5 | 83.6 | 8.8 | 28.0 | **57.3** | **57.2** | 0.5 |
| | RocketKV | **20.2** | **29.8** | 42.6 | 34.8 | 23.6 | 13.8 | 26.9 | 22.1 | 24.6 | **64.5** | 84.1 | 10.1 | 28.5 | 55.9 | 56.9 | 0.5 |
| | Exact-TopK | 20.5 | 29.7 | 42.0 | 33.5 | 24.2 | 14.2 | 30.9 | 23.2 | 26.4 | 65.5 | 84.4 | 13.1 | 28.5 | 55.7 | 59.1 | 0.0 |
| | DuoAttention | 13.5 | 18.1 | 30.7 | 28.2 | **24.8** | 9.2 | 20.2 | 19.9 | 22.9 | 43.5 | 72.8 | 9.7 | 6.5 | 56.6 | 50.6 | 1.5 |
| | SnapKV | 18.9 | 27.1 | 40.5 | **35.5** | 23.1 | 13.4 | 23.9 | 22.1 | 24.9 | 63.0 | 84.2 | **11.9** | 29.0 | 52.0 | 56.2 | 0.0 |
| 1024 | Quest | 13.2 | 29.5 | 42.0 | 33.1 | 24.4 | 11.5 | 23.6 | 21.4 | 25.9 | 63.0 | 76.5 | 4.1 | 32.0 | 55.9 | 50.8 | **2.5** |
| | SparQ | 17.8 | 29.6 | 41.8 | 32.6 | 23.3 | 12.7 | 30.3 | 22.7 | 26.3 | 66.5 | 83.2 | 10.3 | **33.0** | 55.6 | 56.2 | 1.5 |
| | HSA | **20.4** | **30.5** | 41.2 | 34.2 | 23.7 | **14.9** | 31.4 | **23.0** | 26.5 | 65.5 | **84.6** | 11.4 | 28.0 | **57.2** | **58.6** | 0.0 |
| | RocketKV | 20.2 | 29.4 | **43.3** | 34.6 | 23.8 | 14.3 | 28.9 | 22.5 | 25.5 | **67.5** | 84.1 | 11.8 | 29.5 | 54.1 | 56.4 | 0.0 |
| | Exact-TopK | 20.3 | 29.5 | 42.4 | 33.2 | 24.7 | 14.2 | 31.1 | 22.9 | 26.5 | 66.0 | 84.2 | 12.9 | 30.0 | 55.3 | 59.4 | 0.0 |
| | DuoAttention | 13.9 | 26.6 | 37.4 | 27.8 | **25.4** | 7.4 | 23.3 | 19.9 | 26.1 | 55.5 | 74.4 | 9.1 | 6.5 | **56.5** | 53.5 | **1.0** |
| | SnapKV | **20.7** | 28.4 | 41.5 | 33.5 | 24.1 | 13.9 | 26.6 | 22.6 | 25.6 | 64.5 | **84.7** | 12.8 | 29.0 | 52.6 | 56.2 | 0.0 |
| 2048 | Quest | 18.4 | **30.4** | 41.3 | **34.7** | 23.9 | 13.9 | 30.2 | 22.7 | **26.5** | 67.5 | 83.5 | 8.9 | 30.0 | 56.2 | 56.5 | **1.0** |
| | SparQ | 20.3 | 29.5 | 42.0 | 33.3 | 24.7 | **14.4** | **31.3** | **22.9** | **26.5** | 66.0 | 84.5 | **13.3** | 29.0 | 55.1 | 58.4 | 0.0 |
| | HSA | 20.3 | 29.9 | 42.0 | 34.3 | 24.2 | 13.8 | 31.2 | 22.6 | 26.4 | 67.0 | 84.4 | 12.4 | 29.0 | 55.5 | **59.1** | 0.0 |
| | RocketKV | 20.6 | 29.5 | **43.3** | 34.0 | 24.8 | 14.0 | 29.9 | 22.8 | 26.4 | 66.5 | 84.4 | 12.9 | 29.0 | 53.4 | 56.5 | 0.0 |
| | Exact-TopK | 21.0 | 29.4 | 42.6 | 33.0 | 24.3 | 14.6 | 30.7 | 22.6 | 26.5 | 66.5 | 84.0 | 12.8 | 31.0 | 55.1 | 59.4 | 0.0 |
| | DuoAttention | 14.2 | 29.6 | 42.1 | 32.2 | 23.9 | 11.1 | 28.6 | 21.1 | 26.5 | 64.0 | 78.3 | 9.1 | 20.5 | **55.7** | 54.6 | **0.5** |
| | SnapKV | 20.8 | 29.6 | 43.0 | **34.1** | 24.7 | 14.4 | 28.7 | 23.2 | 26.5 | **67.5** | 84.2 | **13.1** | 29.5 | 53.2 | 56.9 | 0.0 |
| 4096 | Quest | 20.5 | **29.8** | 41.9 | 33.9 | 24.8 | 14.0 | **31.4** | 22.8 | **26.7** | 67.0 | **84.3** | 11.6 | 28.5 | 55.4 | 59.2 | 0.0 |
| | SparQ | **21.0** | 29.0 | 42.5 | 33.2 | 24.2 | 14.3 | 30.7 | 22.8 | 26.6 | 66.5 | 84.2 | **13.1** | **30.0** | 55.1 | **59.4** | 0.0 |
| | HSA | 20.8 | 29.2 | **43.2** | 33.5 | 24.2 | 14.1 | 31.0 | 22.9 | 26.5 | 66.0 | 84.2 | 12.9 | 28.5 | 55.3 | 59.3 | 0.0 |
| | RocketKV | 20.3 | 29.2 | 42.8 | 33.5 | **24.9** | **14.5** | 30.5 | **23.3** | 26.5 | 66.0 | 84.2 | 12.8 | 28.5 | 53.5 | 56.9 | 0.0 |

Table 18: Results of RULER-8K for LongChat-7B-v1.5

| Token Budget | Method | single1 | single2 | single3 | multikey1 | multikey2 | multikey3 | multivalue | multiquery | vt | cwe | fwe | qa-1 | qa-2 | avg |
|---|---|---|---|---|---|---|---|---|---|---|---|---|---|---|---|
| N/A | Full-KV | 100.0 | 100.0 | 96.0 | 98.0 | 100.0 | 74.0 | 96.0 | 69.0 | 94.0 | 33.8 | 80.7 | 70.0 | 58.0 | 82.3 |
| 256 | Exact-TopK | 100.0 | 96.0 | 96.0 | 94.0 | 100.0 | 84.0 | 88.0 | 60.0 | 90.8 | 39.4 | 76.7 | 66.0 | 58.0 | 80.7 |
|  | DuoAttention | 4.0 | 0.0 | 0.0 | 2.0 | 0.0 | 0.0 | 2.0 | 2.0 | 1.2 | 1.6 | 3.3 | 28.0 | 24.0 | 5.2 |
|  | SnapKV | **100.0** | 78.0 | 0.0 | 40.0 | 6.0 | 0.0 | 8.5 | 2.0 | 54.0 | 15.0 | 46.0 | 64.0 | 48.0 | 35.5 |
|  | Quest | 52.0 | 56.0 | 0.0 | 28.0 | 0.0 | 0.0 | 4.5 | 7.0 | 30.8 | 12.4 | 46.0 | 6.0 | 18.0 | 20.1 |
|  | SparQ | 4.0 | 0.0 | 0.0 | 2.0 | 0.0 | 0.0 | 0.5 | 0.0 | 20.0 | 30.4 | 35.3 | 46.0 | 40.0 | 13.7 |
|  | HSA | 98.0 | 86.0 | 72.0 | 78.0 | 92.0 | **30.0** | 80.0 | 27.5 | **89.2** | 31.8 | **71.3** | **72.0** | **56.0** | 68.0 |
|  | RocketKV | **100.0** | **98.0** | **96.0** | **94.0** | **100.0** | 18.0 | **94.0** | **56.0** | 87.2 | **33.4** | 69.3 | 66.0 | **56.0** | **74.5** |
| 512 | Exact-TopK | 100.0 | 98.0 | 96.0 | 96.0 | 100.0 | 82.0 | 92.5 | 64.0 | 91.2 | 36.6 | 78.0 | 64.0 | 58.0 | 81.3 |
|  | DuoAttention | 8.0 | 12.0 | 2.0 | 4.0 | 4.0 | 0.0 | 8.5 | 3.0 | 0.0 | 0.4 | 52.7 | 30.0 | 26.0 | 11.6 |
|  | SnapKV | **100.0** | 86.0 | 2.0 | 78.0 | 18.0 | 0.0 | 35.5 | 9.5 | 89.2 | 25.4 | 56.0 | 70.0 | 50.0 | 47.7 |
|  | Quest | 98.0 | 74.0 | 2.0 | 36.0 | 68.0 | 0.0 | 45.5 | 21.0 | 83.6 | 30.4 | 71.3 | 38.0 | 40.0 | 46.8 |
|  | SparQ | 98.0 | 94.0 | 56.0 | 76.0 | 92.0 | 16.0 | 39.0 | 23.0 | 78.0 | **38.4** | 70.0 | **72.0** | 56.0 | 62.2 |
|  | HSA | **100.0** | 98.0 | 84.0 | 94.0 | **100.0** | **74.0** | 85.0 | 51.5 | 88.8 | 37.0 | 71.3 | 68.0 | 58.0 | 77.5 |
|  | RocketKV | **100.0** | **98.0** | **96.0** | **96.0** | **100.0** | 44.0 | **92.0** | **66.0** | 90.0 | 33.6 | **72.0** | 66.0 | **60.0** | **78.0** |
| 1024 | Exact-TopK | 100.0 | 100.0 | 96.0 | 98.0 | 100.0 | 80.0 | 97.0 | 68.0 | 93.2 | 34.4 | 82.0 | 62.0 | 56.0 | 82.0 |
|  | DuoAttention | 12.0 | 12.0 | 0.0 | 4.0 | 4.0 | 0.0 | 8.5 | 5.0 | 1.2 | 0.8 | 62.0 | 32.0 | 30.0 | 13.2 |
|  | SnapKV | **100.0** | 92.0 | 12.0 | 92.0 | 40.0 | 2.0 | 70.5 | 24.5 | 90.8 | 30.6 | 60.0 | 66.0 | **58.0** | 56.8 |
|  | Quest | **100.0** | 90.0 | 30.0 | 80.0 | 78.0 | 24.0 | 80.0 | 49.5 | **92.4** | 32.8 | 73.3 | 66.0 | 54.0 | 65.4 |
|  | SparQ | **100.0** | 98.0 | 94.0 | **98.0** | **100.0** | 66.0 | 93.0 | **67.0** | 91.2 | 35.2 | **79.3** | 66.0 | 56.0 | 80.3 |
|  | HSA | **100.0** | 98.0 | 96.0 | 96.0 | **100.0** | **86.0** | 91.0 | 62.0 | **92.4** | 37.0 | 74.7 | 66.0 | **58.0** | 81.3 |
|  | RocketKV | **100.0** | 98.0 | 96.0 | 96.0 | **100.0** | 74.0 | **93.0** | 63.5 | 91.2 | 34.2 | 77.3 | **68.0** | **58.0** | 80.7 |
| 2048 | Exact-TopK | 100.0 | 100.0 | 96.0 | 98.0 | 100.0 | 80.0 | 96.5 | 69.0 | 92.8 | 34.4 | 82.7 | 66.0 | 58.0 | 82.6 |
|  | DuoAttention | 48.0 | 76.0 | 18.0 | 78.0 | 66.0 | 26.0 | 61.5 | 28.5 | 14.8 | 1.6 | 76.7 | 60.0 | 48.0 | 46.4 |
|  | SnapKV | **100.0** | 98.0 | 64.0 | **98.0** | 54.0 | 6.0 | 80.0 | 53.5 | **92.8** | 34.6 | 64.0 | 64.0 | **58.0** | 66.7 |
|  | Quest | **100.0** | 98.0 | **96.0** | **98.0** | **100.0** | **84.0** | 96.0 | **70.0** | **92.8** | 34.2 | 74.0 | **68.0** | **58.0** | 82.2 |
|  | SparQ | **100.0** | **100.0** | **96.0** | **98.0** | **100.0** | 80.0 | 97.0 | 68.5 | **92.8** | 34.0 | **82.0** | 66.0 | **58.0** | **82.5** |
|  | HSA | **100.0** | **100.0** | **96.0** | **98.0** | **100.0** | 80.0 | **98.0** | 67.0 | 91.2 | **36.6** | 79.3 | 62.0 | **58.0** | 82.0 |
|  | RocketKV | **100.0** | 98.0 | **96.0** | **98.0** | **100.0** | 76.0 | 94.5 | 66.5 | **92.4** | 34.4 | 79.3 | 64.0 | **58.0** | 81.3 |
| 4096 | Exact-TopK | 100.0 | 100.0 | 96.0 | 98.0 | 100.0 | 74.0 | 96.5 | 70.0 | 93.6 | 33.8 | 82.7 | 70.0 | 58.0 | 82.5 |
|  | DuoAttention | 98.0 | **100.0** | 88.0 | 96.0 | **100.0** | 64.0 | **97.0** | **74.5** | 85.6 | **35.0** | 82.0 | 66.0 | 58.0 | 80.3 |
|  | SnapKV | **100.0** | **100.0** | 94.0 | **98.0** | 90.0 | 34.0 | 93.0 | 66.5 | **94.0** | 33.8 | 74.0 | 66.0 | **60.0** | 77.2 |
|  | Quest | **100.0** | **100.0** | **96.0** | **98.0** | **100.0** | 78.0 | 96.5 | 69.5 | 93.6 | 34.0 | 80.0 | 66.0 | 58.0 | 82.3 |
|  | SparQ | **100.0** | **100.0** | **96.0** | **98.0** | **100.0** | 76.0 | 96.5 | 69.5 | 93.2 | 33.8 | 82.0 | **70.0** | 58.0 | 82.5 |
|  | HSA | **100.0** | **100.0** | **96.0** | **98.0** | **100.0** | 80.0 | 96.5 | 69.5 | 92.8 | 34.4 | **82.7** | 66.0 | 58.0 | 82.6 |
|  | RocketKV | **100.0** | **100.0** | **96.0** | **98.0** | **100.0** | 80.0 | **97.0** | 70.0 | 93.2 | 33.6 | **82.7** | 66.0 | 58.0 | **82.7** |

Table 19: Results of RULER-16K for LongChat-7B-v1.5

| Token Budget | Method | single1 | single2 | single3 | multikey1 | multikey2 | multikey3 | multivalue | multiquery | vt | cwe | fwe | qa-1 | qa-2 | avg |
|---|---|---|---|---|---|---|---|---|---|---|---|---|---|---|---|
| N/A | Full-KV | 100.0 | 100.0 | 96.0 | 94.0 | 98.0 | 42.0 | 84.5 | 50.0 | 82.4 | 18.0 | 72.0 | 64.0 | 44.0 | 72.7 |
| 256 | Exact-TopK | 100.0 | 98.0 | 98.0 | 86.0 | 100.0 | 54.0 | 74.0 | 44.5 | 75.6 | 38.4 | 70.0 | 58.0 | 44.0 | 72.3 |
| | DuoAttention | 2.0 | 0.0 | 0.0 | 2.0 | 0.0 | 0.0 | 2.5 | 3.5 | 0.8 | 0.4 | 0.7 | 26.0 | 30.0 | 5.2 |
| | SnapKV | **100.0** | 50.0 | 0.0 | 30.0 | 0.0 | 0.0 | 1.5 | 1.5 | 48.8 | 2.0 | 40.7 | 50.0 | 38.0 | 27.9 |
| | Quest | 0.0 | 0.0 | 0.0 | 0.0 | 0.0 | 0.0 | 0.0 | 0.0 | 0.0 | 0.2 | 12.0 | 0.0 | 6.0 | 1.4 |
| | SparQ | 6.0 | 2.0 | 0.0 | 0.0 | 0.0 | 0.0 | 0.0 | 0.0 | 2.8 | 0.0 | 12.0 | 2.0 | 14.0 | 3.0 |
| | HSA | 96.0 | 86.0 | 24.0 | 38.0 | 22.0 | 0.0 | 36.5 | 11.5 | **76.8** | **15.2** | 39.3 | 50.0 | 40.0 | 41.2 |
| | RocketKV | **100.0** | **98.0** | **92.0** | **74.0** | **96.0** | **10.0** | **78.0** | **49.5** | 73.2 | 13.8 | **60.0** | **58.0** | **46.0** | **65.3** |
| 512 | Exact-TopK | 100.0 | 100.0 | 98.0 | 84.0 | 100.0 | 54.0 | 78.5 | 50.0 | 76.8 | 29.8 | 73.3 | 58.0 | 44.0 | 72.8 |
| | DuoAttention | 2.0 | 2.0 | 2.0 | 2.0 | 2.0 | 0.0 | 5.5 | 4.5 | 0.4 | 0.6 | **68.0** | 20.0 | 30.0 | 10.7 |
| | SnapKV | **100.0** | 86.0 | 0.0 | 54.0 | 4.0 | 0.0 | 10.5 | 4.5 | 75.2 | 5.0 | 47.3 | 56.0 | 44.0 | 37.4 |
| | Quest | 58.0 | 46.0 | 0.0 | 20.0 | 2.0 | 0.0 | 16.0 | 0.5 | 61.2 | 8.4 | 48.0 | 4.0 | 16.0 | 21.5 |
| | SparQ | 46.0 | 54.0 | 0.0 | 6.0 | 10.0 | 0.0 | 5.5 | 0.5 | 64.8 | **32.6** | 22.0 | 30.0 | 36.0 | 23.6 |
| | HSA | **100.0** | 98.0 | 90.0 | 68.0 | 62.0 | 8.0 | 72.0 | 47.5 | **78.8** | 19.6 | 52.0 | 60.0 | **46.0** | 61.5 |
| | RocketKV | **100.0** | **100.0** | **96.0** | **80.0** | **98.0** | **24.0** | **83.0** | **61.5** | 76.8 | 18.4 | 66.0 | **66.0** | **46.0** | **70.4** |
| 1024 | Exact-TopK | 100.0 | 100.0 | 96.0 | 90.0 | 100.0 | 52.0 | 82.0 | 49.5 | 77.6 | 26.4 | 69.3 | 64.0 | 44.0 | 73.1 |
| | DuoAttention | 6.0 | 8.0 | 2.0 | 2.0 | 2.0 | 0.0 | 7.0 | 4.5 | 1.2 | 0.6 | 65.3 | 28.0 | 34.0 | 12.4 |
| | SnapKV | **100.0** | 86.0 | 0.0 | 82.0 | 18.0 | 0.0 | 27.5 | 9.5 | 77.6 | 11.0 | 60.7 | 54.0 | 44.0 | 43.9 |
| | Quest | 94.0 | 98.0 | 8.0 | 74.0 | 42.0 | 0.0 | 55.5 | 19.0 | **80.0** | 26.0 | 58.7 | 36.0 | 34.0 | 48.1 |
| | SparQ | **100.0** | 96.0 | 88.0 | 86.0 | 90.0 | 28.0 | 66.0 | 42.0 | 79.2 | 27.6 | **68.0** | 58.0 | 38.0 | 66.7 |
| | HSA | **100.0** | 98.0 | **96.0** | 84.0 | 88.0 | **44.0** | 76.0 | 46.5 | 77.6 | **29.2** | 60.0 | 56.0 | **46.0** | 69.3 |
| | RocketKV | **100.0** | **100.0** | **96.0** | **90.0** | **100.0** | 40.0 | **81.5** | **54.0** | 78.4 | 22.0 | 65.3 | **64.0** | **46.0** | 72.1 |
| 2048 | Exact-TopK | 100.0 | 100.0 | 96.0 | 92.0 | 98.0 | 50.0 | 82.5 | 51.0 | 81.6 | 25.0 | 71.3 | 62.0 | 44.0 | 73.3 |
| | DuoAttention | 6.0 | 14.0 | 2.0 | 4.0 | 2.0 | 0.0 | 8.5 | 5.0 | 1.2 | 0.4 | 70.0 | 38.0 | 28.0 | 13.8 |
| | SnapKV | **100.0** | 90.0 | 14.0 | 84.0 | 24.0 | 0.0 | 43.0 | 20.0 | 78.0 | 16.2 | 64.0 | 60.0 | 44.0 | 49.0 |
| | Quest | **100.0** | 92.0 | 64.0 | 82.0 | 80.0 | 16.0 | 80.5 | 48.0 | 77.6 | 26.8 | 69.3 | 52.0 | **48.0** | 64.3 |
| | SparQ | **100.0** | **100.0** | 94.0 | 90.0 | **100.0** | 54.0 | **82.0** | 53.5 | 77.6 | 21.2 | **74.7** | **64.0** | 44.0 | 73.5 |
| | HSA | **100.0** | **100.0** | **98.0** | 88.0 | 98.0 | **56.0** | 81.5 | 52.0 | **78.8** | **29.6** | 69.3 | 62.0 | 46.0 | **73.8** |
| | RocketKV | **100.0** | **100.0** | **98.0** | **92.0** | **100.0** | 42.0 | 80.5 | **56.0** | 78.4 | 23.4 | 69.3 | **64.0** | 46.0 | 73.0 |
| 4096 | Exact-TopK | 100.0 | 100.0 | 96.0 | 92.0 | 98.0 | 44.0 | 84.5 | 52.5 | 82.0 | 20.6 | 70.7 | 62.0 | 44.0 | 72.8 |
| | DuoAttention | 82.0 | 90.0 | 38.0 | 90.0 | 58.0 | 4.0 | 68.5 | 46.0 | 34.8 | 0.4 | 70.7 | 58.0 | **46.0** | 52.8 |
| | SnapKV | **100.0** | 96.0 | 52.0 | **92.0** | 42.0 | 4.0 | 69.0 | 37.0 | 78.4 | 21.4 | 65.3 | **62.0** | 44.0 | 58.7 |
| | Quest | **100.0** | **100.0** | **98.0** | **92.0** | 96.0 | **62.0** | 81.5 | 53.5 | 78.4 | **26.6** | 68.7 | **62.0** | 44.0 | **74.1** |
| | SparQ | **100.0** | **100.0** | 96.0 | **92.0** | 98.0 | 48.0 | 81.5 | 52.0 | **81.6** | 22.4 | **72.0** | **62.0** | 44.0 | 73.0 |
| | HSA | **100.0** | **100.0** | 96.0 | **92.0** | 98.0 | 52.0 | **83.0** | 53.0 | 79.2 | 26.4 | 70.7 | **62.0** | 44.0 | 73.6 |
| | RocketKV | **100.0** | **100.0** | **98.0** | **92.0** | **100.0** | 46.0 | **83.0** | **54.0** | 78.8 | 24.6 | **72.0** | **62.0** | 44.0 | 73.4 |

Table 20: Results of RULER-24K for LongChat-7B-v1.5

| Token Budget | Method | single1 | single2 | single3 | multikey1 | multikey2 | multikey3 | multivalue | multiquery | vt | cwe | fwe | qa-1 | qa-2 | avg |
|---|---|---|---|---|---|---|---|---|---|---|---|---|---|---|---|
| N/A | Full-KV | 100.0 | 100.0 | 98.0 | 90.0 | 84.0 | 18.0 | 60.5 | 38.5 | 79.6 | 5.8 | 70.0 | 50.0 | 36.0 | 63.9 |
| 256 | Exact-TopK | 100.0 | 98.0 | 98.0 | 88.0 | 78.0 | 16.0 | 46.5 | 39.0 | 60.0 | 24.8 | 64.7 | 46.0 | 38.0 | 61.3 |
| | DuoAttention | 0.0 | 0.0 | 2.0 | 4.0 | 0.0 | 0.0 | 2.0 | 3.0 | 1.2 | 0.4 | 0.0 | 20.0 | 24.0 | 4.4 |
| | SnapKV | **100.0** | 20.0 | 0.0 | 16.0 | 0.0 | 0.0 | 0.5 | 1.0 | 59.2 | 1.0 | 28.7 | 40.0 | 28.0 | 22.6 |
| | Quest | 0.0 | 0.0 | 0.0 | 0.0 | 0.0 | 0.0 | 0.0 | 0.0 | 0.0 | 0.0 | 16.0 | 0.0 | 2.0 | 1.4 |
| | SparQ | 0.0 | 0.0 | 0.0 | 0.0 | 0.0 | 0.0 | 0.0 | 0.0 | 0.0 | 0.0 | 0.0 | 0.0 | 2.0 | 0.2 |
| | HSA | 78.0 | 40.0 | 0.0 | 22.0 | 0.0 | 0.0 | 5.0 | 1.0 | 64.4 | **7.8** | 47.3 | 28.0 | 28.0 | 24.7 |
| | RocketKV | **100.0** | **100.0** | 78.0 | 80.0 | 70.0 | 4.0 | 51.5 | 37.5 | 70.4 | 3.8 | 53.3 | 48.0 | 34.0 | 56.2 |
| 512 | Exact-TopK | 100.0 | 100.0 | 98.0 | 88.0 | 78.0 | 22.0 | 49.5 | 38.5 | 63.6 | 19.6 | 68.7 | 46.0 | 36.0 | 62.1 |
| | DuoAttention | 2.0 | 4.0 | 4.0 | 4.0 | 2.0 | 0.0 | 4.5 | 4.0 | 0.4 | 0.6 | **55.3** | 22.0 | 22.0 | 9.6 |
| | SnapKV | **100.0** | 34.0 | 0.0 | 32.0 | 2.0 | 0.0 | 3.5 | 4.0 | 69.2 | 2.4 | 32.0 | 48.0 | 28.0 | 27.3 |
| | Quest | 96.0 | 0.0 | 0.0 | 0.0 | 0.0 | 0.0 | 0.5 | 1.5 | 32.8 | 0.6 | 50.7 | 2.0 | 12.0 | 15.1 |
| | SparQ | 10.0 | 0.0 | 0.0 | 2.0 | 0.0 | 0.0 | 0.0 | 0.0 | 6.4 | 12.2 | 34.0 | 20.0 | 18.0 | 7.9 |
| | HSA | 90.0 | 86.0 | 70.0 | 64.0 | 44.0 | 0.0 | 34.0 | 28.0 | 64.0 | **17.6** | 51.3 | **50.0** | **42.0** | 49.3 |
| | RocketKV | **100.0** | **100.0** | 92.0 | 88.0 | 78.0 | 8.0 | 53.0 | 38.0 | 71.6 | 7.6 | 53.3 | **50.0** | 38.0 | 59.8 |
| 1024 | Exact-TopK | 100.0 | 100.0 | 98.0 | 86.0 | 82.0 | 20.0 | 58.0 | 35.5 | 66.8 | 12.2 | 70.7 | 46.0 | 34.0 | 62.2 |
| | DuoAttention | 4.0 | 4.0 | 4.0 | 6.0 | 2.0 | 0.0 | 4.5 | 4.5 | 0.4 | 0.6 | **58.7** | 16.0 | 24.0 | 9.9 |
| | SnapKV | **100.0** | 60.0 | 2.0 | 54.0 | 4.0 | 0.0 | 8.5 | 7.0 | 71.6 | 4.0 | 38.7 | 50.0 | 30.0 | 33.1 |
| | Quest | 98.0 | 44.0 | 2.0 | 72.0 | 6.0 | 0.0 | 40.0 | 17.0 | 73.6 | 13.2 | 56.0 | 20.0 | 34.0 | 36.6 |
| | SparQ | 86.0 | 62.0 | 0.0 | 20.0 | 12.0 | 0.0 | 4.5 | 5.5 | 76.8 | 27.6 | 26.0 | 48.0 | 34.0 | 31.0 |
| | HSA | **100.0** | 98.0 | 90.0 | **88.0** | 76.0 | 10.0 | 40.0 | 36.5 | 72.0 | 26.2 | 52.0 | 50.0 | **42.0** | 60.1 |
| | RocketKV | **100.0** | **100.0** | 98.0 | 86.0 | **82.0** | 12.0 | 53.0 | 45.5 | 76.4 | 9.4 | 58.0 | **52.0** | 38.0 | **62.3** |
| 2048 | Exact-TopK | 100.0 | 100.0 | 98.0 | 88.0 | 84.0 | 22.0 | 60.0 | 36.5 | 66.8 | 9.4 | 69.3 | 48.0 | 34.0 | 62.8 |
| | DuoAttention | 4.0 | 4.0 | 4.0 | 6.0 | 2.0 | 0.0 | 4.5 | 4.5 | 0.4 | 0.4 | 57.3 | 22.0 | 24.0 | 10.2 |
| | SnapKV | **100.0** | 74.0 | 2.0 | 66.0 | 14.0 | 0.0 | 21.0 | 15.5 | 77.6 | 5.4 | 49.3 | **50.0** | 34.0 | 39.1 |
| | Quest | 90.0 | 74.0 | 42.0 | 68.0 | 50.0 | 0.0 | 57.5 | 24.5 | **79.2** | **34.2** | 56.7 | 44.0 | 36.0 | 50.5 |
| | SparQ | **100.0** | **100.0** | 98.0 | 90.0 | 82.0 | 10.0 | 57.0 | 38.5 | 63.6 | 11.0 | 65.3 | 48.0 | **38.0** | 61.6 |
| | HSA | **100.0** | **100.0** | 92.0 | 86.0 | 78.0 | **28.0** | 53.0 | 39.5 | 76.8 | 21.4 | 62.7 | 46.0 | 36.0 | **63.0** |
| | RocketKV | **100.0** | **100.0** | 98.0 | 88.0 | 80.0 | 16.0 | **60.0** | 43.5 | 75.2 | 10.8 | 62.0 | 48.0 | 36.0 | 62.9 |
| 4096 | Exact-TopK | 100.0 | 100.0 | 98.0 | 90.0 | 84.0 | 20.0 | 63.5 | 36.5 | 68.4 | 7.6 | 72.0 | 48.0 | 36.0 | 63.4 |
| | DuoAttention | 16.0 | 14.0 | 4.0 | 10.0 | 2.0 | 0.0 | 6.5 | 5.0 | 0.4 | 0.6 | 65.3 | 28.0 | 22.0 | 13.4 |
| | SnapKV | **100.0** | 86.0 | 18.0 | 76.0 | 22.0 | 2.0 | 33.5 | 23.5 | 78.0 | 8.6 | 56.7 | 48.0 | 34.0 | 45.1 |
| | Quest | **100.0** | 98.0 | 96.0 | **90.0** | 78.0 | 10.0 | **65.0** | **48.0** | 80.0 | 25.2 | 64.0 | 50.0 | 36.0 | 64.6 |
| | SparQ | **100.0** | **100.0** | 98.0 | **90.0** | 82.0 | 18.0 | 61.0 | 38.0 | 65.2 | 7.0 | **69.3** | 48.0 | **36.0** | 62.5 |
| | HSA | **100.0** | **100.0** | 98.0 | 88.0 | 82.0 | **22.0** | 58.5 | 38.5 | 67.2 | 12.2 | 66.0 | 46.0 | 34.0 | 62.5 |
| | RocketKV | **100.0** | **100.0** | 98.0 | **90.0** | **84.0** | 12.0 | 63.5 | 37.5 | 68.8 | 8.2 | 66.7 | 48.0 | 34.0 | 62.4 |

Table 21: Results of RULER-32K for LongChat-7B-v1.5

| Token Budget | Method | single1 | single2 | single3 | multikey1 | multikey2 | multikey3 | multivalue | multiquery | vt | cwe | fwe | qa-1 | qa-2 | avg |
|---|---|---|---|---|---|---|---|---|---|---|---|---|---|---|---|
| N/A | Full-KV | 100.0 | 94.0 | 80.0 | 78.0 | 68.0 | 4.0 | 58.5 | 31.0 | 47.6 | 38.6 | 70.7 | 22.0 | 34.0 | 55.9 |
| 256 | Exact-TopK | 100.0 | 88.0 | 82.0 | 78.0 | 70.0 | 8.0 | 49.0 | 25.0 | 39.2 | 53.4 | 57.3 | 18.0 | 30.0 | 53.7 |
| | DuoAttention | 0.0 | 2.0 | 4.0 | 4.0 | 0.0 | 0.0 | 1.0 | 2.5 | 0.4 | 0.4 | 0.7 | 20.0 | 26.0 | 4.7 |
| | SnapKV | **100.0** | 14.0 | 0.0 | 8.0 | 0.0 | 0.0 | 0.0 | 0.0 | 19.2 | 0.2 | 16.7 | 24.0 | **32.0** | 16.5 |
| | Quest | 0.0 | 0.0 | 0.0 | 0.0 | 0.0 | 0.0 | 0.0 | 0.0 | 0.0 | 0.0 | 4.7 | 0.0 | 0.0 | 0.4 |
| | SparQ | 0.0 | 0.0 | 0.0 | 0.0 | 0.0 | 0.0 | 0.0 | 0.0 | 0.0 | 0.0 | 0.0 | 0.0 | 0.0 | 0.0 |
| | HSA | 60.0 | 4.0 | 0.0 | 8.0 | 0.0 | 0.0 | 1.0 | 0.0 | **38.0** | 2.0 | 14.0 | 8.0 | 22.0 | 12.1 |
| | RocketKV | **100.0** | **94.0** | **70.0** | **72.0** | **74.0** | **2.0** | **51.5** | **27.5** | 33.6 | **8.8** | **32.7** | **26.0** | 30.0 | **47.9** |
| 512 | Exact-TopK | 100.0 | 90.0 | 84.0 | 78.0 | 72.0 | 8.0 | 51.0 | 27.5 | 43.2 | 52.4 | 65.3 | 20.0 | 30.0 | 55.5 |
| | DuoAttention | 2.0 | 6.0 | 8.0 | 4.0 | 0.0 | 0.0 | 2.5 | 1.5 | 0.0 | 0.4 | 33.3 | 20.0 | 28.0 | 8.1 |
| | SnapKV | **100.0** | 18.0 | 0.0 | 22.0 | 0.0 | 0.0 | 3.0 | 2.5 | 34.4 | 1.8 | 11.3 | **24.0** | 30.0 | 19.0 |
| | Quest | 24.0 | 0.0 | 0.0 | 0.0 | 0.0 | 0.0 | 0.0 | 0.0 | 0.4 | 0.4 | 18.7 | 0.0 | 10.0 | 4.1 |
| | SparQ | 0.0 | 0.0 | 0.0 | 0.0 | 0.0 | 0.0 | 0.0 | 0.0 | 6.8 | 1.8 | 10.0 | 2.0 | 14.0 | 2.7 |
| | HSA | 76.0 | 28.0 | 2.0 | 34.0 | 0.0 | 0.0 | 13.5 | 0.5 | 35.2 | 14.0 | 34.7 | 20.0 | 32.0 | 22.3 |
| | RocketKV | 98.0 | **92.0** | **70.0** | **72.0** | **70.0** | **2.0** | **54.0** | **32.0** | **37.6** | **14.8** | **36.7** | 22.0 | **34.0** | **48.9** |
| 1024 | Exact-TopK | 100.0 | 90.0 | 84.0 | 78.0 | 72.0 | 6.0 | 50.0 | 26.0 | 44.0 | 49.8 | 68.0 | 20.0 | 32.0 | 55.4 |
| | DuoAttention | 2.0 | 6.0 | 8.0 | 4.0 | 2.0 | 0.0 | 2.5 | 2.0 | 0.4 | 0.4 | 33.3 | 20.0 | 24.0 | 8.0 |
| | SnapKV | **100.0** | 36.0 | 0.0 | 32.0 | 4.0 | 0.0 | 8.0 | 5.5 | 38.8 | 3.4 | 14.7 | **24.0** | 32.0 | 23.0 |
| | Quest | 96.0 | 20.0 | 0.0 | 8.0 | 0.0 | 0.0 | 9.5 | 1.5 | 35.6 | 3.4 | 42.0 | 2.0 | 18.0 | 18.2 |
| | SparQ | 54.0 | 8.0 | 0.0 | 0.0 | 0.0 | 0.0 | 0.0 | 0.0 | 20.8 | **33.0** | 22.0 | 18.0 | **34.0** | 14.6 |
| | HSA | 94.0 | 64.0 | 72.0 | 70.0 | 12.0 | 0.0 | **54.0** | 17.5 | **40.0** | 29.2 | 42.7 | 22.0 | 30.0 | 42.1 |
| | RocketKV | **100.0** | **92.0** | **80.0** | **74.0** | **68.0** | **2.0** | 51.5 | **27.5** | **40.0** | 23.2 | **46.7** | 18.0 | **34.0** | **50.5** |
| 2048 | Exact-TopK | 100.0 | 90.0 | 82.0 | 78.0 | 68.0 | 6.0 | 51.5 | 27.0 | 47.2 | 47.6 | 73.3 | 20.0 | 30.0 | 55.4 |
| | DuoAttention | 2.0 | 6.0 | 8.0 | 4.0 | 2.0 | 0.0 | 2.5 | 2.0 | 0.8 | 0.2 | 35.3 | 18.0 | 24.0 | 8.1 |
| | SnapKV | **100.0** | 58.0 | 2.0 | 52.0 | 6.0 | 2.0 | 13.5 | 12.0 | 38.4 | 7.8 | 24.7 | 20.0 | 32.0 | 28.3 |
| | Quest | 88.0 | 90.0 | 0.0 | 46.0 | 14.0 | 0.0 | 50.5 | 17.5 | **50.0** | 26.6 | 52.0 | 20.0 | 26.0 | 37.0 |
| | SparQ | 96.0 | 58.0 | 62.0 | **78.0** | 34.0 | 0.0 | 34.0 | 20.5 | 41.2 | 46.2 | 60.0 | **22.0** | 28.0 | 44.6 |
| | HSA | **100.0** | 86.0 | 86.0 | 76.0 | 44.0 | **4.0** | **52.0** | 30.5 | 40.4 | **46.8** | 58.0 | 20.0 | 28.0 | 51.7 |
| | RocketKV | **100.0** | **94.0** | **88.0** | **78.0** | **70.0** | 2.0 | 50.5 | **31.0** | 39.2 | 31.0 | **62.0** | 20.0 | **34.0** | **53.8** |
| 4096 | Exact-TopK | 100.0 | 94.0 | 82.0 | 78.0 | 70.0 | 6.0 | 53.0 | 30.0 | 46.8 | 41.2 | 71.3 | 20.0 | 32.0 | 55.7 |
| | DuoAttention | 6.0 | 8.0 | 8.0 | 6.0 | 2.0 | 2.0 | 2.5 | 2.5 | 1.6 | 0.0 | 49.3 | 18.0 | 26.0 | 10.1 |
| | SnapKV | **100.0** | 76.0 | 6.0 | 70.0 | 10.0 | 0.0 | 22.0 | 17.5 | 40.8 | 12.6 | 38.0 | 18.0 | 32.0 | 34.1 |
| | Quest | **100.0** | **94.0** | 66.0 | **78.0** | 54.0 | 0.0 | **58.0** | 27.0 | 45.2 | 44.8 | 57.3 | **22.0** | 26.0 | 51.7 |
| | SparQ | **100.0** | **94.0** | 84.0 | **78.0** | 58.0 | 2.0 | 49.5 | 26.5 | **45.6** | **47.4** | **72.0** | 20.0 | 32.0 | 54.5 |
| | HSA | **100.0** | 90.0 | **90.0** | 76.0 | 70.0 | **6.0** | 49.5 | **30.5** | 43.2 | 44.2 | 66.0 | 20.0 | 32.0 | 55.2 |
| | RocketKV | **100.0** | 92.0 | **90.0** | **78.0** | **74.0** | **6.0** | 50.5 | 28.0 | 44.8 | 38.6 | 69.3 | 18.0 | **34.0** | **55.6** |

