# OpenReview forum: "RocketKV: Accelerating Long-Context LLM Inference via Two-Stage KV Cache Compression"
_ICML.cc/2025/Conference — ICML 2025 poster_

### Official Review · Reviewer_eFmJ · 2025-03-14

**Overall Recommendation:** 2

**Summary:**

RocketKV is a training-free KV cache compression strategy designed to optimize the inference efficiency of long-context LLMs during the decode phase. The main challenge it addresses is the exponential memory overhead due to KV cache storage, which scales with sequence length. The method is empirically validated on Mistral-7B, LLaMA-3.1-8B, and LongChat-7B, showing up to 3× speedup and 31% peak memory reduction on NVIDIA H100 GPUs, while preserving accuracy across long-context benchmarks (LongBench, Needle-in-a-Haystack, RULER). RocketKV outperforms SnapKV, Quest, SparQ, and DuoAttention, achieving near full-KV accuracy at significantly lower memory footprints.

**Claims And Evidence:**

Yes.

**Essential References Not Discussed:**

No.

**Experimental Designs Or Analyses:**

The experiments are sound but could benefit from:
- Latency breakdown: token retrieval and sparse attention computation.
- Ablation studies on kernel size impact.

**Methods And Evaluation Criteria:**

Yes.

**Other Comments Or Suggestions:**

Regarding the needle-in-the-haystack benchmark, since RocketKV evicts previous tokens, it is unclear whether it might also evict the needle—despite still producing the correct output. Could the authors provide an analysis of which tokens RocketKV evicts and whether the needle is among them?

**Other Strengths And Weaknesses:**

Strengths
- Hardware-Friendly Optimization.
- Significant Performance Gains in Speed and Memory Efficiency.

Weaknesses:
- Novelty is not enough. The two main stages come from SnapKV + QUEST.
- Potential Sensitivity to Kernel Size Selection in SnapKV++. The adaptive pooling kernel size in SnapKV++ is empirically determined, raising concerns about its generalizability to unseen domains.
- Evaluation on Only One Hardware Setup (NVIDIA H100). Test on cheaper GPUs such as A100.
- Lack of Direct Latency Comparison with Alternative Sparse Attention Methods.

**Questions For Authors:**

See Strengths And Weaknesses

**Relation To Broader Scientific Literature:**

- Advancing Sparse Attention Methods for LLMs

**Theoretical Claims:**

No theory.

---

> ### Author Rebuttal · Authors · 2025-04-01
>
> We thank the reviewer for the valuable feedback and finding RocketKV results promising.
>
>
> **Novelty: RocketKV is SnapKV + QUEST:**
> As discussed in the paper, existing methods for KV cache compression typically fall into two categories: permanent KV token eviction and dynamic KV token selection. We would like to clarify that the primary novelty of our work is that we introduce a two-stage approach that effectively combines the strengths of both paradigms into a single framework. While we propose SnapKV++ for the first stage and hybrid attention for the second stage, they can be directly replaced with various other methods of the same category.
>
> In terms of SnapKV++, we admit that our improvement over SnapKV is not significant which is reflected in the name (this is also recognized by Reviewer utBU). However, it is still much more effective than the original SnapKV method as demonstrated in our ablation study (Section 4.4 and Appendix B.1).
>
> We would like to point out that our hybrid attention method is NOT the same as QUEST. QUEST relies on K tensor reduction along the sequence dimension to conduct approximate attention, while our hybrid attention method relies on K tensor reduction along both sequence and head dimensions. This two-dimensional reduction scheme can achieve much higher accuracy than one-dimensional reduction at a given compression ratio. For example, with a compression ratio of 16, hybrid attention can evenly split it into a compression ratio of 4 at each dimension, introducing much lower accuracy loss compared to directly compressing the sequence dimension by 16x in QUEST. This is further confirmed in the ablation study (Section 4.4.3 and Appendix B.1) where we demonstrate the standalone hybrid attention method consistently outperforms QUEST and SparQ.
>
> Overall, we believe our work introduces sufficient novelty and is qualified as a top-tier conference publication.
>
> **Potential Sensitivity to Kernel Size in SnapKV++:**
> We thoroughly examined the impact of pooling kernel size in the ablation study (Section 4.4.2 and Appendix B.1). While the adaptive pooling size is indeed empirically determined based on accuracy results on the RULER benchmark as shown in Figure 7, we found this simple method is quite effective and generalizes well on other benchmarks as shown in Figure 10 in Appendix B.1. The insight we observe from this study is that tasks with longer sequence lengths usually perform better with larger kernel sizes, which can provide practical guidelines for better pooling kernel size selection in future work.
>
> **Test on Cheaper GPUs:**
> Below are the end-to-end speedup numbers of RocketKV over Full KV cache running Llama3.1-8B on A100 with 256 token budget.
> | Sequence length | 16K  | 32K  | 64K  | 96K  |
> |-----------------|------|------|------|------|
> | Speedup         | 1.3x | 1.5x | 2.7x | 3.6x |
>
> Compared to efficiency data on H100 shown in Figure 5(a), we can see that the maximum speedup on A100 is 20% higher (3.6x versus 3x). This is because A100 has a lower memory bandwidth to compute ratio compared to H100. As a result, LLM inference execution is more memory-bound on A100 and can benefit more from memory traffic savings of KV cache offered by RocketKV. We believe the speedup of RocketKV will be even higher on cheaper GPUs such as RTX 4090/5090 since they are not equipped with High Bandwidth Memory (HBM). Unfortunately, these GPUs also have much smaller memory capacity which prevents us from conducting long-context experiments on them. Notice that the memory savings are the same between H100 and A100 so we didn’t show them here.
>
>
> **Lack of Direct Latency Comparison:**
> Since different sparse attention methods are usually implemented under different frameworks with different levels of code optimizations, it is difficult to provide an apple-to-apple comparison between them. In this work we use the token budget to estimate memory traffic, including attention score approximation. For example, with RocketKV at a token budget of 256, half is used for attention approximation (Steps 1 and 2 in Section 3.4) and half for sparse attention (Step 3). Thus, the token budget itself can directly reflect attention latency in highly-optimized implementations since attention operations are mostly memory-bound.
>
> **Is Needle Among Evicted Tokens:**
> RocketKV evicts KV cache tokens independently across attention heads and layers, making it unclear if the needle's tokens are consistently retained. Additionally, later layers can spread the needle's information to other positions through attention, reducing the effectiveness of using needle positions to track information loss.

---

### Official Review · Reviewer_6pQ8 · 2025-03-14

**Overall Recommendation:** 3

**Summary:**

This paper combines the advantages of permanent KV token eviction and dynamic KV token selection. It uses a two-staged kv cache compression method to give strong results and shows that it reduces GPU memory usage.

**Claims And Evidence:**

I mostly agree with it. However, given that "permanent KV token eviction" could lose important information in the tokens, I do not quite understand why combining two kinds of kv cache compression could be lossless.

In this sense, I think two things can be done.

1. add more benchmarks to prove the losslessness.
2. use other architectural models? like deepseek MLA models? this may be too expensive. But I really ponder about the performance of RocketKV on the reasoning models, given they have extremely long reasoning tokens.

**Essential References Not Discussed:**

No.

**Experimental Designs Or Analyses:**

See above coments, thanks.

**Methods And Evaluation Criteria:**

As above, I think we should add more model types and widely used benchmarks.

**Other Comments Or Suggestions:**

Nice work! Hope to see further improvements.

**Other Strengths And Weaknesses:**

see above, I am pondering about the real inference cost, like the time overhead introduced by RocketKV, and whether the overhead can be resolved. If it works quickly and losslessly, it would be perfect.

**Questions For Authors:**

No

**Relation To Broader Scientific Literature:**

Lossless KV cache compression is what a lot of people need. Have the authors tried to implement it on widely used inference engines, say sglang, vllm?

**Theoretical Claims:**

No proof.

---

> ### Author Rebuttal · Authors · 2025-04-01
>
> We thank the reviewer for finding our work interesting and providing valuable suggestions.
>
> **Add More Benchmarks to Prove the Losslessness:**
> We would like to clarify that RocketKV is not a lossless approach, as evident from the provided accuracy results. And none of the other methods we compared are 100% lossless including the Exact-TopK method because they all replace dense attention operations with sparse attention. The primary goal is to achieve comparable (but may not exactly the same) accuracy to Full KV cache with much smaller KV token budgets. We have evaluated RocketKV across three models and three datasets under various token budgets to demonstrate the superiority of our method against existing work on achieving minimal accuracy loss with an extremely high KV cache compression ratio (up to ~500x in our evaluation). Compared to other existing works, we believe we have provided a sufficiently comprehensive evaluation. Moreover, we have added some additional evaluation on the recent SCBench in multi-turn scenarios and demonstrated that RocketKV outperforms other methods by a significant margin. Please refer to our response to reviewer utBU for more details.
>
> **Other Architectural Models Like Deepseek MLA or the Reasoning Models:**
> Thank you for your suggestion. We are also interested to see how RocketKV and other KV cache compression methods perform on top of DeepSeek MLA models. Note that none of the other works on KV cache compression have done it before. Since RocketKV is fully compatible with GQA, we believe it can be applied to MLA with simple modifications (MLA can be considered as a variant of GQA where all attention heads in a layer share the same KV cache tensors). Considering RockeKV outperforms current methods over various models and datasets, we expect to see similar trends when we evaluate various methods on MLA. Unfortunately, due to time and resource constraints during the rebuttal period, we could not add either DeepSeek V3 or R1 model into our evaluation and have decided to leave it for future work.
>
> **The Real Inference Cost:**
> We have provided end-to-end inference results of RocketKV on an NVIDIA H100 GPU showing significant latency speed-up and memory saving against Full KV cache that already includes the time overhead of RocketKV (Section 4.3).

---

### Official Review · Reviewer_eef9 · 2025-03-16

**Overall Recommendation:** 2

**Summary:**

This paper introduces RocketKV, a two-stage KV cache compression approach. The first stage applies permanent KV eviction through adaptive pooling and GQA-compatible SnapKV methodology, while the second stage efficiently retrieves necessary KV components dynamically based on queries via a hybrid attention mechanism. This approach effectively retrieves essential key-value pairs during each decoding step, maintaining high accuracy. The proposed method demonstrates superior accuracy compared to other KV cache compression techniques (SnapKV, Quest, SparQ, etc.) while also achieving memory savings and end-to-end acceleration.

## Update After Rebuttal

After reading other reviewers' opinions and the authors' rebuttal, I have gained a better understanding of RocketKV's contribution and methods. However, I still have remaining concerns regarding the weaknesses I initially pointed out. First, regarding presentation, while the authors' additional explanations resolved many of my questions, considering the state of the initial submission, I believe this paper still requires significant improvements in structure and writing.

Second, regarding contribution, as reviewer eFmj mentioned, I am not fully convinced about the novelty of RocketKV compared to SnapKV and QUEST. This concern is related to the aforementioned presentation issues. For example, looking back at the paper, it is difficult to consider the heuristic search for adaptive pooling size (a differentiating point from SnapKV++) as a fundamental contribution. Additionally, it is challenging to understand why GQA compatibility improves performance in Figure 6 (it is difficult to find even minimal insight on this). Furthermore, rather than briefly mentioning the concept of QUEST in section 3.1, the paper should provide more detailed explanations and insights about the foundational methodology. Such comprehensive understanding would help readers better appreciate the distinguishing points of RocketKV compared to QUEST.

Overall, while I now better understand the core of RocketKV—ensuring accuracy and achieving better efficiency through two-stage KV cache control, as the authors claim—I find it difficult to consider this paper suitable for ICML publication in its current state, even taking into account the additional rebuttal results. This is mainly due to insufficient presentation and structure. Considering all of my points, I will update my score from 1 to 2.

**Claims And Evidence:**

The paper lacks sufficient explanation of the proposed methodology, making it difficult to thoroughly examine the evidence supporting its claims. For instance, the Stage 2 Hybrid Attention appears to be written with the assumption that readers are already familiar with Quest and SparQ, as detailed explanations are notably absent. It's challenging to understand from the main text alone why a sign function is applied to the sum of q, or how approximate attention scores are calculated by retrieving from K_max and K_min based on these values.

Regarding the proposed SnapKV++, the GQA-compatible aggregation method has already been widely used in recent approaches (e.g., AdaKV - https://github.com/FFY0/AdaKV). Similarly, the adaptive pooling method relies on rule-based, heuristic pooling sizes determined by sequence length without clear justification. Despite potentially contributing to higher accuracy, these two contributions are difficult to recognize as novel and central to the paper's contribution.

**Essential References Not Discussed:**

No issue

**Experimental Designs Or Analyses:**

The experimental design appears sound, though the paper lacks detailed analysis.

**Methods And Evaluation Criteria:**

The paper presents performance comparisons across various long context benchmarks at different compression ratios.

**Other Comments Or Suggestions:**

Experimental results (especially ablation studies) seem to occupy excessive space. This could be adjusted using tables or other methods to allow for more comprehensive background explanations. If my overall perspective is somewhat misguided or differs from other reviewers' opinions, I am open to reconsidering my assessment.

**Other Strengths And Weaknesses:**

A strength of the paper is that the proposed method demonstrates high performance across various compression ratios. However, there are significant weaknesses in presentation that cannot be overlooked. Without detailed background on closely related works like Quest and SparQ, readers would struggle to properly understand Stages 1 and 2. Furthermore, there appears to be no analysis in the main text explaining why this method is necessary.

While the paper makes valid contributions regarding performance improvements, considering this is a submission to a top-tier conference, insights and detailed explanations supporting the proposed methodology are essential for readers. Despite the performance results, the current state of the paper makes it difficult to recommend for acceptance. Below are aspects that should be addressed in more detail:

- How exactly is Exact-TopK measured? Does it involve computing Full KV during prefill and retaining only the highest attention scores? Then what would be the difference between SnapKV and Exact-TopK?
- Does Figure 2's illustration of the second stage indicate that different KV tokens are retrieved at each decoding step?
- The process of calculating approximate attention scores using element-wise Max, Min, and the sign result of sum of q is difficult to understand from this paper alone.
- Is the Page always divided into 4 in the dimension? Is there a specific rationale for this approach?

**Questions For Authors:**

See weakness

**Relation To Broader Scientific Literature:**

The proposed KV cache compression methodology seems closely related to H2O and SnapKV, while the dynamic query-based KV retrieval approach appears closely connected to Quest.

**Theoretical Claims:**

There appear to be no theoretical claims in the paper. The adaptive pooling size is presented heuristically based on empirical experimental results, offering both size candidates and determination methods.

---

> ### Author Rebuttal · Authors · 2025-04-01
>
> Thank you for your comprehensive review and providing constructive feedback.
>
> **Lack of Methodology Explanation:**
> Due to Space constraints, we decided to prioritize RocketKV’s performance results, resulting in a briefer method explanation. In the final version, we will move certain ablation studies to the appendix to provide a more comprehensive, self-contained description of our approach as you suggested.
>
> **Why RocketKV Is Necessary:**
> Figure 1 illustrates that existing KV cache compression methods struggle to match the accuracy of Exact-TopK on Qasper under low token budgets.
> For further motivation, we analyzed a random attention head (layer 31 head 0 from Llama 3.1-8B) and calculated the maximum number of unique KV tokens used by Exact-TopK attention across all decoding steps in the Qasper benchmark. As shown in the table below, to keep all important KV tokens (used by at least one TopK attention across all decode steps), a permanent KV cache eviction method needs to keep 2197 out of 6110 tokens (where 6110 is the total seq. length) when K=256. Moreover, a dynamic token selection algorithm needs to accurately select a small set of TopK tokens out of a large set (e.g. 256 out of 6110 when K=256), which is a challenging task. An ideal solution is to keep only important KV tokens and then conduct dynamic token selection among them. Thus, we propose a two-stage approach that first retains only important tokens and then performs dynamic selection on this smaller set. This fusion evicts unimportant tokens and makes the dynamic selection more accurate, motivating our RocketKV design.
>
> | TopK value                    | 256  | 512  | 1024 | 2048 | 4096  |
> |------------------------------|------|------|------|------|-------|
> | Max num of unique TopK tokens  | 2197 | 3223 | 5229 | 8272 | 11993 |
> | Total num of tokens            | 6110 | 6110 | 17789| 17789| 17789 |
>
> **Explanation of Calculating Approximate Attention Scores:**
> RocketKV’s hybrid attention focuses on accurately pointing KV tokens with the TopK attention scores. We split the K tensor into pages of consecutive tokens. Within each page, we record element-wise min *(Kmin)* and max *(Kmax)* to estimate the upper bound of attention scores for a query *q*. Specifically, we compute *max⁡(q x Kmin⁡, q x Kmax⁡)* for each page to approximate highest possible attention scores within a page, then select pages with higher scores. To further reduce the approximation overhead, we only calculate on partial positions along the head dimensions where the magnitude of *q* is large and ignore other positions, and only fetching either from *Kmin* or *Kmax* at a given position based on the corresponding sign of q as shown in Figure 3. Hence, we approximate attention scores via both sequence and head dimension reductions. To be fully compatible with GQA, we select based on the sum of *q* or *|q|* at group dimension as needed to guarantee that all attention heads within a group are making the same selection at each step.
>
> **Measurement of Exact-TopK:**
> As explained in Appendix A-3, Exact-TopK is an oracle-based method that assumes prior knowledge of token importance and dynamically chooses TopK KV tokens for each attention head and decoding step. By contrast, SnapKV permanently prunes tokens of the input prompt via one-time TopK filtering.
>
> **Are Different KV Tokens Retrieved at Each Decoding Step?**
> Yes. Our second-stage hybrid attention dynamically selects KV tokens at each decoding step, aiming to pick those most relevant to the current query vector.
>
> **Is Page Always Divided Into 4?**
> No. Page size depends on the overall compression ratio. For a token budget of 512 and a total length of 128K, the compression ratio is 256x. We then split this ratio evenly between the first and second stages (16x each), and again between sequence and head dimensions (4x each) in hybrid attention as mentioned in Section 3.5. Consequently, the page size is four tokens in that scenario, but it can vary for other compression ratios.
>
> **Novelty of SnapKV++:**
> Thank you for pointing out that a similar GQA-compatible enhancement for SnapKV has already been proposed in AdaKV. This feature was added to GitHub in Nov.'24 and the arXiv paper in Jan.'25. We proposed GQA enhancement for SnapKV independently and concurrently. We are happy to give AdaKV credit for this concurrent contribution in the final version. However, per reviewer instructions outlined in ICML website, authors should not be held responsible for papers that were made public within 4 months of the submission deadline. We are discussing with AC to see whether we should give up GQA enhancement as a claimed contribution. Importantly, this does not diminish our core novelty: a two-stage KV cache compression framework uniquely combining permanent KV eviction with dynamic token selection, which is broadly generalizable to various compression methods at each stage.
>
> Regarding adaptive pooling size, please see our response to reviewer eFmJ.

---

### Official Review · Reviewer_utBU · 2025-03-27

**Overall Recommendation:** 4

**Summary:**

This paper presents RocketKV, a method that leverages observation made upon existing permanent and dynamic token eviction. Specifically, RocketKV aims to conduct a permanent eviction with a large budget first and refine it to target a budget with fine-grained dynamic evictions.

**Claims And Evidence:**

Yes.

**Essential References Not Discussed:**

-

**Experimental Designs Or Analyses:**

Yes. All three of them.

**Methods And Evaluation Criteria:**

The needle dataset seems to be basic if it strictly follows the GKamradt setup (with the background filler being the reputation of texts). This needle setup is known to be weak per findings like [1] and [2] and it is recommended for the authors to opt for a more comprehensive needle setup. A common practice is to adopt PGraham Essay as background and a passkey-like needle, as done in [2]. This is possibly not much of a concern due to the adaptation of RULER, which is a much more standardized needle task, but it should still be considered.

Further, while LongBench and RULER are popular datasets for long context evaluation, these alone might be a bit outdated by today's standard. As the authors are familiar with SnapKV and Razor/DuoAttention, one major challenge of SnapKV is it is query-position sensitive, which is showcased in literature like SCBench. I am interested in seeing if the proposed method can perform well on such multi-round datasets.

Last, per A.2, the LongBench input is truncated even for long context-capable models like Llama 3.1. It is recommended to have non-truncated input for such models or adopt a longer dataset like $\infty$Bench.


[1] InfLLM: Training-Free Long-Context Extrapolation for LLMs with an Efficient Context Memory
[2] KV Cache Compression, But What Must We Give in Return? A Comprehensive Benchmark of Long Context Capable Approaches
[3] SCBench: A KV Cache-Centric Analysis of Long-Context Methods

**Other Comments Or Suggestions:**

Nothing major but it might be worth noting the proper quotation marks for LaTeX are `` and ''. It seems like sometimes only the right quotation marks are used.


---

**Post Rebuttal Update**



I thank the authors for the added results and clarifications. Without memory efficiency gains, RocketKV-MT remains a rather pilot study akin to methods like DeepSeek NSA (though one may argue NSA actually generates *more* KV cache), and while I agree there are channels for optimization in a disaggregated service scenario, significant consideration would incur under that setting. I hope to see a faithful discussion of RocketKV-MT in this regard in the updated manuscript. Further, on the presentation side, the current writing of SnapKV++ can also use a bit more background for unfamiliar readers. I understand it is a small module of the presented method, but Quest/SparQ/(and even AdaKV)'s contribution should also be explicitly stated.

Back to the evaluation note: I encourage authors to construct more complete tests on newer, more challenging benchmarks like SCBench, LongBench v2, and HELMET with more compression settings and baseline features. KV cache studies have been long limited under simple needle tasks and LongBench, which is no longer a proper standard for modern KV cache evaluation. While that might reveal gaps between full precision and the compressed model, making many authors reluctant to feature such evaluations, I firmly believe such gaps would guide future development and should be highlighted more than perfect results. **SnapKV should also be fully featured, by aligning with SCBench's report setting it should be doable.**

Furthermore, I wonder why DuoAttention underperforms. It relies on the mechanistic proper of attention heads, and it seems unintuitive to perform badly under a multi-round scenario if the setting is right. Some investigations in this regard would be appreciated.

Last, regarding the LongBench truncation, apologize for reading A.2. wrong, the current setting sounds proper. The needle test is also proper if it is following [2] (and as a kindly reminder, this paper is wrongly cited).

I now improved the rating to 4 as promised.

**Other Strengths And Weaknesses:**

See other sections for weaknesses. Generally I am very fond of this work as it is:
- Nicely written
- Makes some clear and correct taxonomy of existing works, and pay proper tribute when the improvement is incremental in nature (e.g., SnapKV++).
- Goes for a simple but effective approach when such a design works.
- Solid evaluation on executed ones — I might have some reservations regarding its dataset comprehensiveness, but for the conducted experiments, the model and setting coverage seem nicely done.

I am open to improve the rating upon a satisfactory rebuttal.

**Questions For Authors:**

-

**Relation To Broader Scientific Literature:**

-

**Theoretical Claims:**

-

---

> ### Author Rebuttal · Authors · 2025-04-01
>
> We sincerely appreciate the reviewer’s insightful feedback and recognition of our work in several aspects.
>
> **Needle Dataset:**
> Our needle dataset already follows reference [2] mentioned by the reviewer which adopts PGraham Essay as background and a passkey-like needle as explained in Appendix A.2.
>
>
> **SCBench Results:**
> This is a great suggestion. We have conducted additional experiments with SCBench under multi-turn mode and below are the preliminary results. Due to time and resource constraints, we only present results with 4K token budget on Llama3.1-8B-Instruct, but do plan to show more comprehensive SCBench results in the final version of the paper. Notice that some results for SnapKV are missing because it only compresses the KV cache of the input prompt but not the generated tokens, therefore it cannot meet the 4K token budget requirement for tasks with more than 4K generated tokens.
>
> Our results below demonstrate that RocketKV still greatly outperforms other baseline methods in multi-turn scenarios, especially for string retrieval tasks. However, there is still a noticeable gap between RocketKV and Exact-TopK, showing room for further improvement. As you already pointed out, we believe this is due to the limitation of SnapKV which relies on the query at the end of the input prompt to filter out unimportant KV tokens. In multi-turn scenarios, the unimportant KV tokens evicted by the previous turns might be essential for queries in later turns, and this could cause a significant accuracy drop in later turns. To address this challenge, we propose a variant of RocketKV called RocketKV-MT where we do not perform permanent KV token eviction in SnapKV++ but keep them all for later turns. However, the decode phase is still restricted to perform dynamic selection on the filtered KV tokens by SnapKV++. For instance, if SnapKV++ identifies 4K important tokens out of 16K input tokens in the prefill phase, all 16K KV tokens are kept in memory but the hybrid attention method only selects among the 4K important tokens in the decode phase. In the next turn, all 16K input tokens are added as prefixes in the prefill phase and SnapKV++ will perform another round of filtering based on unfiltered history. By doing this, RocketKV-MT still achieves the same performance benefit as RocketKV but does not introduce memory storage savings. Our results below demonstrate that RocketKV-MT performs much better than RocketKV in SCBench and achieves almost the same accuracy as Exact-TopK. We would like to also point it out that RocketKV-MT would be a great fit for disaggregated serving where different GPUs are used for prefill and decode (see NVIDIA’s Dynamo[4] and Moonshot AI’’s Mooncake[5]), as it can keep the full KV cache on the prefill node and send only the filter KV cache to the decode node in each turn. This would lead to not only memory storage savings on the decode node but also significant communication traffic savings between the prefill and decode node.
>
> | Method           | Retr.String | Retr.Semantic | Global | Multi-task | AVG.  |
> |------------------|-------------|----------------|--------|-------------|--------|
> | Full-KV          | 49.3        | 40.9           | 36.3   | 64.7        | 50.0   |
> | Exact-TopK       | 43.4        | 40.0           | 36.4   | 63.8        | 47.9   |
> | DuoAttention     | 0.1         | 25.2           | 34.0   | 12.7        | 20.8   |
> | SnapKV           | 2.9         | N/A            | 35.4   | N/A         | N/A    |
> | Quest            | 8.6         | 26.0           | 28.0   | 23.2        | 23.7   |
> | SparQ            | 3.0         | 27.9           | 29.1   | 28.7        | 24.1   |
> | RocketKV         | 36.9        | 26.2           | 35.9   | 31.8        | 35.2   |
> | RocketKV-MT      | **47.8**    | **37.3**       | **37.0** | **61.2**    | **47.8** |
>
> [4] NVIDIA Dynamo, https://github.com/ai-dynamo/dynamo
>
> [5] Mooncake: A KVCache-centric Disaggregated Architecture for LLM Serving
>
> **LongBench Input Truncation:**
> We follow the original setting from LongBench to truncate the input if the sequence length is longer than the maximum sequence length enabled by the model. However, the maximum sequence length across all tasks in LongBench is within 30K tokens (as shown in Table 3 of Appendix D in the SnapKV paper), while all three models we evaluated have a maximum sequence length of at least 32K tokens. Therefore, no truncation occurs in practice during our evaluation.
>
> **Proper Quotation Marks:**
> Thank you for pointing this out, we will correct those quotation marks in the final version of the paper.

---

### Decision · Program_Chairs · 2025-05-01

**Decision:**

Accept (poster)

**Comment:**

RocketKV introduces a practical and effective two-stage KV cache compression framework for long-context LLM inference that yields strong efficiency gains while maintaining accuracy. The paper presents a comprehensive evaluation across multiple models and tasks, and features thoughtful ablation studies that support the design choices. Despite limited novelty, the authors demonstrate clear improvements through a synergistic integration and introduce unique elements like hybrid attention over both sequence and head dimensions. While some reviewers noted concerns about presentation clarity and comparisons with combined baselines, the rebuttal offers detailed clarifications and new results, partially mitigating these concerns. Given the practical impact, clarity of exposition, and responsiveness during the discussion, I recommend acceptance.